# Hybrid Architectures for Language Models: Systematic Analysis and Design Insights

## Abstract

Recent progress in large language models demonstrates that hybrid architectures–combining self-attention mechanisms with structured state space models like Mamba–can achieve a compelling balance between modeling quality and computational efficiency, particularly for long-context tasks. While these hybrid models show promising performance, systematic comparisons of hybridization strategies and analyses on the key factors behind their effectiveness have not been clearly shared to the community. In this work, we present a holistic evaluation of hybrid architectures based on inter-layer (sequential) or intra-layer (parallel) fusion. We comprehensively evaluate these designs across multiple dimensions: language modeling and downstream task performance, long-context capabilities, scaling analysis, and training and inference efficiency. By investigating the core characteristics of their computational primitive, we identify the most critical elements for each hybridization strategy and further propose optimal design recipes for hybrid models. Our comprehensive analysis provides practical guidance and valuable insights for developing hybrid language models, facilitating the optimization of architectural configurations.

## 1 Introduction

Recent language models (Llama Team, 2024; Microsoft Research, 2024; DeepSeek-AI, 2024; OpenAI et al., 2024; Qwen Team, 2025; Gemini Team, 2025) have demonstrated strong scalability and human-like performance across a wide range of tasks. Most of these models are based on the Transformer architecture (Vaswani et al., 2017), which alternates self-attention and feed-forward layers. However, the self-attention mechanism exhibits quadratic complexity with respect to input sequence length, leading to slow inference and a substantial memory footprint. To address these limitations, a new class of computational primitives—state space models (SSMs)—has emerged, inspired by signal processing (Gu et al., 2020; 2021a;b; 2022; Fu et al., 2022; Gu & Dao, 2023; Dao & Gu, 2024; Yang et al., 2024). These architectures scale more efficiently to long sequences by compressing prior context into a finite-dimensional state. Among these, the Mamba model (Gu & Dao, 2023; Dao & Gu, 2024) stands out for being competitive with Transformer on language modeling, while also accelerating training speed through work-efficient parallel scan algorithms (Blelloch, 1990; Smith et al., 2022).

These new primitives expand the architecture design space, opening up new possibilities for hybrid models that leverage the strengths of different architectural choices (Glorioso et al., 2024b; Ren et al., 2024; Lieber et al., 2024; Wang et al., 2024a; Poli et al., 2024; Dong et al., 2024; Ren et al., 2025; Basant et al., 2025). Notably, mixing Transformer and Mamba blocks often outperforms homogeneous architectures, while maintaining high efficiency. Most prior work on hybrid models has focused on the sequential interleaving of standard Transformer and Mamba blocks—an *inter*-layer hybrid approach (Glorioso et al., 2024b; Ren et al., 2024; Jamba Team et al., 2024; Hunyuan Team et al., 2025). This practical strategy allows for a balance between model quality and throughput by adjusting the ratio of quadratic (Transformer) to linear (Mamba) attention blocks. In addition, a few *intra*-layer hybrid models have been proposed, which fuse the two primitives in parallel within individual layers. These approaches use either head-wise (Dong et al., 2024; Xiao et al., 2025) or sequence-wise splits (Zhang et al., 2024; Li et al., 2025b) to further combine the benefits of both architectures at a finer granularity. Figure 1a summarizes which attention primitives each hybridization type can use.

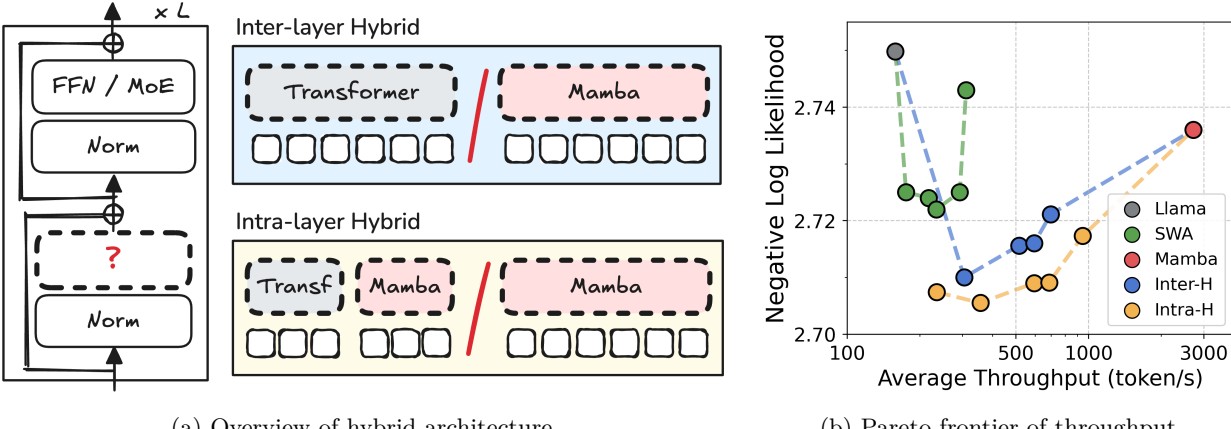

(a) Overview of hybrid architecture
(b) Pareto frontier of throughput

Figure 1: **(a) Two hybridization strategies construct attention using either inter-layer (Transformer / Mamba) or intra-layer (Transformer+Mamba / Mamba) blocks.** Specifically, the inter-layer hybrid model is built by choosing between Transformer and Mamba blocks. In contrast, the intra-layer hybrid model chooses between an intra-hybrid block and a standard Mamba block. Inside this intra-hybrid block, the input heads are evenly split and processed by half-sized Transformer and Mamba primitives in parallel. Varying the capacity ratios of these blocks controls the degree of hybridization. **(b) The hybrid architectures achieve superior quality-throughput trade-offs compared to homogeneous architectures.** Negative log likelihood (i.e., loss) is measured on the DCLM validation set, and decoding throughput is averaged over a batch size of 4 across total lengths of 2K, 4K, 8K, 16K, and 32K, with the prompt length fixed at 512. All models have 1B parameters and are trained with the same FLOPs budget of 4.5e20 and 8K context length. For sliding window attention (SWA) and Mamba hybrid models, we connect results for different block ratios (1:0, 1:1, 1:3, 1:5, 1:12, 0:1)—where each ratio denotes (Transformer or Intra-hybrid : SWA or Mamba)—with dashed lines.

Despite the emergence of various hybrid architectures, the current literature lacks openly shared, in-depth analysis of hybridization strategies, making it difficult to understand their relative strengths and design trade-offs (Sun et al., 2025). In particular, most prior works focus on introducing specific hybrid architectures rather than providing detailed, systematic comparisons across possible hybridization approaches. As a result, the key intuitions driving final design choices, as well as the relative merits of different strategies from multiple perspectives, remain open questions for the community.

In this work, we address these research questions by conducting a holistic evaluation of hybrid architectures via comprehensive, multi-faceted comparisons. Specifically, for *model architecture* choices, we compare inter-layer and intra-layer hybridization strategies with Transformer (Llama Team, 2024), Mamba (Dao & Gu, 2024), and striped models of global and sliding window attention (SWA) (Beltagy et al., 2020; Gemma Team et al., 2025) with attention sink (Xiao et al., 2023). From *quality* perspectives, we identify optimal design choices for hybrid models by providing key insights into critical architectural considerations, based on language modeling perplexity and downstream task accuracy (§4.1), and extensive ablation studies (§4.5 and §4.6). Additionally, we analyze long-context retrieval performance (Rae et al., 2019a; Kamradt, 2023) with the characteristics of each computational primitive (§4.3). Our exploration of Mixture-of-Experts (Fedus et al., 2022; DeepSeek-AI, 2024) and compute-optimal scaling (Kaplan et al., 2020; Hoffmann et al., 2022) further offers practical guidance for scaling up hybrid models (§4.4). On the *efficiency* front, we perform an in-depth efficiency comparison of both training and inference, measuring training time, memory usage, inference throughput, and cache size (§4.2). Based on our extensive experiments, we summarize the principal insights regarding hybrid architectures as follows:

**[Quality]** Both hybrid model types (up to 3B scales and 32K context) outperform homogeneous architectures by up to 2.9% in few-shot accuracy and even surpass widely adopted SWA models. Intra-layer hybrid models take the pareto-frontier of model quality and efficiency (see Figure 1b and Table 2).

**[Quality]** Hybrid architectures overcome the locality biases inherent in Mamba and SWA models, consistently outperforming the standard Transformer on various downstream tasks (see Table 3).

**[Quality]** While baselines show poor in-context retrieval and length generalization, all hybrid architectures achieve robust and superior long-context retrieval (see Figures 3c and 4).

**[Quality]** Hybrid models are fully compatible with MoE structures and achieve an optimal compute-scaling slope between those of Transformer and Mamba (see Table 4).

**[Efficiency]** By fully leveraging Mamba's efficiency strengths (i.e., linear complexity with respect to sequences), hybrid models achieve faster end-to-end training time, lower peak memory usage (see Figure 2), and higher inference throughput with smaller cache sizes (see Figures 1b, 3a, and 3b).

**[Design]** Insights about optimal block ratios, ordering of computational primitives for hybrid architectural configurations are thoroughly explored (see Tables 5 and 6).

## 2 Background

**Transformer architecture.** The Transformer architecture (Vaswani et al., 2017) consists of two main components: the multi-head attention (MHA) module and a feed-forward network (FFN). The MHA mechanism leverages multiple attention heads to capture diverse dependencies within the input sequence. For each head, attention is computed as follows:

$$\text{Attention}(\mathbf{Q}, \mathbf{K}, \mathbf{V}) = \mathsf{softmax}\left(\mathbf{Q}\mathbf{K}^T / \sqrt{d_k}\right)\mathbf{V},$$

where $\mathbf{Q}$, $\mathbf{K}$, and $\mathbf{V}$ are derived from learned linear projections of the input using $\mathbf{W}_\ell^Q$, $\mathbf{W}_\ell^K$, and $\mathbf{W}_\ell^V$, respectively. The outputs of all heads are concatenated and transformed by $\mathbf{W}_\ell^O$ to restore the original model dimension. To reduce key-value cache size, recent models divide query heads into groups, with each group sharing a single key and value head—a structure known as grouped-query or multi-query attention (Ainslie et al., 2023). On the other hand, sliding window attention (SWA) (Beltagy et al., 2020; Jiang et al., 2023; Gemma Team et al., 2025; Cohere et al., 2025; OpenAI et al., 2025) reduces the context length over which queries attend to key and value states. For the FFN component, recent models adopt a gating structure based on the SiGLU mechanism, which is a variant of SwiGLU (Shazeer, 2020; Chowdhery et al., 2023) but uses the SiLU activation instead of Swish. This FFN can be replaced with a Mixture-of-Experts layer (Shazeer et al., 2017; Fedus et al., 2022), where multiple FFNs (experts) are adaptively routed for each token.

**Mamba architecture.** The Mamba architecture (Gu & Dao, 2023) is a recent advancement in sequence modeling, building on structured state space models like S4 (Gu et al., 2021a), H3 (Fu et al., 2022), and S4D (Gu et al., 2022). Mamba replaces the attention mechanism with a state space model (SSM) layer, enabling efficient and expressive modeling of long-range dependencies:

$$\mathbf{h}_t = \bar{\mathbf{A}}\mathbf{h}_{t-1} + \bar{\mathbf{B}}\mathbf{x}_t, \quad \mathbf{y}_t = \mathbf{C}\mathbf{h}_t + \mathbf{D}\mathbf{x}_t$$

Here, $\mathbf{h}_t$ represents the latent state at time $t$, $\mathbf{x}_t$ is the input, and $\mathbf{y}_t$ is the output. In practice, $\bar{\mathbf{A}}$ is discretized using zero-order hold (ZOH), and $\bar{\mathbf{B}}$ is discretized via the Euler method. A key innovation is the input-dependent modulation of the state space parameters $\mathbf{B}$, $\mathbf{C}$, and the discretization step $\mathbf{\Delta}$, allowing the model to flexibly control context retention and input integration for each token. This selectivity enables Mamba to match the modeling quality of Transformers. Additionally, inspired by gated attention units (Hua et al., 2022), Mamba incorporates a SiLU-based gating mechanism. Mamba 2 (Dao & Gu, 2024) advances this design by introducing the state space duality (SSD) layer, which reformulates SSM computations as matrix multiplications highly optimized for modern hardware (e.g., GPUs, TPUs). This results in significantly faster training and improved practical efficiency. Throughout this paper, we primarily utilize the Mamba 2 architecture as the computational primitive, with each block always followed by a FFN layer, which can optionally be replaced with a MoE layer.

**Compute and memory costs comparison.** When analyzing costs in hybrid architectures (also for SWA models combining global and local attention), we reference the per-block costs in Table 1 (see Appendix C for details). For example, in a 1B model with 8K context, Transformer uses about 18% more FLOPs per sample than Mamba, due to its quadratic scaling versus Mamba's linear scaling. From a memory perspective, Mamba uses 2.5× more parameters per block (25M vs. 10M), but its cache size is 95% smaller than that of a Transformer (256 MiB vs. 13.4 MiB).

Table 1: **Overall comparison of per-block training compute and static (weight) and dynamic (KV Cache) memory costs during decode across different primitives.** For simplicity, we exclude the additional term of 6 times the block parameter count from FLOPs. Here, $N$ denotes number, $d$ is dimension, and $L$ is sequence length. $L_{\text{swa}}$ is the sum of sliding window size and attention sink sizes, and $d_{\text{ssm}}$ is typically $2d_{\text{model}}$. For Transformers, we assume grouped-query attention, and cache sizes are calculated with `bfloat16` precision. Refer to Appendix C for further details.

| Models | Compute | Memory | |
|---|---|---|---|
| | FLOPs per Sample | Parameter Counts | Cache Size |
| Llama | $12d_{\text{model}}L_{\text{ctx}} \times (L_{\text{ctx}}+1)/2$ | $2d_{\text{model}}d_{\text{model}}$ $+2d_{\text{model}}d_{\text{head}}N_{kv}$ | $2d_{\text{head}}N_{kv}L_{\text{ctx}}$ $+2d_{\text{head}}N_{kv}L_{\text{ctx}}$ |
| Sliding Window | $12d_{\text{model}}L_{\text{swa}}$ $\times((L_{\text{swa}}+1)/2+(L_{\text{ctx}}-L_{\text{swa}}))$ | $2d_{\text{model}}d_{\text{model}}$ $+2d_{\text{model}}d_{\text{head}}N_{kv}$ | $2d_{\text{head}}N_{kv}L_{\text{swa}}$ $+2d_{\text{head}}N_{kv}L_{\text{swa}}$ |
| Mamba | $3L_{\text{ctx}} \times (9d_{\text{ssm}}d_{\text{state}}+2d_{\text{ssm}})$ | $d_{\text{model}}(2d_{\text{ssm}}+2d_{\text{state}}+N_{\text{head}})$ $+d_{\text{state}}(N_{\text{conv}}+d_{\text{model}})+2N_{\text{head}}$ | $2d_{\text{ssm}}d_{\text{state}}$ $+2N_{\text{conv}}(2d_{\text{state}}+d_{\text{ssm}})$ |

# 3 Hybrid Architecture

Hybrid architectures can be categorized by their integration approach: inter-layer and intra-layer hybridization. We define each strategy and outline related research questions (RQs) below.

## 3.1 Inter-layer Hybrid Model

**Definition.** The inter-layer hybrid model alternates softmax attention layers with linear sequence modeling layers at specific intervals (see Figure 1a). A key design choice in this approach is determining the order and proportion in which Transformer and Mamba primitives are arranged and interpolated. Its practical effectiveness and straightforward implementation have made the inter-layer approach a dominant strategy in hybrid architecture development.

**Related works.** Most hybrid models combine Mamba (Gu & Dao, 2023; Dao & Gu, 2024) with various attention mechanisms to enhance quality and efficiency. Notable examples—H3 (Fu et al., 2022), MambaFormer (Park et al., 2024), Zamba (Glorioso et al., 2024b;a), Jamba (Lieber et al., 2024; Jamba Team et al., 2024), Samba (Ren et al., 2024; 2025), Hunyuan-TurboS (Hunyuan Team et al., 2025), Nemotron nano 2 (Basant et al., 2025), and IBM Granite 4.0—leverage Mamba's efficiency with global or local attention, scaling to over 1M context length. Some post-training approaches—such as MambaInLLaMA (Wang et al., 2024a), MOHAWK (Bick et al., 2024), Zebra-Llama (Yang et al., 2025), and Jet-Nemotron (Gu et al., 2025)—use knowledge distillation to convert parts of a pretrained Transformer into linear modules, while STAR (Thomas et al., 2024), Nemotron-Flash (Fu et al., 2025), and Composer (Acun et al., 2025) perform architecture search. Other hybrids—RecurrentGemma (Botev et al., 2024), Griffin (De et al., 2024), Titans (Behrouz et al., 2024), RWKV-X (Hou et al., 2025), MiniMax-01 (Li et al., 2025a), Qwen3-Next, and Kimi Linear (Kimi Team, 2025)—combine diverse self-attention variants (Beltagy et al., 2020; DeepSeek-AI, 2024; Qin et al., 2024; Lu et al., 2025; Qiu et al., 2025) with different linear modules (Qin et al., 2022; De et al., 2024; Yang et al., 2024; Peng et al., 2025).

**RQ1: Block ratios in inter-layer hybrid models.** One of the key research questions in designing inter-layer hybrids is determining the optimal ratio of Transformer blocks to Mamba blocks when stacking layers. This ratio determines the degree of interpolation between two computational primitives, impacting not only model quality but also efficiency metrics like throughput and cache size. Prior work (Jamba Team et al., 2024; Lieber et al., 2024; Basant et al., 2025) has generally favored configurations with a high proportion of Mamba layers—for example, a 1:7 ratio. However, existing literature, including Jamba (Lieber et al., 2024; Jamba Team et al., 2024), RWKV-X (Hou et al., 2025), Kimi-Linear (Kimi Team, 2025), and Waleffe et al. (2024), has mostly restricted the analysis to a limited set of block ratios. We revisit this design choice and offer deeper insights by comparing various ratios from both quality and efficiency perspectives.

**RQ2: Positioning of computational primitives.** Another open question is whether the position of interleaved blocks affects model quality, and how best to arrange them for optimal performance. The optimal positioning remains unclear, as it may depend on factors such as block ratio and model scale. While prior work (Lieber et al., 2024; Ren et al., 2025; Basant et al., 2025) often places Transformer blocks in the middle layers, and Waleffe et al. (2024) and Samba (Ren et al., 2024) investigate the positioning of FFN or self-attention blocks in highly specific settings, comprehensive studies on this topic are still lacking. To fill this gap, we conduct extensive ablation experiments, varying the positions of Transformer blocks (e.g., early, middle, and late layers) across different ratios and model sizes.

## 3.2 Intra-layer Hybrid Model

**Definition.** Intra-layer hybrid models achieve fine-grained fusion within individual layers by blending softmax attention and linear attention in parallel. A common approach (Dong et al., 2024; Zuo et al., 2025; Xiao et al., 2025) is head-wise splitting, where some heads use Transformer attention while others use Mamba (see intra-hybrid block in Figure 1a). Here, we use intra-hybrid and Mamba blocks as primitive candidates, so that this includes an interleaving mechanism akin to inter-layer hybrid models.

**Related works.** Intra-layer hybrid models can be classified by how each token interacts with different modules. In the head-wise splitting approach (e.g., Hymba (Dong et al., 2024), Falcon-H1 (Zuo et al., 2025), WuNeng (Xiao et al., 2025), and Dragon), attention heads are divided into groups, with each group assigned to a distinct module. Alternatively, works like Liger (Lan et al., 2025) process each token through all modules in parallel and fuse the outputs. For sequence-wise splitting, different modules are applied to context tokens based on their positions when computing attention scores. This splitting can be determined by an absolute point in the sequence, as in TransMamba (Li et al., 2025b), or by relative position, as in LoLCATs (Zhang et al., 2024). While not hybrid, differential architectures such as Differential Transformer (Ye et al., 2024) and Differential Mamba (Schneider et al., 2025) also use head-wise splitting, but use the same primitive type and subtract their attention scores.

**RQ1: Architectural variants for intra-layer hybrid.** We adopt a head-wise splitting approach, assigning different primitives to two groups of attention heads. Specifically, query and key states in the Transformer branch are projected to reduced dimensions, while the value state is expanded back to the original size (Ye et al., 2024; Schneider et al., 2025). Similarly, for Mamba, the hidden dimension of SSMs is reduced based on the configuration. However, establishing the optimal fusion design within this framework is non-trivial, as experimental evidence regarding design variants remains limited. For instance, Hymba (Dong et al., 2024) shares results for only a single concatenation variant, and Falcon-H1 (Zuo et al., 2025) focuses its deep ablations primarily on the Mamba component itself rather than the hybridization strategy.

To fill this gap, we explore a comprehensive set of fusion designs, investigating: (1) *normalization strategies*, such as using group normalization (Wu & He, 2018) as in Hymba (Dong et al., 2024); (2) *scaling mechanisms*, including learnable scalars (Dong et al., 2024; Ye et al., 2024; Schneider et al., 2025) or gating parameters (Lan et al., 2025; Xiao et al., 2025); (3) *fusion operations*, ranging from addition (Dong et al., 2024; Lan et al., 2025; Xiao et al., 2025) and subtraction (Ye et al., 2024; Schneider et al., 2025) to concatenation (Xiao et al., 2025); and finally, (4) *output projection types* and (5) *value state sharing strategies*. We thoroughly evaluate these variants at both 350M and 1B scales. Refer to Appendix D for further details.

**RQ2: Dimension ratios in intra-hybrid block.** The ratio of parameter sizes between Transformer and Mamba modules within intra-hybrid blocks—controlled by their hidden dimension allocations—is one of key architectural considerations, as evidenced by ablation studies in both Hymba (Dong et al., 2024) and Falcon-H1 (Zuo et al., 2025). By observing how model quality changes as this ratio varies, we can infer the relative importance of each primitive. Furthermore, assuming expert parallelism (Rajbhandari et al., 2022) enables parallel execution of the modules, overall efficiency will be bounded by the slower Transformer component. Thus, we investigate how much we can reduce the dimension assigned to the Transformer, aiming to enhance efficiency without compromising overall quality.

**RQ3: Block ratios and positioning of primitives.** In our intra-layer hybridization strategy, each block is either an intra-hybrid block or a Mamba block. By varying the block ratio to control the interpolation

Table 2: **Hybrid models achieve better quality than baselines under equal data and compute constraints.** $N_{\text{hyb}}$ denotes the number of intra-hybrid block primitive. We calculate total training compute FLOPs and cache sizes for a sequence length of 8K tokens. We use an approximate block ratio of 1:5 for SWA and hybrid architectures, which mix two different types of block.

| Models | Base Config | | | | | Compute | | | NLL ($\downarrow$) | | Few-shot Accuracy ($\uparrow$) | | | | | |
|---|---|---|---|---|---|---|---|---|---|---|---|---|---|---|---|---|
| | N-emb | $N_{\text{attn}}$ | $N_{\text{swa}}$ | $N_{\text{ssm}}$ | $N_{\text{hyb}}$ | Tok | FLOPs | Cache | DCLM | PG19 | LD | HS | PQ | ARC | OB | Avg |
| Llama | 0.97B | 16 | - | - | - | 60B | 4.5 e20 | 256 | 2.750 | 2.875 | 56.1 | 55.2 | 72.8 | 43.2 | 32.7 | 52.0 |
| SWA | 0.97B | 3 | 13 | - | - | 60B | 3.8 e20 | 63 | 2.741 | 2.867 | 56.0 | 55.9 | 72.6 | 44.2 | 34.8 | 52.7 |
| Mamba | 0.99B | - | - | 13 | - | 60B | 3.7 e20 | 13 | 2.758 | 2.891 | 53.7 | 55.1 | **73.8** | 43.9 | 35.2 | 52.3 |
| Inter-H | 0.96B | 2 | - | 11 | - | 60B | 3.7 e20 | 43 | 2.735 | 2.861 | 56.4 | 55.4 | 73.1 | 44.8 | **36.6** | 53.3 |
| Intra-H | 0.98B | - | - | 11 | 2 | 60B | 3.7 e20 | 38 | **2.728** | **2.853** | **58.7** | **57.4** | 73.3 | **47.9** | 36.4 | **54.7** |
| SWA | 0.97B | 3 | 13 | - | - | 71B | 4.5 e20 | 63 | 2.724 | 2.845 | 57.0 | 56.9 | 73.1 | 46.0 | 36.0 | 53.8 |
| Mamba | 0.97B | - | - | 13 | - | 73B | 4.5 e20 | 13 | 2.736 | 2.865 | 55.2 | 57.1 | 73.8 | 45.3 | 36.0 | 53.5 |
| Inter-H | 0.96B | 2 | - | 11 | - | 73B | 4.5 e20 | 43 | 2.716 | 2.842 | 56.7 | 57.6 | 73.3 | 46.5 | 35.8 | 54.0 |
| Intra-H | 0.98B | - | - | 11 | 2 | 72B | 4.5 e20 | 38 | **2.709** | **2.831** | 58.4 | 57.7 | 74.4 | 46.9 | 37.0 | **54.9** |

degree, we analyze how different configurations affect both quality and efficiency, aiming to understand the trade-offs in intra-hybridization. We further examine how the placement of these intra-hybrid blocks at different depths influences model quality. Since both Transformer and Mamba are included, unlike using homogeneous primitives, we explore how its behavior differs from that of inter-layer hybrid models.

# 4 Experiments

We build hybrid models with computational primitives following the configurations of Llama 3.2 (Llama Team, 2024) and Mamba 2 (Dao & Gu, 2024) architectures. We use the TorchTitan framework (Liang et al., 2024) for large-scale LLM training with H200 GPUs. Refer to Appendix E for further details.

## 4.1 Main Results on Quality

**Hybrid architectures significantly outperform homogeneous models.** Table 2 compares our two hybridization strategies—optimized along positioning and design choice axes (see §4.5 and §4.6)—with conventional baselines (Llama Team, 2024; Dao & Gu, 2024; Gemma Team et al., 2025). Our controlled experiments show that hybrid models, including SWA, consistently outperform homogeneous models in negative log-likelihood (NLL) and few-shot accuracy under the same data budget. This demonstrates that effectively combining complementary inductive biases from different primitives is key to improving model quality through hybridization. Notably, we provide a meaningful comparison between inter- and intra-hybrid architectures under matched budgets, which has not been previously explored or shared (Ren et al., 2024; Jamba Team et al., 2024; Dong et al., 2024; Basant et al., 2025). Under the same FLOP budget, hybrids achieve even more substantial quality gains (e.g., 2.9% increase in accuracy and 0.04 reduction in NLL). We find that these advantages are consistent across different block ratios and model scales, demonstrating the robustness of the hybrid approach (see Figure 1b). Refer to Appendix F for further details and Appendix H for longer context pretraining results.

**Hybrid architectures achieve superior convergence with comparable stability.** The left panel of Table 3 illustrates the learning curves used to analyze the training dynamics, specifically focusing on convergence speed and stability. While Mamba initially converges slower than the Transformer, it eventually matches Llama's loss after full pretraining (60 billion tokens). In contrast, hybrid models track Llama's trajectory early on but achieve lower loss from the mid-training stage, suggesting that combining these primitives unlocks superior optimization dynamics. Regarding stability, we observe no significant deviations across architectures; all models exhibit consistent fluctuation patterns, with minor differences in Llama attributed to its distinct data parallelism configuration (DP = 16 vs. 8).

**Hybrid architectures ensure robust downstream performance.** As shown in the right panel of Table 3, fine-tuning results reveal that while SWA and Mamba exhibit competitive language modeling

Table 3: **(Left) Hybrid architectures demonstrate stable training dynamics.** The models are pretrained on 60B tokens with checkpoints logged every 100 steps. For the hybrid models, we adopt a block ratio of 1:5. Llama model is trained with $DP = 16$, while all the other models use $DP = 8$. **(Right) Hybrids outperform baselines on downstream tasks, like summarization and question-answering (QA).** We fine-tune five distinct architectures, all pre-trained with a compute budget of 4.5e20 FLOPs. Our evaluation benchmarks include GovReport (GR; Huang et al. (2021)), Task $\mathbb{A}$, and Task $\mathbb{B}$ for summarization, alongside SQuAD (SQA; Rajpurkar et al. (2016)), TriviaQA (TQA; Joshi et al. (2017)), Task $\mathbb{C}$ for QA. For all anonymized tasks, the reported scores denote the relative performance against the Transformer baseline.

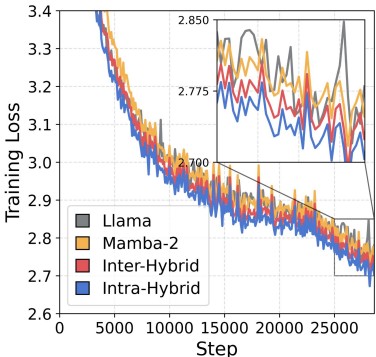

| Models | Base Config | | | | | Sum Tasks | | | QA Tasks | | |
|---|---|---|---|---|---|---|---|---|---|---|---|
| | N-emb | $N_{attn}$ | $N_{swa}$ | $N_{ssm}$ | $N_{hyb}$ | GR | Task $\mathbb{A}$ | Task $\mathbb{B}$ | SQA | TQA | Task $\mathbb{C}$ |
| Llama | 0.97B | 16 | - | - | - | 20.55 | 0.00 | 0.00 | 54.72 | 51.89 | 0.00 |
| SWA | 0.97B | 3 | 13 | - | - | 18.91 | −2.79 | −2.41 | 48.55 | **51.99** | +0.17 |
| Mamba | 0.99B | - | - | 13 | - | 13.08 | −2.26 | −19.49 | **55.55** | 36.50 | −0.71 |
| Inter-H | 0.96B | 2 | - | 11 | - | 19.56 | **+0.34** | **+0.61** | 54.91 | 51.66 | +0.04 |
| Intra-H | 0.98B | - | - | 11 | 2 | **20.76** | −0.33 | −0.35 | 55.42 | 51.00 | **+1.13** |

performance, this capability does not fully transfer to downstream tasks. Specifically, limited by strong locality biases (Beltagy et al., 2020; Ben-Kish et al., 2024; Ali et al., 2025; Schneider et al., 2025), these models suffer significant degradation in retrieval-intensive summarization tasks. Although they remain competitive on a few QA benchmarks, their overall performance is inconsistent. In contrast, hybrid architectures demonstrate remarkable robustness regardless of the integration strategy (inter- or intra-hybrid), consistently matching or outperforming the standard Transformer. This confirms that hybridization effectively yields superior model quality while maintaining efficiency, thereby resolving the limitations of homogeneous Mamba models. Refer to Appendix I for further details.

## 4.2 Main Results on Efficiency

**FLOPs reduction in hybrid models translate into faster actual end-to-end training time.** Mamba's linear complexity leads to lower FLOPs than Transformer (18% lower for 1B scale at 8K lengths; see Table 1 for details). As shown in Figure 2, hybrid models leverage these advantage for superior performance in compute-bound scenarios. Empirically, lower FLOPs almost directly yield faster training, aided by parallel scan algorithms (Blelloch, 1990; Smith et al., 2022). For intra-hybrid models, the expert parallelism (Rajbhandari et al., 2022) can theoretically optimize training speed further. Although Mamba and hybrid models may use slightly more memory during training, adjusting the block ratio provides flexibility to balance between efficiency and model quality.

**Hybrid models achieve a superior Pareto frontier of inference throughput and quality.** Figures 3a and 3b demonstrate that hybrid architectures, by leveraging Mamba's linear complexity and constant cache size, sustain high throughput and sub-quadratic memory growth across context lengths. This enables faster generation compared to baselines, while still maintaining high output quality (see Figure 1b). Although SWA uses a $512 + 64$ (window + sink) attention map, it remains slower than Mamba's linear attention, making hybrids overall faster. Interestingly, intra-layer hybrids, which utilize half-sized Transformer, further outperform inter-layer hybrids even when executed sequentially. Refer to Appendix J for further details.

## 4.3 Long-context Capability Evaluation

**Position-wise loss on PG19 shows hybrid-Mamba models extrapolates well to longer contexts.** Figure 3c compares position-wise loss on the PG19 corpus (Rae et al., 2019a) to examine the long-context capabilities of various architectures. All models show decreasing NLL up to the 8K-token pretraining

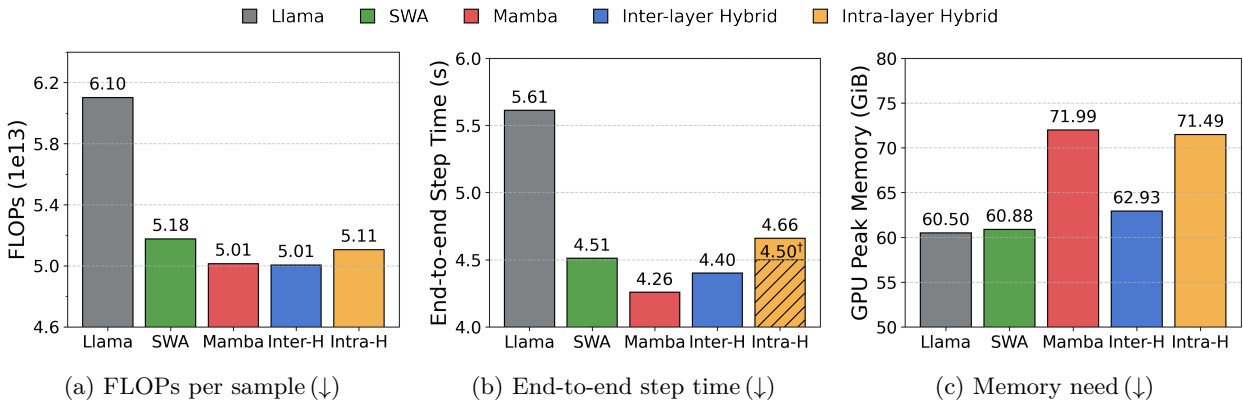

(a) FLOPs per sample (↓)  (b) End-to-end step time (↓)  (c) Memory need (↓)

Figure 2: **Hybrid models have lower FLOPs, which directly leads to reduced actual training time.** We measure metrics for 1B model using 8 H200 GPUs with FSDP, `torch.compile`, 8K lengths, and a local batch size of 4, without activation checkpointing. Both SWA and hybrid models use a 1:5 block ratio. † indicates a theoretical time achievable with parallelism (Rajbhandari et al., 2022). For memory usage, we report the peak memory allocated on a single H200 GPU.

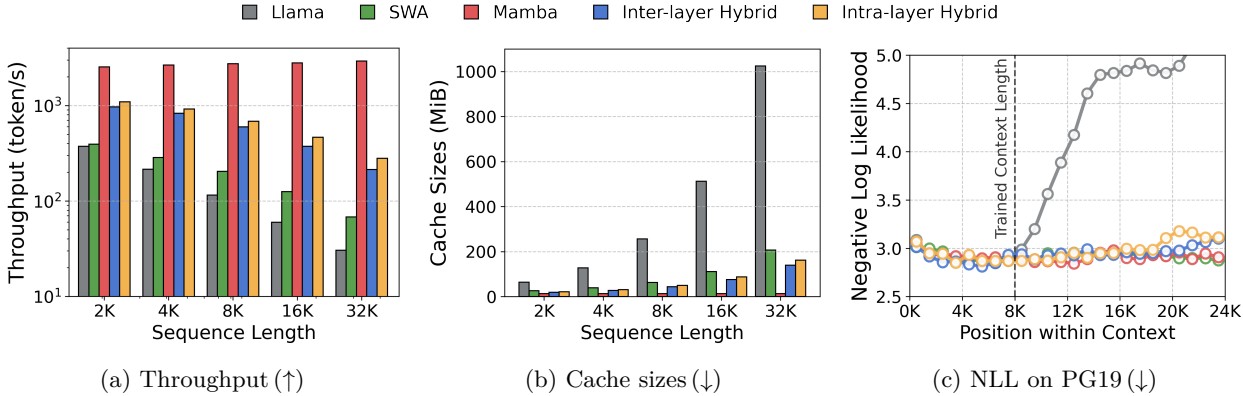

(a) Throughput (↑)  (b) Cache sizes (↓)  (c) NLL on PG19 (↓)

Figure 3: **(a, b) Hybrid architectures show sub-quadratic scaling of inference throughput and memory as sequence increases.** SWA and hybrid 1B models use block ratio of 1:5. For throughput, we set the prompt length to 512 and the batch size to 4. **(c) Mamba enables length generalization in terms of perplexity, allowing hybrids to maintain strong performance.** We use 1B models trained with a compute budget of 4.5e20 and 8K lengths. Loss is averaged every 1K positions over 30 samples.

context, as later tokens benefit from more context. Beyond 8K, SSMs can effectively extrapolate (Gu & Dao, 2023; Dao & Gu, 2024; Yang et al., 2024), while Transformer struggles due to its positional encoding methods (Press et al., 2021; Kazemnejad et al., 2023; Zhou et al., 2024). Therefore, hybrid architectures with a 1:5 block ratio leverage Mamba to demonstrate strong length extrapolation without being excessively constrained by the Transformer component. Furthermore, given findings that explicit positional information is redundant in hybrids (Park et al., 2024; Lieber et al., 2024; Waleffe et al., 2024), adopting a NoPE (No Positional Encoding) structure (Kazemnejad et al., 2023) allows us to further resolve the length generalization limitations stemming from the Transformer.

**Hybrid models overcome weakness of Transformer and Mamba in in-context retrieval.** Figure 4 presents in-context retrieval performance on the Needle-In-A-Haystack benchmark (Kamradt, 2023; Kuratov et al., 2024). A needle is inserted at a specific depth in the context, and models are evaluated on their ability to retrieve (generate) it. Transformer accuracy drops to near zero beyond 8K context length, as expected. On the other hand, SWA and Mamba also struggle with retrieval outside their local window and sink token regions, or in long-range tokens, despite strong perplexity scores in Figure 3c; this is likely due

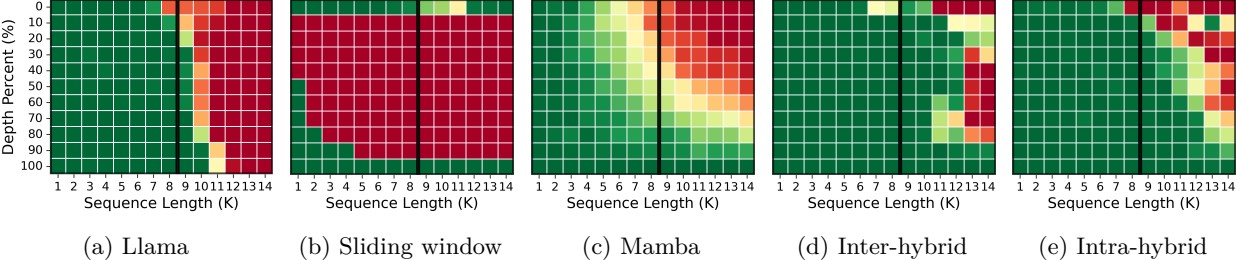

|              | (a) Llama | (b) Sliding window | (c) Mamba | (d) Inter-hybrid | (e) Intra-hybrid |

Figure 4: **Hybrid models overcome the limitations of both foundational primitives, achieving superior in-context retrieval performance.** We insert a needle (random 7-digit number associated with random city name (Gemini Team et al., 2024)) across the 0–100% depth range (y-axis) for context lengths up to 14K (x-axis). Over 100 trials, green indicates 100% accuracy, while red denotes 0% accuracy. We use 1B model checkpoints trained with 8K length and a FLOPs budget of 4.5e20. SWA and hybrid models use 1:5 block ratio.

Table 4: **(Left)** **All architectures consistently achieve significant quality gains from MoE integration.** All models are trained on 60B tokens, with hybrid models using a 1:5 block ratio. We use one shared expert and the selected top-1 expert among eight. Additionally, a token-choice router with loss-free balancing algorithm is utilized (Wang et al., 2024b; DeepSeek-AI, 2024). **(Right) Hybrid architectures demonstrates a compute-optimal scaling line that lies between those of Transformer and Mamba.** We train models at four different scales—100M, 350M, 1B, and 3B—across five compute budgets. Each marker indicates the optimal model size for a given compute budget. For hybrid models, we use a 1:5 block ratio.

| Models | Base Config | | | | | | MoE | | Performance ($\downarrow$ / $\downarrow$ / $\uparrow$) | | |
|---|---|---|---|---|---|---|---|---|---|---|---|
| | Act | Total | $N_{attn}$ | $N_{ssm}$ | $N_{hyb}$ | FLOPs | Num | Act | DCLM | PG19 | Acc |
| Llama | 0.97B | 0.97B | 16 | - | - | 4.5 e20 | - | - | 2.750 | 2.875 | 52.0 |
| | 0.97B | 3.79B | 16 | - | - | 4.5 e20 | 1+8 | 1+1 | **2.656** | **2.775** | **55.8** |
| Mamba | 0.99B | 0.99B | - | 13 | - | 3.7s e20 | - | - | 2.758 | 2.891 | 52.3 |
| | 0.99B | 3.28B | - | 13 | - | 3.7 e20 | 1+8 | 1+1 | **2.673** | **2.803** | **55.0** |
| Inter-H | 0.96B | 0.96B | 2 | 11 | - | 3.7 e20 | - | - | 2.735 | 2.861 | 53.3 |
| | 0.96B | 3.25B | 2 | 11 | - | 3.7 e20 | 1+8 | 1+1 | **2.653** | **2.780** | **56.0** |
| Intra-H | 0.98B | 0.98B | - | 11 | 2 | 3.7 e20 | - | - | 2.728 | 2.853 | 54.7 |
| | 0.98B | 3.27B | - | 11 | 2 | 3.7 e20 | 1+8 | 1+1 | **2.648** | **2.775** | **56.9** |

to their focus on local information (Ben-Kish et al., 2024). In contrast, both inter- and intra-hybrid models surprisingly maintain strong retrieval performance up to about 1.5x the pretraining length, overcoming the limitations of the base primitives rather than simply inheriting them. While retrieval accuracy in the middle of extrapolated contexts does decline (Liu et al., 2023), hybrid models consistently demonstrate improved retrieval capabilities. Refer to Appendix K for further details.

## 4.4 Scaling Analysis

**Mixture-of-Experts are fully compatible with hybrid architectures.** A few recent inter-layer hybrid models (Lieber et al., 2024; Jamba Team et al., 2024) have incorporated Mixture-of-Experts (MoE) (Shazeer et al., 2017; Fedus et al., 2022) to improve performance at fixed compute. In Table 4, we re-examine MoE in inter-layer hybrids and also evaluate intra-layer hybridization, comparing both to baseline models. Across all architectures, MoE yields a substantial reduction in NLL (by 0.08) and 4 point increase in few-shot accuracy. Since hybridization is applied to the attention component, integrating MoE into the FFN layer remains compatible with all hybrid models. The data-hungry nature of MoE also enables more efficient scaling of hybrid models for a fixed number of activated parameters. Refer to Appendix L for further details.

Table 5: **(Left) Inter-layer hybrid achieves the best quality at a 1:1 block ratio, but the 1:5 ratio is preferable to balance efficiency and quality.** Transformer blocks are evenly distributed in the middle. All models are trained on 60B tokens. **(Right) Interleaving Transformer blocks at intermediate depths is key for optimal performance.** The upper figure shows results of ablating the position of a single Transformer block in 1B (13 layers) and 350M (11 layers) models with a 1:12 ratio. In the lower figure, after distributing Transformer blocks in the middle of 1B model, we move the first block to the first layer (Front) or the last block to the last layer (End). All models are trained on 60B tokens.

| | | Base Config | | | Performance ($\downarrow$ / $\downarrow$ / $\uparrow$) | | |
|---|---|---|---|---|---|---|---|
| Sizes | Ratio | N-emb | $N_{\text{attn}}$ | $N_{\text{ssm}}$ | DCLM | PG19 | Acc |
| 1B | 1:0 | 0.97B | 16 | - | 2.750 | 2.875 | 52.0 |
| | 0:1 | 0.99B | - | 13 | 2.758 | 2.891 | 52.3 |
| | 1:1 | 0.96B | 7 | 7 | **2.725** | **2.847** | **54.0** |
| | 1:3 | 0.94B | 3 | 10 | 2.733 | 2.859 | 54.0 |
| | 1:5 | 0.96B | 2 | 11 | 2.735 | 2.861 | 53.3 |
| | 1:12 | 0.97B | 1 | 12 | 2.741 | 2.866 | 53.1 |
| 350M | 1:0 | 0.35B | 14 | - | 2.882 | 3.015 | 48.7 |
| | 0:1 | 0.35B | - | 11 | 2.880 | 3.024 | 48.7 |
| | 1:1 | 0.35B | 6 | 6 | **2.850** | **2.985** | 49.3 |
| | 1:3 | 0.34B | 3 | 8 | 2.858 | 2.994 | **50.2** |
| | 1:5 | 0.35B | 2 | 9 | 2.860 | 2.999 | 49.4 |
| | 1:12 | 0.36B | 1 | 10 | 2.864 | 3.003 | 49.3 |

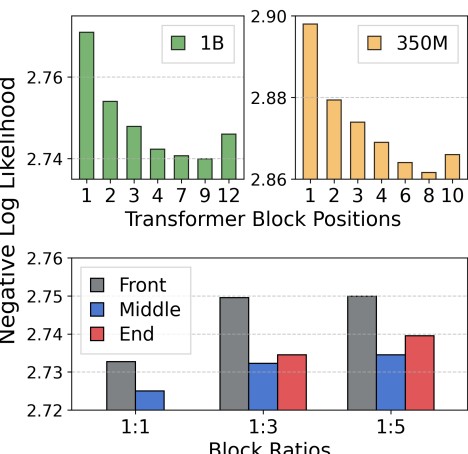

**Hybrid models are efficient and scalable architectures.** The right figure of Table 4 analyzes the scalability of different architectures and identifies compute-optimal scaling strategies (Kaplan et al., 2020; Hoffmann et al., 2022). The compute-optimal line illustrates the best achievable quality and parameter sizes for each architecture under a given compute budget. Mamba performs best with larger models and less data feeding, while Transformers favor a higher token-to-parameter ratio of around 20 (Hoffmann et al., 2022). Hybrid models show intermediate scaling behavior, with intra-layer hybrids being a little slightly more data-hungry. These results provide clear guidance on how to optimally scale up hybrid models. Refer to Appendix M for further details.

### 4.5 Ablation Studies for Inter-layer Hybridization

**To ensure quality, aim for a high 1:1 ratio, but to balance with efficiency, use about 1:5 ratio.** In Table 5, we compare inter-hybrid models with six block ratios at two scales (1:0 = Transformer, 0:1 = Mamba). Hybrid architectures consistently outperform homogeneous ones, with the balanced 1:1 ratio yielding the best quality. However, due to the Transformer's quadratic complexity, inference throughput gains become limited. To balance both efficiency and quality, a ratio of around 1:5 appears to be optimal, which aligns with the lower ratios (e.g., 1:5, 1:7) adopted by large hybrid language models (Lieber et al., 2024; Wang et al., 2025; Basant et al., 2025). Refer to Appendix N.1 for further details.

**Never place Transformer blocks at the front.** Deciding the order of Transformer and Mamba blocks is a key design choice. Our ablation study (see figures next to Table 5) shows that, for a 1:12 ratio, placing the Transformer block in the early layers leads to significant performance drop, while positioning it in the middle yields the best results. Similar experiments with higher block ratios (bottom-right figure) also confirm that putting Transformer blocks at the front consistently leads to worse performance than homogeneous models, regardless of ratio. These finding support prior work favoring middle placement of Transformer (Jamba Team et al., 2024; Dong et al., 2024). Refer to Appendix N.2 for further details.

To investigate this degradation, we analyze the attention score distributions (detailed in Appendix O). Our visualizations reveal that Transformer blocks placed in the early layers exhibit a highly uniform attention distribution across all tokens, functioning essentially as a global context encoder. We hypothesize that this global attending behavior severely clashes with Mamba's local inductive bias. This conflict ultimately disrupts the network's representational flow. In the future, more efficient methods like layer-wise sensitivity analysis (Yang et al., 2025) or architecture search (Thomas et al., 2024) may help optimize block positions.

Table 6: **(Left) We identify a more optimal architecture than previous designs through ablation studies on intra-hybrid block.** We train a 1B model on 60B tokens, replacing all layers with intra-hybrid blocks (i.e., 1:0 block ratio). All heads are split into two, with each linked to a half-sized primitive. † denotes our re-implementation of the two prior works, equipped with value dimension expansion for GQA. **(Right) Keeping larger Transformer dimension within intra-hybrid block improves quality (despite reduced efficiency), and placing the intra-hybrid block in the middle yields the best results.** We train the 1B models using the concatenation variant (third row from the bottom) for the dimension ratio ablation, and our final architecture (bottom row) for the block ratio ablation. In the figure above, we use a block ratio of 1:0, where all layers become intra-hybrid blocks. For lower figure, we place intra-hybrid blocks in the middle by: placing consecutively (Cluster), distributing evenly (Scatter), or placing at the beginning and end as well (Sandwich).

| Models | Design Choices | | | | | Performance ($\downarrow$ / $\downarrow$ / $\uparrow$) | | |
| | Module | Norm | Scalar | Fusion | Out | DCLM | PG19 | Acc |
|---|---|---|---|---|---|---|---|---|
| Llama | T | - | - | - | 1 | 2.750 | 2.875 | 52.0 |
| Mamba | M | - | - | - | 1 | 2.758 | 2.891 | 52.3 |
| Diff-T | T / T | - | Diff-T | Diff | 1 | 2.727 | 2.848 | 53.6 |
| Diff-M | M / M | - | Diff-M | Diff | 2 | 2.759 | 2.892 | 53.0 |
| Hymba† | T / M | Group | Scale | Add | 1 | 2.726 | 2.846 | 52.4 |
| Falcon-H1† | T / M | - | - | Conc | 1 | 2.718 | 2.840 | 53.6 |
| Variants | T / M | - | Scale | Add | 1 | 2.740 | 2.865 | 53.9 |
| | T / M | Group | - | Add | 1 | 2.721 | 2.840 | 54.5 |
| | T / M | Group | Gate | Add | 1 | 2.751 | 2.876 | 53.1 |
| | T / M | Group | - | Diff | 1 | 2.721 | 2.842 | 54.0 |
| | T / M | Group | - | Conc | 1 | 2.715 | 2.833 | 54.8 |
| | T / M | Group | - | Add | 2 | 2.714 | 2.834 | 54.5 |
| | T / M | Group | - | Diff | 2 | **2.712** | **2.831** | **54.9** |

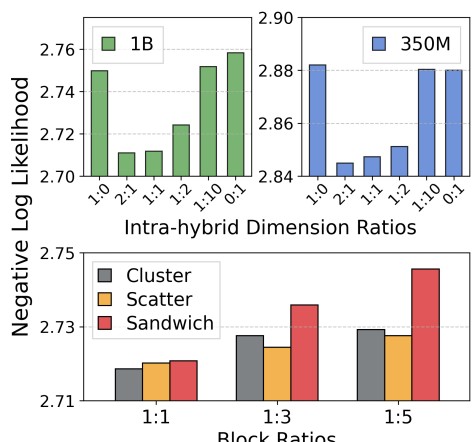

## 4.6 Ablation Studies for Intra-layer Hybridization

**Novel outperforming architectural variant for intra-hybrid block.** In designing intra-hybrid blocks, we explore four axes: normalization layer, learnable scalar, fusion operation, and output projection. In Table 6, our experiments reveal that normalization is crucial due to the scale differences between modules (Dong et al., 2024), which makes additional scaling factors unnecessary. For output fusion, either subtracting the outputs to mitigate attention noise or simply concatenating them yield the best quality. The resulting architecture surpasses prior intra-hybrid models (e.g., Hymba (Dong et al., 2024), Falcon-H1 (Zuo et al., 2025)), and Transformer-Mamba hybrids outperform homogeneous differential architectures (e.g., Differential Transformer (Ye et al., 2024), Differential Mamba (Schneider et al., 2025)). Refer to Appendix P.1 for further details.

**Enlarging Transformer dimension improves quality, despite reduced efficiency.** The upper-right figure presents ablation results for dimension allocation ratios within the intra-hybrid block, defined by the proportion of query / key dimension in Transformer versus pre-expansion dimension in Mamba (with 1:0 and 0:1 representing pure Transformer and Mamba, respectively). The results indicate that a balanced allocation is important for quality, with larger Transformer dimensions leading to greater performance gains (similar observations in Zuo et al. (2025))—suggesting that the Transformer component even plays a more critical role than Mamba. However, since throughput is limited by the Transformer under parallel execution, 1:1 dimension ratio offers a practical and effective balance. Refer to Appendix P.2 for further details.

**Evenly scattering intra-hybrid blocks across depths yields the best quality.** We further investigate block ratio and ordering strategies for intra-hybrid models (see lower figure next to Table 6). Increasing the proportion of Transformer-containing intra-hybrid blocks consistently improves quality, in line with previous findings on primitive importance from dimension ratio ablations. However, using more Mamba blocks remains a practical choice for efficiency. For block positioning, motivated by lessons from §4.5, we keep intra-hybrid blocks in the middle positions for ablation studies. Notably, evenly distributing these intra-hybrid blocks across depths yields the best results, while placing them at the ends (Sandwich strategy) leads to huge performance drops. As shown in the Appendix O, we hypothesize that the uniform scoring of Transformer blocks in the early stages causes this performance degradation.

**Learned routing validates the synergistic specialization of hybrid components.** To investigate the optimal composition of intra-hybrid blocks akin to neural architecture search (Thomas et al., 2024; Acun et al., 2025; Fu et al., 2025), we train a model using a learnable interpolation weight ($\alpha$) that dynamically balances the contributions of the Transformer and Mamba modules inside an intra-hybrid block (see Appendix Q). The resulting $\alpha$ distribution is strongly aligned with our manual block positioning findings, automatically favoring Mamba in early layers and the Transformer in the middle. Furthermore, token-level analysis reveals a natural ensemble effect: the Transformer strategically activates at critical syntactic boundaries to guide global reasoning, whereas Mamba efficiently handles highly localized sub-word dependencies.

## 5 Distinctions from Prior Works

While numerous hybrid architectures have been proposed, the rationale behind their specific design choices remains underexplored. Existing research has predominantly focused on scaling foundational hybrid models (Glorioso et al., 2024b; Lieber et al., 2024; Ren et al., 2024; Dong et al., 2024). Although these efforts successfully demonstrate reasoning and long-context performance by even leveraging extensive post-training, they often present final architectures without fully disclosing the *recipes* or systematic design processes that justified those decisions. Most works rely on verbal justifications or selective ablations; for instance, investigations into block ratios for inter-layer hybrids are sparse (Lieber et al., 2024; Hou et al., 2025; Kimi Team, 2025), and experimental evidence regarding design variants in intra-layer hybrids is not exhaustively covered (Dong et al., 2024; Zuo et al., 2025). Moreover, a rigorous comparison of model quality and efficiency between inter-layer and intra-layer approaches—two streams traditionally researched in parallel—has been absent.

We bridge this gap by conducting a comprehensive study that integrates these previously isolated domains. Through systematic ablation studies, we provide architectural insights and determine optimal design choices for each hybridization strategy. Beyond individual designs, we present the first in-depth comparison of the trade-offs between inter-layer and intra-layer hybrids to pinpoint the most robust configurations.

## 6 Conclusion and Future Work

In this work, we present a holistic analysis for inter-layer and intra-layer hybrid architectures. Our comprehensive evaluation demonstrates that hybrid models consistently outperform homogeneous architectures across multiple quality metrics, with intra-layer hybrids showing the strongest results. Notably, these hybrid approaches also deliver superior efficiency, achieving faster training and inference, much more than widely adopted sliding window attention models. Our findings shed new insights and practical guidance for designing high-quality, efficient hybrid architectures.

**Validation at scale.** Our study is limited to 3B models, pretrained on 60B tokens. Since standalone Mamba models often show diminishing returns at scale (Waleffe et al., 2024), a crucial next step is to validate whether the performance advantages of our hybrid model persist with longer training, at larger scales, and across a more comprehensive suite of evaluation benchmarks. However, we are optimistic they will, based on our scaling analysis up to the 3B parameter size and prior work suggesting that hybrid models scale more effectively (Waleffe et al., 2024; Jamba Team et al., 2024; Basant et al., 2025).

**Compatibility of advanced primitives.** Our work combined base Transformer and Mamba blocks, whereas recent models incorporate more advanced variants. For instance, some models (De et al. (2024), Ren et al. (2025), Qwen3-Next) employ self-attention variants like local (Beltagy et al., 2020), differential (Ye et al., 2024), and gated attention (Qiu et al., 2025), while others (RWKV-X (Hou et al., 2025), Qwen3-Next, Kimi Linear (Kimi Team, 2025)) use linear attention mechanisms such as RWKV-7 (Peng et al., 2025), Gated DeltaNet (GDN) (Yang et al., 2024), Kimi Delta Attention (Kimi Team, 2025), or Mamba 3 (Lahoti et al., 2026). A key question is whether our design insights still apply to these newer hybrids and if the advantages of each component are preserved when combined. While GDN has recently gained significant traction, it is essentially a delta over the Mamba 2 architecture. Furthermore, recent findings indicate that Mamba 2 actually outperforms GDN when its $A$ and dt_bias parameters are randomly and correctly initialized—a rigorous initialization protocol that we strictly follow in our experiments. Thus, we strongly anticipate that our core structural findings and hybridization principles will broadly generalize to them.

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

# Contents

## A The Use of Large Language Models

We used large language models based on Llama 3.2 (Llama Team, 2024) and Llama 4 to polish the overall writing after drafting the paper ourselves. Additionally, we utilized the same models for vibe coding when fixing bugs or when making the initial drafts of the figures.

## B Broader Impact Statement

This work systematically analyzes and optimizes hybrid Transformer-Mamba architectures, demonstrating substantial gains in computational efficiency and long-context modeling. By reducing training compute, memory requirements, and inference latency, our findings promote more accessible and environmentally sustainable AI development. Conversely, these same efficiency gains could be exploited by malicious actors to generate disinformation at scale. We therefore emphasize that future applications of these architectures must be coupled with rigorous safety alignment protocols.

## C Details for Computational and Memory Costs Comparison

**FLOPs per sample.** Most parameters are linear weight matrices, so total FLOPs can be estimated by multiplying the parameter count by $6L_{\mathrm{ctx}}$ (accounting for addition, multiplication, and forward / backward passes, with the backward pass being roughly twice as expensive as the forward pass). For Transformer, attention further adds FLOPs from query-key dot products and value scaling, totaling $12N_L d_{\mathrm{model}} L_{\mathrm{ctx}}(L_{\mathrm{ctx}}+1)/2$, considering only causal attention and excluding Flash-Attention overhead (Dao et al., 2022). In SWA, FLOPs depend on the number of activated tokens based on window and sink sizes. Mamba, on the other hand, introduces additional FLOPs from its work-efficient parallel scan algorithm (Blelloch, 1990; Smith et al., 2022), calculated as $3L_{\mathrm{ctx}}(9d_{\mathrm{ssm}}d_{\mathrm{state}} + 2d_{ssm})$. Thus, FLOPs per sample scale quadratically with sequence length ($L_{\mathrm{ctx}}$) for Transformer, while Mamba scales linearly. For a 1B model with 8K context, Mamba uses about 18% fewer FLOPs per sample than a Transformer.

**Parameter counts.** The attention in Transformer block uses four projection weights—query, key, value, and output. Variants like GQA reduce parameter count by decreasing the number of key and value heads, shrinking the corresponding projection matrices. In contrast, Mamba featurizes the input for state space parameters using several projection weights and 1D convolution layers, typically projecting the hidden dimension to twice its original size (i.e., $d_{\mathrm{ssm}} = d_{\mathrm{model}}$). It also includes time-invariant parameters **A** and **D**, as well as gating and output projection weights. As a result, when comparing only the attention components (excluding FFN), Mamba's parameter count is approximately 2.5 times higher than that of a Transformer block in a 1B model (e.g., 25M vs. 10M parameters per block).

**Cache size.** Cache size is a major inference bottleneck due to the GPU memory hierarchy, increasing HBM–SRAM communication (Dao et al., 2022). In Transformer, the cache consists of key and value states. The cache size is $4N_L d_{\mathrm{head}} N_{\mathrm{kv}} L_{\mathrm{ctx}}$, where 4 accounts for key, value, and 2 bytes per `bfloat16` element. Meanwhile, Mamba maintains two types of caches: hidden states for convolution layer and memory states compressed into a finite space. Its total cache size is $2N_L(d_{\mathrm{ssm}}d_{\mathrm{state}} + N_{\mathrm{conv}}(2d_{\mathrm{state}} + d_{\mathrm{ssm}}))$, with `bfloat16` precision. Notably, Mamba's cache size is independent of sequence length, so its efficiency benefits increase with longer contexts. For a 1B model with 8K context, Mamba's cache footprint is about 95% smaller than a Transformer's (e.g., 256 MiB vs. 13.4 MiB).

# D  Details for Intra-layer Hybrid Architectures

Since intra-layer hybrid architectures integrate two distinct modules within a single layer, they offer a vast design space for optimization. We explore this flexibility through five primary axes of variation: output projection, scaling factors, normalization, fusion operation, and shared input values. These design choices can be formulated as the following generalized equation:

$$f(\mathbf{X}) = \mathbf{W}_{\text{out}} \left( \alpha_{ssm} \cdot \text{Norm} \left( \mathcal{M}_{\text{ssm}}(\mathbf{X}) \right) \oplus \alpha_{attn} \cdot \text{Norm} \left( \mathcal{M}_{\text{attn}}(\mathbf{X}) \right) \right),$$

$$\text{where} \quad \mathcal{M}_{\text{ssm}}(\mathbf{X}) = \mathbf{G} \odot \left( \mathbf{C} \prod \exp(\mathbf{A}\Delta)\mathbf{B}\Delta \right) \mathbf{W}^{\text{ssm}}\mathbf{X},$$

$$\mathcal{M}_{\text{attn}}(\mathbf{X}) = \text{softmax} \left( \mathbf{Q}\mathbf{K}^\top / \sqrt{d_k} \right) \mathbf{W}^V \mathbf{X}.$$

Here, $\mathcal{M}_{\text{attn}}$ and $\mathcal{M}_{\text{ssm}}$ denote the global attention and SSM modules, respectively. In the SSM formulation, $\mathbf{G}$ represents the gating branch, and the middle term denotes the discretized convolution kernel derived from the state-space parameters (indices are omitted for simplicity). $\mathbf{W}^{\text{ssm}}$ includes a projection layer and a 1D convolution layer. The colored terms highlight the five configurable axes. Table 7 summarizes the specific variants explored across each design axis. Our search space is comprehensive, encompassing strategies adopted in prior studies as well as previously unexplored configurations to ensure robust validation.

Table 7: **Overview of the architectural design space for intra-layer hybrid architectures.** We explore various design choices across five distinct axes.

| Design Axis | Explored Variants |
|---|---|
| **Normalization** (Norm) | (1) **Module-wise**: Applied individually to each module output. |
| | (2) **None**: Raw module outputs are used without normalization. |
| **Scaling Strategy** ($\alpha$) | (1) **Element-wise**: Per-module learnable vector ($\alpha \in \mathbb{R}^d$). |
| | (2) **Asymmetric**: Learnable scalar only for second module ($\alpha \in \mathbb{R}$). |
| | (3) **Identity**: No scaling applied ($\alpha = 1$). |
| **Fusion Operation** ($\oplus$) | (1) **Addition**: Standard summation ($\mathbf{y} = \mathcal{M}_S + \mathcal{M}_A$) |
| | (2) **Subtraction**: Differential interaction ($\mathbf{y} = \mathcal{M}_S - \mathcal{M}_A$) |
| | (3) **Concatenation**: Channel-wise concat ($\mathbf{y} = [\mathcal{M}_S; \mathcal{M}_A]$) |
| **Value Sharing** | (1) **None**: Independent projections from layer input $\mathbf{X}$. |
| | (2) **Value State**: Mamba reuses Attention's value ($\mathbf{V} = \mathbf{W}^V \mathbf{X}$). |
| **Output Projection** ($W_{\text{out}}$) | (1) **Unified**: Single projection shared after fusion. |
| | (2) **Separate**: Individual projections for each module before fusion. |

While most axes are self-explanatory, the *Value Sharing* strategy needs further elaboration, particularly regarding its inspiration from differential architectures (Ye et al., 2024). The Differential Transformer is designed to cancel noise in attention scores by subtracting the attention maps derived from two distinct head groups (i.e., $\text{softmax}(\mathbf{Q}_1\mathbf{K}_1^\top) - \text{softmax}(\mathbf{Q}_2\mathbf{K}_2^\top)$). Crucially, to preserve model expressivity, it applies the differential attention map to a Value state ($\mathbf{V} = \mathbf{W}^{\mathbf{V}}\mathbf{X}$) that retains the original full head dimension. Adopting this insight, we incorporate the value dimension expansion technique into our $\mathcal{M}_{\text{attn}}$ module.

Furthermore, we extend this logic to the hybrid domain: we hypothesize that sharing the Value states can induce a denoising or synergistic fusion effect between the global attention and Mamba's implicit attention mechanism. Consequently, we introduce a variant where the Mamba module utilizes the Attention's value projection directly (i.e., using $\mathbf{W}^V \mathbf{X}$ as input instead of a separate projection $\mathbf{W}^{\text{ssm}}\mathbf{X}$). Finally, unlike previous works, we rigorously benchmark our models against homogeneous differential architectures (Ye et al., 2024; Schneider et al., 2025) employing identical module configurations to validate the distinct advantages of hybridization.

## E  Detailed Experimental Setup

**Model architecture details.** Each model is constructed based on the Llama-based Transformer (Llama Team, 2024) and the Mamba architectures (Dao & Gu, 2024). We primarily follow to the configurations of Llama 3.2 and Mamba 2 across various model scales. Table 8 provides a summary of the detailed architectures, which serve as the foundational computational primitives for our hybrid architecture variants. The model combining sliding window attention and global self-attention also follows the standard Transformer configuration, utilizing a window size of 512 and an attention sink size of 64. For Rotary Positional Embeddings (RoPE), we adopt the frequency scaling strategy employed in Llama 3 (Llama Team, 2024). Specifically, we configure the scaling factor to 32, with low and high frequency factors of 1.0 and 4.0, respectively. We set the original context length to 8K and the base frequency ($\theta$) to 500,000. However, for the sliding window attention component, we use a smaller $\theta$ value of 10,000. We tie the weights of the embedding layer and the classifier head. Recent Mamba 2 reproductions show $A$ and dt_bias initialization heavily impacts performance. While we retained the original random initialization for dt_bias, we randomly sampled $A \in [0.01, 2.0]$ (original: $[1, 16]$).

Table 8: **Architectural configurations for Transformer and Mamba models across different scales.** For clarity, we separately detail the specific configurations for Self-Attention and SSM components. When constructing the inter- and intra-layer hybrid architectures, we refer to these configurations as the base computational primitives. The sliding window attention models use a window size of 512 and an attention sink size of 64 under the same Llama configuration.

| Models | Sizes | Base Configuration | | | | | | | Self-Attention | | SSM | | | |
|---|---|---|---|---|---|---|---|---|---|---|---|---|---|---|
| | | N-emb | Emb | Vocab | $N_L$ | $d_{\text{model}}$ | $d_{\text{ffn}}$ | $N_{\text{head}}$ | $N_{kv}$ | $d_{\text{head}}$ | $d_{\text{ssm}}$ | $d_{\text{head}}$ | $d_{\text{state}}$ | $N_{\text{conv}}$ |
| Llama | 100M | 0.10B | 0.13B | 128K | 8 | 1024 | 3072 | 16 | 4 | 64 | - | - | - | - |
| | 350M | 0.35B | 0.20B | 128K | 14 | 1536 | 4096 | 24 | 8 | 64 | - | - | - | - |
| | 1B | 0.97B | 0.26B | 128K | 16 | 2048 | 8192 | 32 | 8 | 64 | - | - | | |
| | 3B | 2.78B | 0.39B | 128K | 28 | 3072 | 8192 | 32 | 8 | 96 | - | - | - | - |
| Mamba | 100M | 0.10B | 0.13B | 128K | 6 | 1024 | 3072 | 16 | - | - | 2048 | 128 | 128 | 4 |
| | 350M | 0.37B | 0.20B | 128K | 11 | 1536 | 4096 | 24 | - | - | 3072 | 128 | 128 | 4 |
| | 1B | 0.98B | 0.26B | 128K | 13 | 2048 | 8192 | 32 | - | - | 4096 | 128 | 128 | 4 |
| | 3B | 2.80B | 0.39B | 128K | 21 | 3072 | 8192 | 32 | - | - | 6144 | 192 | 256 | 4 |

**Training settings.** We conduct experiments using TorchTitan (Liang et al., 2024), a PyTorch-native platform designed for large-scale training of LLMs. We pretrain different hybrid models variants from scratch on DCLM-Baseline pretraining dataset (Li et al., 2024), which contains 4T tokens from 3B documents. We pretrain on randomly sampled 60B tokens using 8 H200 GPUs by default. We pack the corpus using the Llama 3.2 tokenizer (Llama Team, 2024), which has a vocabulary size of 128K. A `bos` token is prepended to every sample before packing them into a context length of 8K tokens. We enable the model to attend to all previous tokens, not just those within the same document, while the `eos` token is used to distinguish document boundaries. We employ only FSDP (Zhao et al., 2023) with a degree equal to the number of GPUs, and activation checkpointing to reduce memory footprint. We also utilize `torch.compile` (Ansel et al., 2024) to accelerate training. We use a batch size of 2M tokens and utilize a trapezoid learning rate scheduler (Xing et al., 2018) consisting of warmup (about 25%), stable, and cool down (about 20%) phases. The AdamW optimizer (Loshchilov & Hutter, 2017) and gradient clipping are used for all experiments. For the different model scales—100M, 350M, 1B, and 3B parameters—the learning rates are set to 6e-3, 3e-3, 6e-4, and 3e-4, respectively.

**Evaluation settings.** To assess model quality, we evaluate language modeling performance using 8,000 samples from the validation set of DCLM-Baseline (Li et al., 2024) and 150 samples from the test set of PG19 dataset (Rae et al., 2019b). Additionally, following the settings of prior work (Gemini Team et al., 2024; Ho et al., 2024), we conduct evaluations on the Needle-In-a-Haystack long-context benchmark (Kamradt, 2023; Kuratov et al., 2024). We further measure few-shot accuracy on five benchmarks using the Language Model Evaluation Harness (Gao et al., 2024): LAMBADA (LD), HellaSwag (HS), PIQA (PQ), ARC (Easy and Challenge), and OpenBookQA (OB). For all the few-shot datasets except LAMBADA, accuracy is normalized by the byte length of the target string. We adhere to the standard number of shots for each dataset. All evaluation experiments are performed on a single H200 GPU.

# F   Expanded Main Results on Quality

We evaluate the pretraining results across two model sizes (350M and 1B) under both data and FLOPs budget constraints, as the FLOPs computation per token varies across model architectures. For the SWA and hybrid architectures, we investigate block ratios (expensive primitive : cheap primitive) of 1:1, 1:3, 1:5, and 1:12. For the SWA model, we follow the optimal design configuration detailed in Appendix G. Similarly, for inter- and intra-hybrids, we adopt the optimal design choices identified in §4.5 and §4.6. Tables 9 and 10 summarizes the key configurations, DCLM and PG19 validation losses, and few-shot accuracies for each model.

Table 9: **Performance comparison of 1B models under data and FLOPs constraints.** $N_{\text{hyb}}$ denotes the number of intra-hybrid block primitive. We calculate total training compute FLOPs and cache sizes for a sequence length of 8K tokens. SWA and hybrid architectures, which mix two different types of blocks, are evaluated at ratios of 1:1, 1:3, 1:5, 1:12, or 0:1.

| Models | Base Config | | | | | Compute | | | NLL (↓) | | Few-shot Accuracy (↑) | | | | | |
|---|---|---|---|---|---|---|---|---|---|---|---|---|---|---|---|---|
| | N-emb | $N_{\text{attn}}$ | $N_{\text{swa}}$ | $N_{\text{ssm}}$ | $N_{\text{hyb}}$ | Tok | FLOPs | Cache | DCLM | PG19 | LD | HS | PQ | ARC | OB | Avg |
| Llama | 0.97B | 16 | - | - | - | 60B | 4.5 e20 | 256 | 2.750 | 2.875 | 56.1 | 55.2 | 72.8 | 43.2 | 32.7 | 52.0 |
| SWA | 0.97B | 8 | 8 | - | - | 60B | 3.8 e20 | 137 | 2.745 | 2.865 | 54.9 | 55.7 | 72.4 | 42.8 | 35.8 | 52.3 |
| | 0.97B | 4 | 12 | - | - | 60B | 3.8 e20 | 78 | 2.740 | 2.862 | 57.5 | 55.5 | 72.5 | 43.6 | 34.6 | 52.7 |
| | 0.97B | 3 | 13 | - | - | 60B | 3.8 e20 | 63 | 2.741 | 2.867 | 56.0 | 55.9 | 72.6 | 44.2 | 34.8 | 52.7 |
| Mamba | 0.99B | - | - | 13 | - | 60B | 3.7 e20 | 13 | 2.758 | 2.891 | 53.7 | 55.1 | 73.8 | 43.9 | 35.2 | 52.3 |
| Inter-H | 0.96B | 7 | - | 7 | - | 60B | 3.9 e20 | 119 | 2.725 | 2.847 | 57.8 | 57.1 | 73.0 | 45.5 | 36.6 | 54.0 |
| | 0.94B | 3 | - | 10 | - | 60B | 3.6 e20 | 58 | 2.733 | 2.859 | 57.2 | 56.5 | 73.7 | 47.0 | 35.8 | 54.0 |
| | 0.96B | 2 | - | 11 | - | 60B | 3.7 e20 | 43 | 2.735 | 2.861 | 56.4 | 55.4 | 73.1 | 44.8 | 36.6 | 53.3 |
| | 0.97B | 1 | - | 12 | - | 60B | 3.7 e20 | 28 | 2.741 | 2.866 | 56.6 | 55.8 | 72.5 | 45.1 | 35.4 | 53.1 |
| Intra-H | 0.95B | - | - | 0 | 13 | 60B | 4.2 e20 | 180 | 2.712 | 2.831 | 59.8 | 57.9 | 73.1 | 47.3 | 36.2 | 54.9 |
| | 0.97B | - | - | 6 | 7 | 60B | 3.9 e20 | 89 | 2.720 | 2.843 | 58.7 | 57.4 | 74.1 | 47.0 | 35.2 | 54.5 |
| | 0.98B | - | - | 10 | 3 | 60B | 3.8 e20 | 50 | 2.724 | 2.848 | 55.9 | 57.0 | 74.5 | 46.5 | 35.4 | 53.9 |
| | 0.98B | - | - | 11 | 2 | 60B | 3.7 e20 | 38 | 2.728 | 2.853 | 58.7 | 57.4 | 73.3 | 47.9 | 36.4 | 54.7 |
| | 0.99B | - | - | 12 | 1 | 60B | 3.7 e20 | 23 | 2.736 | 2.863 | 56.3 | 56.3 | 73.6 | 44.9 | 35.6 | 53.4 |
| SWA | 0.97B | 8 | 8 | - | - | 66B | 4.5 e20 | 137 | 2.725 | 2.843 | 58.9 | 56.8 | 73.1 | 48.1 | 33.4 | 54.1 |
| | 0.97B | 4 | 12 | - | - | 70B | 4.5 e20 | 78 | 2.724 | 2.847 | 56.9 | 57.3 | 74.3 | 45.7 | 35.4 | 53.9 |
| | 0.97B | 3 | 13 | - | - | 71B | 4.5 e20 | 63 | 2.724 | 2.845 | 57.0 | 56.9 | 73.1 | 46.0 | 36.0 | 53.8 |
| | 0.97B | 1 | 15 | - | - | 73B | 4.5 e20 | 33 | 2.725 | 2.853 | 58.1 | 57.2 | 74.1 | 47.3 | 36.2 | 54.6 |
| | 0.97B | 0 | 16 | - | - | 74B | 4.5 e20 | 18 | 2.743 | 2.894 | 58.8 | 57.3 | 74.4 | 48.3 | 35.8 | 54.9 |
| Mamba | 0.97B | - | - | 13 | - | 73B | 4.5 e20 | 13 | 2.736 | 2.865 | 55.2 | 57.1 | 73.8 | 45.3 | 36.0 | 53.5 |
| Inter-H | 0.96B | 7 | - | 7 | - | 68B | 4.5 e20 | 119 | 2.710 | 2.836 | 58.8 | 57.9 | 73.5 | 44.0 | 34.8 | 53.8 |
| | 0.94B | 3 | - | 10 | - | 73B | 4.5 e20 | 58 | 2.716 | 2.836 | 54.9 | 56.5 | 73.8 | 44.5 | 36.2 | 53.2 |
| | 0.96B | 2 | - | 11 | - | 73B | 4.5 e20 | 43 | 2.716 | 2.842 | 56.7 | 57.6 | 73.3 | 46.5 | 35.8 | 54.0 |
| | 0.97B | 1 | - | 12 | - | 73B | 4.5 e20 | 28 | 2.721 | 2.843 | 53.4 | 57.3 | 73.7 | 45.7 | 37.8 | 53.6 |
| Intra-H | 0.95B | - | - | 0 | 13 | 65B | 4.5 e20 | 180 | 2.707 | 2.826 | 61.1 | 57.6 | 74.1 | 47.8 | 36.2 | 55.3 |
| | 0.97B | - | - | 6 | 7 | 69B | 4.5 e20 | 89 | 2.706 | 2.825 | 59.1 | 57.8 | 73.5 | 47.0 | 37.4 | 55.0 |
| | 0.98B | - | - | 10 | 2 | 71B | 4.5 e20 | 50 | 2.709 | 2.830 | 58.5 | 58.0 | 74.2 | 46.1 | 35.6 | 54.5 |
| | 0.98B | - | - | 11 | 2 | 72B | 4.5 e20 | 38 | 2.709 | 2.831 | 58.4 | 57.7 | 74.4 | 46.9 | 37.0 | 54.9 |
| | 0.98B | - | - | 12 | 1 | 73B | 4.5 e20 | 23 | 2.717 | 2.843 | 54.9 | 56.7 | 74.2 | 46.3 | 36.8 | 53.8 |

Table 10: **Performance comparison of 350M models under data and FLOPs constraints.** $N_{\mathrm{hyb}}$ denotes the number of intra-hybrid block primitive. We calculate total training compute FLOPs and cache sizes for a sequence length of 8K tokens. SWA and hybrid architectures, which mix two different types of blocks, are evaluated at ratios of 1:1, 1:3, 1:5, 1:12, and 0:1.

| Models | Base Config | | | | | Compute | | | NLL (↓) | | Few-shot Accuracy (↑) | | | | | |
| | N-emb | $N_{\mathrm{attn}}$ | $N_{\mathrm{swa}}$ | $N_{\mathrm{ssm}}$ | $N_{\mathrm{hyb}}$ | Tok | FLOPs | Cache | DCLM | PG19 | LD | HS | PQ | ARC | OB | Avg |
|---|---|---|---|---|---|---|---|---|---|---|---|---|---|---|---|---|
| Llama | 0.35B | 14 | - | - | - | 60B | 1.9 e20 | 224 | 2.882 | 3.015 | 51.5 | 48.3 | 70.4 | 40.4 | 33.0 | 48.7 |
| SWA | 0.35B | 3 | 11 | - | - | 60B | 1.5 e20 | 60 | 2.869 | 3.005 | 50.2 | 48.7 | 70.0 | 40.7 | 32.8 | 48.5 |
| Mamba | 0.35B | - | - | 11 | - | 60B | 1.4 e20 | 9 | 2.880 | 3.024 | 49.4 | 49.1 | 71.3 | 41.1 | 32.6 | 48.7 |
| Inter-H | 0.35B | 6 | - | 6 | - | 60B | 1.6 e20 | 103 | 2.850 | 2.985 | 52.8 | 50.3 | 71.0 | 40.7 | 32.0 | 49.3 |
| | 0.34B | 3 | - | 8 | - | 60B | 1.4 e20 | 57 | 2.858 | 2.994 | 53.3 | 50.4 | 72.2 | 40.7 | 34.4 | 50.2 |
| | 0.35B | 2 | - | 9 | - | 60B | 1.4 e20 | 42 | 2.860 | 2.999 | 51.7 | 49.8 | 71.7 | 40.4 | 33.4 | 49.4 |
| | 0.36B | 1 | - | 10 | - | 60B | 1.4 e20 | 27 | 2.864 | 3.003 | 48.9 | 49.8 | 72.4 | 41.0 | 34.4 | 49.3 |
| Intra-H | 0.36B | - | - | 0 | 11 | 60B | 1.9 e20 | 110 | 2.847 | 2.983 | 51.5 | 50.0 | 71.5 | 41.1 | 32.8 | 49.4 |
| | 0.35B | - | - | 5 | 6 | 60B | 1.5 e20 | 60 | 2.854 | 2.988 | 51.8 | 50.1 | 70.6 | 40.3 | 34.0 | 49.4 |
| | 0.39B | - | - | 8 | 3 | 60B | 1.5 e20 | 34 | 2.854 | 2.991 | 51.0 | 49.7 | 70.1 | 40.6 | 32.8 | 49.0 |
| | 0.36B | - | - | 9 | 2 | 60B | 1.4 e20 | 26 | 2.858 | 2.997 | 52.4 | 49.6 | 71.1 | 42.4 | 33.2 | 49.8 |
| | 0.37B | - | - | 10 | 1 | 60B | 1.4 e20 | 17 | 2.861 | 2.999 | 51.7 | 49.7 | 70.8 | 40.8 | 34.0 | 49.4 |
| SWA | 0.35B | 3 | 11 | - | - | 78B | 1.9 e20 | 60 | 2.855 | 2.988 | 52.4 | 50.3 | 70.7 | 42.1 | 33.6 | 49.8 |
| Mamba | 0.35B | - | - | 11 | - | 82B | 1.9 e20 | 9 | 2.858 | 3.001 | 52.3 | 50.3 | 71.0 | 41.6 | 34.2 | 49.9 |
| Inter-H | 0.35B | 6 | - | 6 | - | 72B | 1.9 e20 | 103 | 2.840 | 2.973 | 52.9 | 51.4 | 71.5 | 42.9 | 34.6 | 50.7 |
| | 0.34B | 3 | - | 8 | - | 80B | 1.9 e20 | 57 | 2.842 | 2.980 | 53.8 | 51.3 | 71.2 | 41.6 | 32.6 | 50.1 |
| | 0.35B | 2 | - | 9 | - | 81B | 1.9 e20 | 42 | 2.842 | 2.983 | 52.1 | 51.1 | 71.3 | 41.9 | 33.0 | 49.9 |
| | 0.36B | 1 | - | 10 | - | 81B | 1.9 e20 | 27 | 2.844 | 2.984 | 52.8 | 51.0 | 71.2 | 41.3 | 35.6 | 50.4 |
| Intra-H | 0.36B | - | - | 0 | 11 | 67B | 1.9 e20 | 110 | 2.842 | 2.981 | 55.9 | 51.1 | 70.9 | 41.5 | 33.6 | 50.6 |
| | 0.35B | - | - | 5 | 6 | 75B | 1.9 e20 | 60 | 2.841 | 2.976 | 51.2 | 51.0 | 71.0 | 41.4 | 31.8 | 49.3 |
| | 0.39B | - | - | 8 | 3 | 78B | 1.9 e20 | 34 | 2.838 | 2.972 | 53.0 | 50.7 | 70.5 | 42.4 | 32.0 | 49.7 |
| | 0.36B | - | - | 9 | 2 | 79B | 1.9 e20 | 26 | 2.839 | 2.977 | 54.0 | 50.9 | 70.9 | 42.1 | 34.8 | 50.5 |
| | 0.37B | - | - | 10 | 1 | 81B | 1.9 e20 | 17 | 2.845 | 2.982 | 50.6 | 50.6 | 71.3 | 43.1 | 33.6 | 49.8 |

# G   Expanded Results of Sliding Window Attention Ablation

To ensure optimal performance of SWA models, we conducted the following ablation study. In our SWA configuration, we utilize an attention sink size of 64 and a sliding window size of 512. Table 11 details our experiments on the optimal placement of self-attention layers (i.e., Transformer blocks) across three distinct block ratios. Consistent with findings in other hybrid architectures, Mamba-Transformer models, positioning the standard self-attention blocks in the middle layers proves most beneficial for overall performance. Based on these results, we adopt this mid-layer placement strategy for all main experiments.

Table 11: **Detailed ablation results for SWA model with positioning strategies across various block ratios.** Comparison includes different attention-to-SWA ratios at 1B scale, pre-trained on 60B tokens. 'Middle' indicates a configuration where the Transformer blocks are scattered throughout, while 'Front' and 'End' represent ablation settings where the blocks are placed at the very beginning or the very end, respectively. FLOPs and cache are calculated based on an 8K token sequence length. We highlight the best-performing settings in gray.

| | Base Config | | | Positioning | | Compute | | | NLL ($\downarrow$) | | Few-shot Accuracy ($\uparrow$) | | | | | |
|---|---|---|---|---|---|---|---|---|---|---|---|---|---|---|---|---|
| Ratio | N-emb | $N_{attn}$ | $N_{swa}$ | Pattern | Attn Indices | Tok | FLOPs | Cache | DCLM | PG19 | LD | HS | PQ | ARC | OB | Avg |
| 1:1 | 0.97B | 8 | 8 | Front | $\{1, 3, \ldots, 15\}$ | 60B | 3.8 e20 | 137 | 2.745 | **2.865** | 54.9 | 55.7 | 72.4 | 42.8 | **35.8** | 52.3 |
| 1:1 | 0.97B | 8 | 8 | End | $\{2, 4, \ldots, 16\}$ | 60B | 3.8 e20 | 137 | **2.743** | 2.868 | **55.3** | 55.9 | **73.5** | 44.8 | 35.2 | **52.9** |
| 1:3 | 0.97B | 4 | 12 | Front | $\{1, 4, 7, 10\}$ | 60B | 3.8 e20 | 78 | 2.742 | 2.867 | 56.7 | 55.5 | **73.7** | 44.5 | 34.0 | **52.9** |
| 1:3 | 0.97B | 4 | 12 | Middle | $\{4, 7, 10, 13\}$ | 60B | 3.8 e20 | 78 | **2.740** | 2.862 | **57.5** | 55.5 | 72.5 | 43.6 | 34.6 | 52.7 |
| 1:3 | 0.97B | 4 | 12 | End | $\{4, 7, 10, 16\}$ | 60B | 3.8 e20 | 78 | 2.742 | 2.867 | 55.3 | 56.3 | 73.5 | 44.3 | **35.2** | 52.9 |
| 1:5 | 0.97B | 3 | 13 | Front | $\{1, 9, 13\}$ | 60B | 3.7 e20 | 63 | 2.742 | 2.868 | 55.8 | **56.0** | 72.5 | 43.5 | 32.8 | 52.1 |
| 1:5 | 0.97B | 3 | 13 | Middle | $\{5, 9, 13\}$ | 60B | 3.7 e20 | 63 | **2.741** | **2.867** | 56.0 | 55.9 | 72.6 | 44.2 | **34.8** | 52.7 |
| 1:5 | 0.97B | 3 | 13 | End | $\{5, 9, 16\}$ | 60B | 3.7 e20 | 63 | **2.741** | 2.868 | **56.4** | **56.0** | **72.9** | **45.3** | 33.8 | **52.9** |

# H   Expanded Results on Longer Context Pretraining

While our main experiments summarize the pretraining results for an 8K context length, we further conduct experiments on the extended contexts of 16K and 32K. Table 12 summarizes the measured perplexity and few-shot accuracy. Even at 2x and 4x extended context lengths, hybrid architectures demonstrate significantly superior performance compared to their counterparts. Notably, while Transformers suffer from performance degradation, Mamba-hybrid models achieve excellent performance while benefiting from a substantially smaller cache size.

Table 12: **Performance comparison of 1B models on extended contexts under FLOPs constraints.** The top and bottom blocks summarize the performance at 16K and 32K contexts, respectively. $N_{hyb}$ denotes the number of intra-hybrid block primitive. We calculate total training compute FLOPs and cache sizes for a sequence length of 16K and 32K tokens. We use the block ratio of 1:5 for SWA and hybrid architectures.

| | Base Config | | | | | Compute | | | | NLL ($\downarrow$) | | Few-shot Accuracy ($\uparrow$) | | | | | |
|---|---|---|---|---|---|---|---|---|---|---|---|---|---|---|---|---|---|---|
| Models | N-emb | $N_{attn}$ | $N_{swa}$ | $N_{ssm}$ | $N_{hyb}$ | Len | Tok | FLOPs | Cache | DCLM | PG19 | LD | HS | PQ | ARC | OB | Avg |
| Llama | 0.97B | 16 | - | - | - | 16K | 60B | 4.5 e20 | 512 | 2.703 | 2.759 | 57.0 | 57.3 | 73.5 | 46.0 | 36.0 | 54.0 |
| SWA | 0.97B | 3 | 13 | - | - | 16K | 67B | 4.5 e20 | 110 | 2.672 | 2.729 | 57.6 | 59.2 | 73.9 | 46.8 | 36.4 | 54.8 |
| Mamba | 0.99B | - | - | 13 | - | 16K | 73B | 4.5 e20 | 13 | 2.686 | 2.755 | 56.1 | 59.3 | 73.9 | **49.3** | 36.4 | 55.0 |
| Inter-H | 0.96B | 2 | - | 11 | - | 16K | 71B | 4.5 e20 | 75 | 2.667 | 2.725 | **58.8** | 59.0 | **74.6** | 46.9 | **38.6** | 55.6 |
| Intra-H | 0.98B | - | - | 11 | 2 | 16K | 70B | 4.5 e20 | 63 | **2.661** | **2.721** | **58.8** | **60.0** | **74.6** | 47.1 | 38.0 | **55.7** |
| Llama | 0.97B | 16 | - | - | - | 32K | 60B | 4.5 e20 | 1024 | 2.807 | 2.831 | 51.9 | 52.5 | 71.8 | 42.5 | 35.0 | 50.7 |
| SWA | 0.97B | 3 | 13 | - | - | 32K | 62B | 4.5 e20 | 207 | 2.738 | 2.750 | 53.8 | 52.6 | 70.7 | 43.6 | 34.0 | 51.0 |
| Mamba | 0.99B | - | - | 13 | - | 32K | 73B | 4.5 e20 | 13 | **2.716** | 2.744 | 53.7 | 58.0 | **73.8** | 45.2 | **36.2** | 53.4 |
| Inter-H | 0.96B | 2 | - | 11 | - | 32K | 67B | 4.5 e20 | 139 | 2.722 | **2.735** | **58.1** | **58.3** | 73.5 | 45.9 | 34.4 | **54.0** |
| Intra-H | 0.98B | - | - | 11 | 2 | 32K | 67B | 4.5 e20 | 113 | 2.722 | **2.735** | 54.6 | 57.7 | 73.5 | **46.9** | 35.2 | 53.5 |

## I    Experimental Setup for Downstream Fine-tuning Tasks

We extend our evaluation to downstream summarization and question-answering (QA) tasks using the five 1B checkpoints listed in Table 2, all pre-trained with an identical budget of 4e20 FLOPs. The benchmark suite includes GovReport (GR; Huang et al. (2021)) and two anonymized tasks (Task 𝔸 and Task 𝔹) for summarization, alongside SQuAD (SQA; Rajpurkar et al. (2016)), TriviaQA (TQA; Joshi et al. (2017)), and an anonymized task (Task ℂ) for QA. Fine-tuning is performed for 1,000 steps with a global batch size of 16, utilizing a trapezoidal learning rate scheduler with a peak of 1e-4, configured with 100 warmup steps and a 200-step cool down phase. The maximum sequence length is set to 8K for all datasets, except SQuAD (2K). Within this window, we allocate specific maximum lengths for targets (summaries or answers): and 128 for QA datasets, dedicating the remaining capacity to the input context. Unused slots are padded with EOS tokens and masked during training. For evaluation, we sample 1,000 test instances per dataset, with the exception of SQuAD where the validation set is used.

## J    Expanded Main Results on Efficiency

Table 13 provides more details of throughput and latency across different architectures and block ratios. Specifically, we measure these metrics at a batch size of 4 with a fixed prompt length of 512, across sequence lengths ranging from 2K to 32K. As illustrated in the Figures 1b and 3a, hybrid architectures demonstrate exceptionally high throughput across all settings.

Table 13: **Throughput and latency comparison of 1B models across diverse architectures.** We evaluate the models with a batch size of 4 and a 512-token prompt, scaling the total sequence lengths from 2K up to 32K (e.g., generating 1,536 tokens for a total length of 2K). $N_{hyb}$ denotes the number of intra-hybrid block primitive. For architectures utilizing two distinct computational primitives (SWA and two hybrid models), we measure performance by ablating the block ratios across 1:0, 1:1, 1:3, 1:5, 1:12, and 0:1.

| Models | Base Config | | | | | Throughput (↑) | | | | | Latency (↓) | | | | |
|---|---|---|---|---|---|---|---|---|---|---|---|---|---|---|---|
| | N-emb | $N_{attn}$ | $N_{swa}$ | $N_{ssm}$ | $N_{hyb}$ | 2K | 4K | 8K | 16K | 32K | 2K | 4K | 8K | 16K | 32K |
| Llama | 0.97B | 16 | - | - | - | 373 | 214 | 115 | 60 | 30 | 16 | 67 | 267 | 1063 | 4240 |
| SWA | 0.97B | 8 | 8 | - | - | 331 | 244 | 158 | 93 | 51 | 19 | 59 | 195 | 684 | 2527 |
| | 0.97B | 4 | 12 | - | - | 343 | 291 | 219 | 147 | 89 | 18 | 49 | 141 | 432 | 1443 |
| | 0.97B | 3 | 13 | - | - | 345 | 302 | 241 | 173 | 110 | 18 | 47 | 127 | 368 | 1172 |
| | 0.97B | 1 | 15 | - | - | 354 | 335 | 307 | 263 | 204 | 17 | 43 | 100 | 241 | 632 |
| | 0.97B | 0 | 16 | - | - | 309 | 308 | 324 | 308 | 307 | 20 | 47 | 95 | 206 | 420 |
| Mamba | 0.99B | - | - | 13 | - | 2528 | 2647 | 2738 | 2782 | 2917 | 2 | 5 | 11 | 23 | 44 |
| Inter-H | 0.96B | 7 | - | 7 | - | 669 | 418 | 241 | 130 | 68 | 9 | 34 | 127 | 489 | 1903 |
| | 0.94B | 3 | - | 10 | - | 971 | 722 | 462 | 272 | 150 | 6 | 20 | 67 | 233 | 860 |
| | 0.96B | 2 | - | 11 | - | 969 | 828 | 595 | 373 | 213 | 6 | 17 | 52 | 170 | 605 |
| | 0.97B | 1 | - | 12 | - | 969 | 859 | 754 | 546 | 355 | 6 | 17 | 41 | 116 | 363 |
| Intra-H | 0.95B | - | - | 0 | 13 | 509 | 326 | 187 | 101 | 53 | 12 | 44 | 164 | 626 | 2437 |
| | 0.97B | - | - | 7 | 6 | 708 | 496 | 308 | 175 | 95 | 9 | 29 | 100 | 363 | 1355 |
| | 0.98B | - | - | 10 | 2 | 1056 | 795 | 562 | 360 | 205 | 6 | 18 | 55 | 176 | 630 |
| | 0.98B | - | - | 11 | 2 | 1089 | 918 | 683 | 464 | 280 | 6 | 16 | 45 | 137 | 461 |
| | 0.98B | - | - | 12 | 1 | 1325 | 1184 | 999 | 739 | 487 | 5 | 12 | 31 | 86 | 265 |

## K   Experimental Setup for Long-context Evaluation Tasks

**PG19 language modeling task.**   Following prior work (Xiao et al., 2023; Yu et al., 2023; Zhang et al., 2023; Ho et al., 2024), we evaluate position-wise perplexity on the PG19 test set (Rae et al., 2019b). Given that PG19 documents are sufficiently long, we evaluate 30 randomly selected samples individually without input packing. We use 1B-parameter models pretrained with a FLOP budget of 4.5e20 and an 8K context length (see Table 2). Although the models are pretrained on 8K lengths, we extend the maximum length of the positional encodings in the Transformer components to 24K for evaluation.

**Needle-In-a-Haystack task.**   To precisely evaluate the long-context modeling capabilities of LLMs, recent benchmarks such as Needle-In-a-Haystack (NIAH) (Kamradt, 2023), LongBench (Bai et al., 2024), and RULER (Hsieh et al., 2024) are widely utilized. However, since our models are just pre-trained without instruction tuning, standard instruction-based prompts may not be optimal. Therefore, we slightly modified the evaluation prompts to suit the nature of base models. Following prior work (Reid et al., 2024), we construct the context using concatenated essays by Paul Graham as the "haystack". We evaluate the model across a range of sequence lengths $L \in \{960, 1984, \ldots, 16320\}$ and insert a "needle" containing key information at specific depth percentages $d \in \{0\%, 10\%, \ldots, 100\%\}$. Consistent with Reid et al. (2024), we use the needle format: "The special magic {city} number is: {number}", where {city} is a randomly chosen city name and {number} is a random 7-digit number.

To query the model, we append a *verbatim* prompt configured as follows: "<context>\ncontext\n</context> \n\nquestion\n\nThe special magic {city} number is:". By utilizing the exact same format as the inserted needle, we frame the retrieval task as a pattern completion problem suitable for pre-trained models. Finally, we measure accuracy by generating 20 subsequent tokens and considering a prediction correct if the generated text contains the target 7-digit number. Since the models are pretrained with an 8K context length, we extend the maximum length of the positional encodings accordingly to accommodate longer sequences during evaluation.

## L   Experimental Setup for Integration of Mixture-of-Experts

We train a Mixture-of-Experts (MoE) variant utilizing a token-choice routing mechanism, where the feed-forward networks (FFNs) in all layers are replaced with MoE layers. Drawing inspiration from recent MoE architectures (DeepSeek-AI (2024), Llama 4), we incorporate one shared expert and select the top-1 expert from a pool of eight. To maintain parity in total parameter count and computational cost with the dense baseline, we halve the intermediate dimension of the expert FFNs.

Regarding the routing strategy, we apply a `sigmoid` function to scale the router scores, following prior work (DeepSeek-AI, 2024). To address load balancing without introducing auxiliary losses, we adopt an auxiliary loss-free balancing algorithm (Wang et al., 2024b; DeepSeek-AI, 2024) with a bias update coefficient of 1e-3. Finally, to accelerate training without resorting to expert parallelism, we perform parallel computation using `torch._grouped_mm`.

## M    Expanded Results of Scaling Laws Experiment

We conduct a scaling law analysis by training four model scales (100M, 350M, 1B, and 3B) across five compute budgets (ranging from 2e19 to 4e20 FLOPs). We efficiently conduct all budget experiments with a Trapezoid learning rate scheduler and reusing intermediate checkpoints from the training run with the largest budget. The warmup stage is therefore defined as one-quarter of the step count of the smallest budget. The learning rates used for the 100M, 350M, 1B, and 3B models are 6e-3, 3e-3, 6e-4, and 3e-4, respectively. We use 0.5M tokens/step, and all other settings, such as the context length of 8K, remain identical to the main experiment. The models are trained on the DCLM training set and evaluated on 8,000 validation samples. Based on this, we derive the optimal compute-scaling lines, as shown in the right panel of Table 4. The IsoFLOPs analysis results for each architecture are visualized in Figure M, with detailed numerical results summarized in Tables 14 and 15.

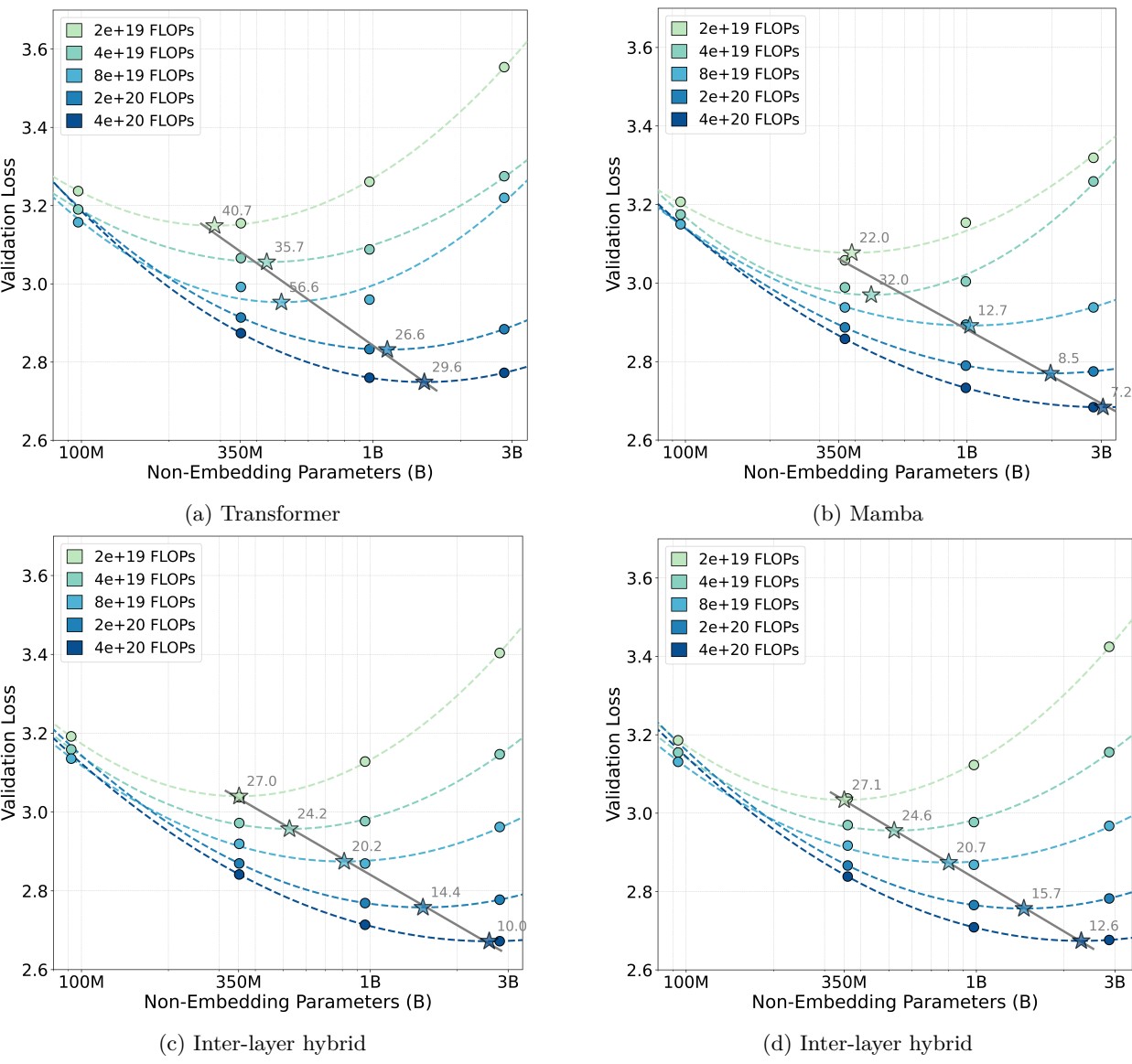

Figure 5: Visualization of the scaling law results for each architecture. We apply a quadratic fit to the model results at each compute budget. Star markers denote the optimal parameter size that achieves the best quality for each budget. The numbers shown on the fitted optimal scaling line represent the ratio of the number of tokens to the parameter count.

Table 14: **Detailed scaling law experiments for Transformer and Mamba models.** We pre-train four scales of models, up to 3B parameters, from scratch across five Training FLOPs budgets and evaluated their negative log-likelihood (NLL). The models are trained on the DCLM training set and evaluated on 8,000 validation samples.

| Models | N-emb | Base Config | | | | | | | | | | Compute | | | NLL ($\downarrow$) |
|---|---|---|---|---|---|---|---|---|---|---|---|---|---|---|---|
| | | $N_{attn}$ | $N_{ssm}$ | $N_{hyb}$ | $d_{model}$ | $d_{ffn}$ | $N_{head}$ | $N_{kv}$ | $d_{ssm}$ | $d_{state}$ | $N_{conv}$ | FLOPs | Tok | Steps | DCLM |
| Llama | 0.10B | 8 | - | - | 1024 | 3072 | 16 | 4 | - | - | - | 2e19 | 20B | 39K | 3.237 |
| | | | | | | | | | | | | 4e19 | 41B | 78K | 3.190 |
| | | | | | | | | | | | | 8e19 | 82B | 155K | 3.157 |
| | 0.35B | 14 | - | - | 1536 | 4096 | 24 | 8 | - | - | - | 2e19 | 6B | 12K | 3.155 |
| | | | | | | | | | | | | 4e19 | 13B | 24K | 3.066 |
| | | | | | | | | | | | | 8e19 | 25B | 48K | 2.992 |
| | | | | | | | | | | | | 2e20 | 63B | 120K | 2.914 |
| | | | | | | | | | | | | 4e20 | 126B | 241K | 2.874 |
| | 0.97B | 16 | - | - | 2048 | 8192 | 32 | 8 | - | - | - | 2e19 | 3B | 5K | 3.261 |
| | | | | | | | | | | | | 4e19 | 5B | 10K | 3.088 |
| | | | | | | | | | | | | 8e19 | 11B | 20K | 2.960 |
| | | | | | | | | | | | | 2e20 | 27B | 51K | 2.833 |
| | | | | | | | | | | | | 4e20 | 54B | 102K | 2.760 |
| | 2.82B | 28 | - | - | 3072 | 8192 | 32 | 8 | - | - | - | 2e19 | 1B | 2K | 3.554 |
| | | | | | | | | | | | | 4e19 | 2B | 4K | 3.275 |
| | | | | | | | | | | | | 8e19 | 4B | 7K | 3.212 |
| | | | | | | | | | | | | 2e20 | 10B | 18K | 2.884 |
| | | | | | | | | | | | | 4e20 | 19B | 37K | 2.773 |
| Mamba | 0.10B | - | 6 | - | 1024 | 3072 | 16 | - | 2048 | 128 | 4 | 2e19 | 27B | 51K | 3.173 |
| | | | | | | | | | | | | 4e19 | 54B | 102K | 3.134 |
| | 0.34B | - | 11 | - | 1536 | 4096 | 24 | - | 3072 | 128 | 4 | 2e19 | 8B | 16K | 3.050 |
| | | | | | | | | | | | | 4e19 | 17B | 32K | 2.983 |
| | | | | | | | | | | | | 8e19 | 34B | 65K | 2.933 |
| | | | | | | | | | | | | 2e20 | 85B | 162K | 2.884 |
| | | | | | | | | | | | | 4e20 | 170B | 324K | 2.855 |
| | 0.99B | - | 13 | - | 2048 | 8192 | 32 | - | 4096 | 128 | 4 | 2e19 | 3B | 6K | 3.159 |
| | | | | | | | | | | | | 4e19 | 6B | 12K | 3.003 |
| | | | | | | | | | | | | 8e19 | 12B | 24K | 2.885 |
| | | | | | | | | | | | | 2e20 | 31B | 59K | 2.772 |
| | | | | | | | | | | | | 4e20 | 62B | 119K | 2.712 |
| | 2.81B | - | 21 | - | 3072 | 8192 | 32 | - | 6144 | 256 | 4 | 2e19 | 1B | 2K | 3.367 |
| | | | | | | | | | | | | 4e19 | 2B | 4K | 3.133 |
| | | | | | | | | | | | | 8e19 | 4B | 8K | 2.962 |
| | | | | | | | | | | | | 2e20 | 11B | 20K | 2.790 |
| | | | | | | | | | | | | 4e20 | 21B | 41K | 2.691 |

Table 15: **Detailed scaling law experiments for Inter- and Intra-layer hybrid models.** Both hybrid models use a 1:5 block ratio. The Inter-hybrid model adopts a structure where the global attention layers are uniformly distributed throughout the depth (see Section 4.5). The Intra-hybrid model follows the final structure selected within the block (see Section 4.6). The global attention component fundamentally follows the structure of the Differential Transformer (Ye et al., 2024). In this setup, the query and key have a half-size dimension ($d_{\text{head}}/2$) corresponding to the half the number of assigned heads, while the value retains the full $d_{\text{model}}$. For the Mamba component, a separate output projection is used to restore the $d_{\text{model}}$ from the $d_{\text{ssm}}$ before the two modules are fused. To ensure a fair parameter count, the 1B Intra-layer hybrid model specifically uses a query and key dimension of 1152 and a $d_{\text{ssm}}$ of 2304.

| Models | N-emb | Base Config | | | | | | | | | | Compute | | | NLL ($\downarrow$) |
| | | $N_{\text{attn}}$ | $N_{\text{ssm}}$ | $N_{\text{hyb}}$ | $d_{\text{model}}$ | $d_{\text{ffn}}$ | $N_{\text{head}}$ | $N_{\text{kv}}$ | $d_{\text{ssm}}$ | $d_{\text{state}}$ | $N_{\text{conv}}$ | FLOPs | Tok | Steps | DCLM |
|---|---|---|---|---|---|---|---|---|---|---|---|---|---|---|---|
| Inter-H | 0.09B | 5 | 1 | - | 1024 | 3072 | 16 | 4 | 2048 | 128 | 4 | 2e19 | 31B | 60K | 3.192 |
| | | | | | | | | | | | | 4e19 | 63B | 119K | 3.159 |
| | | | | | | | | | | | | 8e19 | 125B | 239K | 3.136 |
| | 0.35B | 2 | 9 | - | 1536 | 4096 | 24 | 8 | 3072 | 128 | 4 | 2e19 | 8B | 16K | 3.037 |
| | | | | | | | | | | | | 4e19 | 17B | 32K | 2.972 |
| | | | | | | | | | | | | 8e19 | 34B | 65K | 2.919 |
| | | | | | | | | | | | | 2e20 | 85B | 162K | 2.870 |
| | | | | | | | | | | | | 4e20 | 170B | 324K | 2.842 |
| | 0.96B | 2 | 11 | - | 2048 | 8192 | 32 | 8 | 4096 | 128 | 4 | 2e19 | 3B | 6K | 3.128 |
| | | | | | | | | | | | | 4e19 | 7B | 12K | 2.977 |
| | | | | | | | | | | | | 8e19 | 13B | 25K | 2.869 |
| | | | | | | | | | | | | 2e20 | 33B | 62K | 2.769 |
| | | | | | | | | | | | | 4e20 | 65B | 125K | 2.714 |
| | 2.81B | 4 | 18 | - | 3072 | 8192 | 32 | 8 | 6144 | 256 | 4 | 2e19 | 1B | 2K | 3.403 |
| | | | | | | | | | | | | 4e19 | 2B | 4K | 3.147 |
| | | | | | | | | | | | | 8e19 | 4B | 8K | 2.962 |
| | | | | | | | | | | | | 2e20 | 11B | 21K | 2.777 |
| | | | | | | | | | | | | 4e20 | 22B | 42K | 2.672 |
| Intra-H | 0.10B | - | 5 | 1 | 1024 | 3072 | 8+8 | 2 | 1024 | 128 | 4 | 2e19 | 31B | 59K | 3.186 |
| | | | | | | | | | | | | 4e19 | 62B | 118K | 3.155 |
| | | | | | | | | | | | | 8e19 | 124B | 236K | 3.131 |
| | 0.36B | - | 9 | 2 | 1536 | 4096 | 16+16 | 4 | 1536 | 128 | 4 | 2e19 | 8B | 16K | 3.037 |
| | | | | | | | | | | | | 4e19 | 17B | 32K | 2.969 |
| | | | | | | | | | | | | 8e19 | 33B | 64K | 2.917 |
| | | | | | | | | | | | | 2e20 | 83B | 160K | 2.867 |
| | | | | | | | | | | | | 4e20 | 167B | 318K | 2.838 |
| | 0.98B | - | 11 | 2 | 2048 | 8192 | 16+16 | 4 | 2304 | 128 | 4 | 2e19 | 3B | 6K | 3.123 |
| | | | | | | | | | | | | 4e19 | 6B | 12K | 2.977 |
| | | | | | | | | | | | | 8e19 | 13B | 24K | 2.868 |
| | | | | | | | | | | | | 2e20 | 32B | 61K | 2.765 |
| | | | | | | | | | | | | 4e20 | 64B | 122K | 2.709 |
| | 2.89B | - | 18 | 4 | 3072 | 8192 | 16+16 | 4 | 3072 | 256 | 4 | 2e19 | 1B | 2K | 3.425 |
| | | | | | | | | | | | | 4e19 | 2B | 4K | 3.156 |
| | | | | | | | | | | | | 8e19 | 4B | 8K | 2.968 |
| | | | | | | | | | | | | 2e20 | 11B | 20K | 2.782 |
| | | | | | | | | | | | | 4e20 | 21B | 41K | 2.676 |

# N Expanded Results of Inter-layer Hybridization Ablation

## N.1 Insights for Block Ratios

In Table 16., we present the comprehensive results of our block ratio ablation study across both 350M and 1B parameter scales. While the balanced 1:1 ratio achieves the highest overall quality, its heavy reliance on self-attention introduces a significant inference bottleneck due to quadratic complexity. As we increase the proportion of Mamba blocks (e.g., 1:3, 1:5, 1:12), FLOPs and cache requirements decrease significantly with only a marginal degradation in few-shot accuracy and perplexity. This consistent trend across both scales demonstrates that a 1:5 ratio strikes the optimal balance; it effectively mitigates the computational overhead of Transformer blocks, almost matching the efficiency of pure Mamba models.

Table 16: **Detailed ablation results related to block ratios for inter-layer hybridization at 1B and 350M scale.** All models are pretrained on 60B tokens from the DCLM dataset. We calculate total training compute FLOPs and cache sizes for a sequence length of 8K tokens. We highlight the best-performing settings in gray, considering compute, NLL, and few-shot accuracy.

| Sizes | Ratio | Base Config | | | Compute | | | NLL (↓) | | Few-shot Accuracy (↑) | | | | | |
|---|---|---|---|---|---|---|---|---|---|---|---|---|---|---|---|
| | | N-emb | $N_{\text{attn}}$ | $N_{\text{ssm}}$ | Tok | FLOPs | Cache | DCLM | PG19 | LD | HS | PQ | ARC | OB | Avg |
| 1B | 1:0 | 0.97B | 16 | - | 60B | 4.5 e20 | 256 | 2.750 | 2.875 | 56.1 | 55.2 | 72.8 | 43.2 | 32.7 | 52.0 |
| | 0:1 | 0.99B | - | 13 | 60B | 3.7 e20 | 13 | 2.758 | 2.891 | 53.7 | 55.1 | **73.8** | 43.9 | 35.2 | 52.3 |
| | 1:1 | 0.96B | 7 | 7 | 60B | 3.9 e20 | 119 | **2.725** | **2.847** | **57.8** | **57.1** | 73.0 | 45.5 | **36.6** | **54.0** |
| | 1:3 | 0.94B | 3 | 10 | 60B | 3.6 e20 | 58 | 2.733 | 2.859 | 57.2 | 56.5 | 73.7 | **47.0** | 35.8 | **54.0** |
| | 1:5 | 0.96B | 2 | 11 | 60B | 3.7 e20 | 43 | 2.735 | 2.861 | 56.4 | 55.4 | 73.1 | 44.8 | **36.6** | 53.3 |
| | 1:12 | 0.97B | 1 | 12 | 60B | 3.7 e20 | 28 | 2.741 | 2.866 | 56.6 | 55.8 | 72.5 | 45.1 | 35.4 | 53.1 |
| 350M | 1:0 | 0.35B | 14 | - | 60B | 1.9 e20 | 224 | 2.882 | 3.015 | 51.5 | 48.3 | 70.4 | 40.4 | 33.0 | 48.7 |
| | 0:1 | 0.35B | - | 11 | 60B | 1.4 e20 | 9 | 2.880 | 3.024 | 49.4 | 49.1 | 71.3 | **41.1** | 32.6 | 48.7 |
| | 1:1 | 0.35B | 6 | 6 | 60B | 1.6 e20 | 103 | **2.850** | **2.985** | 52.8 | 50.3 | 71.0 | 40.7 | 32.0 | 49.3 |
| | 1:3 | 0.34B | 3 | 8 | 60B | 1.4 e20 | 57 | 2.858 | 2.994 | **53.3** | **50.4** | 72.2 | 40.7 | **34.4** | **50.2** |
| | 1:5 | 0.35B | 2 | 9 | 60B | 1.4 e20 | 42 | 2.860 | 2.999 | 51.7 | 49.8 | 71.7 | 40.4 | 33.4 | 49.4 |
| | 1:12 | 0.36B | 1 | 10 | 60B | 1.4 e20 | 27 | 2.864 | 3.003 | 48.9 | 49.8 | **72.4** | 41.0 | **34.4** | 49.3 |

## N.2 Insights for Positioning of Computational Primitives

Table 17 details our comprehensive ablation study on the placement of Transformer blocks within the inter-hybrid architecture. To thoroughly investigate how the spatial distribution of self-attention layers impacts overall model quality, we evaluated various positioning strategies—categorized generally into Front, Middle, and End placements—across multiple block ratios (1:1, 1:3, 1:5, and 1:12) and two distinct parameter scales (350M and 1B). The extended results across both validation NLL and few-shot accuracy metrics consistently reaffirm that distributing Transformer blocks toward the middle or later stages is crucial. This trend robustly holds regardless of the specific model scale or the proportion of Mamba blocks, demonstrating that avoiding early-layer attention is a universal requisite for maximizing the capacity of hybrid architectures.

Table 17: **Detailed ablation results related to block positioning for inter-layer hybridization at 1B and 350M scale.** We denote the indices of the Transformer blocks. 'Middle' indicates a configuration where the Transformer blocks are scattered throughout, while 'Front' and 'End' represent ablation settings where the blocks are placed at the very beginning or the very end, respectively. All models are pretrained on 60B tokens from the DCLM dataset. We highlight the best-performing settings in gray, considering NLL and few-shot accuracy.

| Sizes | Ratio | N-emb | $N_{attn}$ | $N_{ssm}$ | Pattern | Attn Indices | DCLM | PG19 | LD | HS | PQ | ARC | OB | Avg |
|---|---|---|---|---|---|---|---|---|---|---|---|---|---|---|
| | | **Base Config** | | | **Positioning** | | **NLL ($\downarrow$)** | | **Few-shot Accuracy ($\uparrow$)** | | | | | |
| 1B | $1:0$ | 0.97B | 16 | - | - | $\{1,\ldots,16\}$ | 2.750 | 2.875 | 56.1 | 55.2 | 72.8 | 43.2 | 32.7 | 52.0 |
| | $0:1$ | 0.99B | - | 13 | - | $\emptyset$ | 2.758 | 2.891 | 53.7 | 55.1 | 73.8 | 43.9 | 35.2 | 52.3 |
| | $1:1$ | 0.96B | 7 | 7 | Front | $\{1,3,\ldots,15\}$ | 2.733 | 2.855 | 57.2 | 56.5 | 72.8 | 45.0 | 36.0 | 53.6 |
| | $1:1$ | 0.96B | 7 | 7 | End | $\{2,4,\ldots,16\}$ | **2.725** | **2.847** | **57.8** | **57.1** | **73.0** | **45.5** | **36.6** | **54.0** |
| | $1:3$ | 0.94B | 3 | 10 | Front | $\{1,7,11\}$ | 2.750 | 2.875 | 56.8 | 55.8 | **73.7** | 44.3 | 35.0 | 53.1 |
| | $1:3$ | 0.94B | 3 | 10 | Middle | $\{3,7,11\}$ | **2.733** | **2.859** | **57.2** | **56.5** | **73.7** | **47.0** | 35.8 | **54.0** |
| | $1:3$ | 0.94B | 3 | 10 | End | $\{3,7,13\}$ | 2.735 | **2.858** | **57.2** | 55.9 | 73.1 | 45.4 | **36.2** | 53.6 |
| | $1:5$ | 0.96B | 2 | 11 | Front | $\{1,8\}$ | 2.750 | 2.878 | **56.4** | 55.8 | 73.1 | **45.5** | 34.6 | 53.1 |
| | $1:5$ | 0.96B | 2 | 11 | Middle | $\{4,8\}$ | **2.735** | **2.861** | **56.4** | 55.4 | 73.1 | 44.8 | **36.6** | 53.3 |
| | $1:5$ | 0.96B | 2 | 11 | End | $\{4,11\}$ | 2.740 | 2.865 | 55.5 | **56.0** | **74.2** | 45.3 | 36.4 | **53.5** |
| | $1:12$ | 0.97B | 1 | 12 | Single | $\{1\}$ | 2.771 | 2.906 | 52.9 | 55.2 | 72.8 | **45.4** | 35.4 | 52.3 |
| | $1:12$ | 0.97B | 1 | 12 | Single | $\{2\}$ | 2.754 | 2.885 | 54.3 | 55.7 | 73.2 | 45.3 | 35.6 | 52.8 |
| | $1:12$ | 0.97B | 1 | 12 | Single | $\{3\}$ | 2.748 | 2.880 | 54.2 | 55.6 | **74.1** | 45.0 | 35.4 | 52.9 |
| | $1:12$ | 0.97B | 1 | 12 | Single | $\{4\}$ | 2.742 | 2.872 | 52.2 | 56.0 | 73.7 | 44.4 | 35.0 | 52.3 |
| | $1:12$ | 0.97B | 1 | 12 | Single | $\{7\}$ | 2.741 | **2.866** | **56.6** | 55.8 | 72.5 | 45.1 | 35.4 | 53.1 |
| | $1:12$ | 0.97B | 1 | 12 | Single | $\{9\}$ | **2.740** | **2.866** | 55.1 | 55.9 | 73.1 | 44.1 | 34.0 | 52.4 |
| | $1:12$ | 0.97B | 1 | 12 | Single | $\{11\}$ | 2.741 | 2.868 | 56.0 | **56.6** | 73.6 | 45.0 | **36.0** | 53.5 |
| | $1:12$ | 0.97B | 1 | 12 | Single | $\{12\}$ | 2.746 | 2.871 | 55.3 | 55.4 | 72.5 | 44.4 | 34.8 | 52.5 |
| 350M | $1:0$ | 0.35B | 14 | - | - | $\{1,\ldots,14\}$ | 2.882 | 3.015 | 51.5 | 48.3 | 70.4 | 40.4 | 33.0 | 48.7 |
| | $0:1$ | 0.35B | - | 11 | - | $\emptyset$ | 2.880 | 3.024 | 49.4 | 49.1 | 71.3 | 41.1 | 32.6 | 48.7 |
| | $1:1$ | 0.35B | 6 | 6 | Front | $\{1,3,\ldots,11\}$ | 2.869 | 3.003 | 51.8 | 49.3 | 70.5 | **41.0** | **35.2** | **49.6** |
| | $1:1$ | 0.35B | 6 | 6 | End | $\{2,4,\ldots,12\}$ | **2.850** | **2.985** | **52.8** | **50.3** | **71.0** | 40.7 | 32.0 | 49.3 |
| | $1:3$ | 0.34B | 3 | 8 | Front | $\{1,6,9\}$ | 2.873 | 3.012 | 51.1 | 49.4 | 71.0 | 40.6 | 33.2 | 49.1 |
| | $1:3$ | 0.34B | 3 | 8 | Middle | $\{3,6,9\}$ | **2.858** | **2.994** | **53.3** | **50.4** | **72.2** | 40.7 | **34.4** | **50.2** |
| | $1:3$ | 0.34B | 3 | 8 | End | $\{3,6,11\}$ | 2.860 | 2.997 | 51.8 | 49.8 | 70.0 | **41.6** | 34.0 | 49.5 |
| | $1:5$ | 0.35B | 2 | 9 | Front | $\{1,8\}$ | 2.880 | 3.019 | 51.3 | 48.7 | 71.5 | **40.4** | 33.2 | 49.0 |
| | $1:5$ | 0.35B | 2 | 9 | Middle | $\{4,8\}$ | **2.860** | **2.999** | **51.7** | **49.8** | **71.7** | **40.4** | **33.4** | **49.4** |
| | $1:5$ | 0.35B | 2 | 9 | End | $\{4,11\}$ | 2.861 | **2.999** | 50.0 | 49.2 | 71.0 | 39.7 | 31.8 | 48.4 |
| | $1:12$ | 0.36B | 1 | 10 | Single | $\{1\}$ | 2.898 | 3.039 | 48.0 | 48.5 | 70.0 | 41.3 | 34.2 | 48.4 |
| | $1:12$ | 0.36B | 1 | 10 | Single | $\{2\}$ | 2.879 | 3.023 | 48.5 | 49.3 | 70.2 | 40.5 | **34.6** | 48.6 |
| | $1:12$ | 0.36B | 1 | 10 | Single | $\{3\}$ | 2.874 | 3.014 | 47.8 | 49.2 | 71.0 | 41.0 | 31.8 | 48.2 |
| | $1:12$ | 0.36B | 1 | 10 | Single | $\{4\}$ | 2.869 | 3.008 | 50.1 | 49.3 | 70.9 | 41.0 | 32.4 | 48.7 |
| | $1:12$ | 0.36B | 1 | 10 | Single | $\{6\}$ | **2.861** | **2.999** | 51.7 | **49.7** | 70.8 | 40.8 | 34.0 | 49.4 |
| | $1:12$ | 0.36B | 1 | 10 | Single | $\{8\}$ | 2.862 | 3.000 | **51.8** | **49.7** | **71.1** | 41.5 | 33.4 | **49.5** |
| | $1:12$ | 0.36B | 1 | 10 | Single | $\{11\}$ | 2.866 | 3.002 | 49.7 | 49.1 | 69.9 | **41.7** | 33.4 | 48.8 |

## O    Visualization of Attention Scores

Figures 6 and 7 visualize the attention scores of Transformer and Mamba, respectively. For Mamba, we visualize the implicit attention map of the SSM module by following the approach of Deci-Mamba (Ben-Kish et al., 2024). In the case of the Transformer, the attention scores in the initial layers show a highly uniform distribution, suggesting that these layers primarily serve to encode the hidden states. We confirm that this phenomenon is not unique to homogeneous models; even in an inter-layer hybrid setting, front-placed Transformer blocks exhibit the exact same uniform distribution as in a pure Transformer. In contrast, Mamba predominantly captures local context, as evidenced by a strong focus along the diagonal and very small effective receptive fields (ERFs). This is the opposite of the Transformer's tendency to attend to a wider range of token positions. As a result, using Transformer blocks in the early stages of an inter-layer hybrid architecture may negatively impact performance due to this uniform attending behavior.

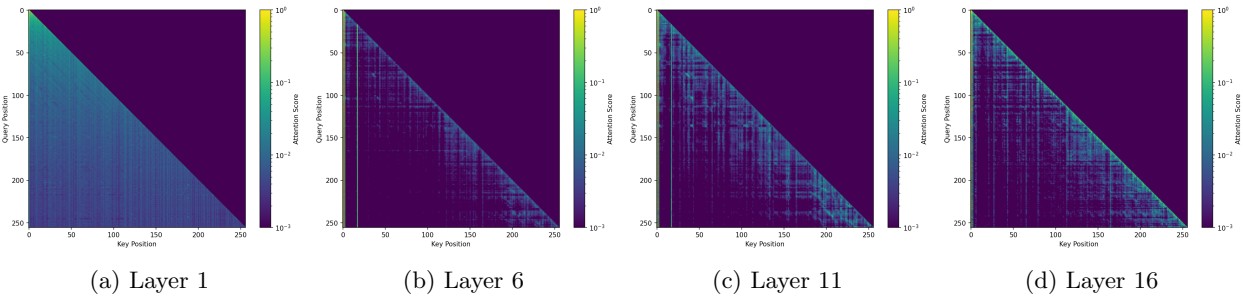

(a) Layer 1                (b) Layer 6                (c) Layer 11                (d) Layer 16

Figure 6: **Qualitative results of attention scores from Llama 1B model**. We measure the score at a context length of 256 using the PG19 test set. The evaluation is conducted with a Llama 1B model trained on 60B tokens. Values closer to yellow indicate scores near 1, while values closer to purple indicate scores near 0.

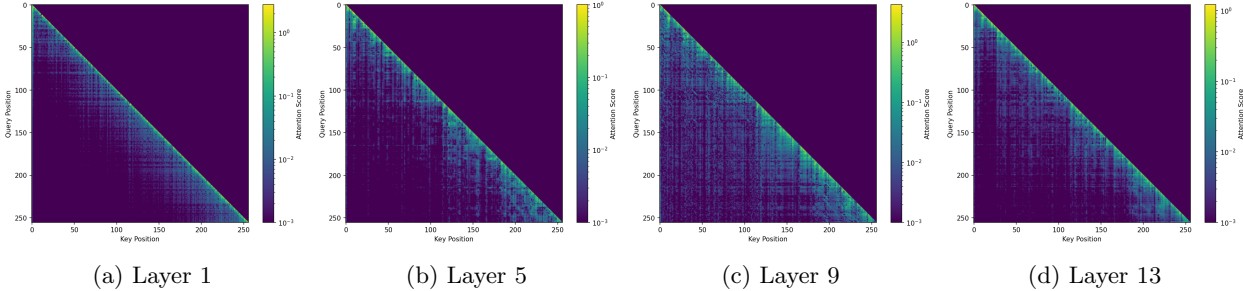

(a) Layer 1                (b) Layer 5                (c) Layer 9                (d) Layer 13

Figure 7: **Qualitative results of attention scores from Mamba 1B model.** We measure the score at a context length of 256 using the PG19 test set. The evaluation is conducted with a Mamba 1B model trained on 60B tokens. Values closer to yellow indicate scores near 1, while values closer to purple indicate scores near 0. We take the absolute value of the implicit attention scores in Mamba and normalize each row by its maximum value.

# P Expanded Results of Intra-layer Hybridization Ablation

## P.1 Insights for Architectural Variants

To thoroughly analyze the design space of intra-layer hybrid architectures, we conduct a comprehensive evaluation at both the 350M and 1B parameter scales. We specifically examine six pivotal design axes: primitive types, normalization layers, scaling formulations, fusion operations, value sharing strategies, and output projection configurations. Tables 18 and 19 present the results of our ablation study.

Due to the significant variance in few-shot accuracy, especially at smaller scales like 350M, we prioritize the NLL metric and the number of layers ($N_L$)—a key factor for efficiency—to select the best-performing candidates (highlighted in gray). Consequently, we determine the Diff fusion strategy as our final choice for the remaining experiments.

Table 18: **Detailed ablation results related to architectural variants for intra-layer hybridization at 1B scale.** All models are pretrained on 60B tokens from the DCLM dataset. We replace all depths to intra-hybrid blocks (1:0 ratio), which consist of global attention (GQA) and Mamba primitives in a head-wise splitting manner. † denotes our re-implementation of prior methods, incorporating value dimension expansion (Ye et al., 2024) for GQA. We highlight the final best-performing candidates in gray. Note that we prioritize the NLL metric given the high variance observed in few-shot accuracy.

| Models | Base Config | | Design Choices | | | | | | NLL (↓) | | Few-shot Accuracy (↑) | | | | | |
|---|---|---|---|---|---|---|---|---|---|---|---|---|---|---|---|---|
| | N-emb | $N_L$ | Module | Norm | Scalar | Fusion | Value | Out | DCLM | PG19 | LD | HS | PQ | ARC | OB | Avg |
| Llama | 0.97B | 16 | - | - | - | - | - | - | 2.750 | 2.875 | **56.1** | 55.2 | 72.8 | 43.2 | 32.7 | 52.0 |
| Mamba | 0.99B | 13 | - | - | - | - | - | - | 2.758 | 2.891 | 53.7 | 55.1 | 73.8 | 43.9 | 35.2 | 52.3 |
| Diff-T | 0.97B | 16 | T / T | - | Diff-T | Diff | Joint | Joint | 2.727 | 2.848 | 58.2 | 57.1 | 72.9 | 44.3 | 35.4 | 53.6 |
| Diff-M | 0.99B | 13 | M / M | - | Diff-M | Diff | Sep | Sep | 2.759 | 2.892 | 55.0 | 56.0 | 73.7 | 45.2 | 35.0 | 53.0 |
| ∟Variants | 0.97B | 16 | T / T | - | Diff-T | Add | Joint | Joint | 2.733 | 2.856 | 56.0 | 56.5 | 73.4 | 45.5 | 36.0 | 53.5 |
| | 0.96B | 15 | T / M | - | Diff-T | Diff | Joint | Joint | 2.733 | 2.854 | 56.3 | 56.5 | 74.3 | 46.4 | 36.8 | 54.1 |
| | 0.96B | 15 | T / M | Group | Diff-T | Diff | Joint | Joint | 2.735 | 2.855 | 55.1 | 56.3 | 73.3 | 45.0 | 35.2 | 53.0 |
| | 0.94B | 14 | T / M | - | Diff-T | Diff | Sep | Joint | 2.731 | 2.852 | 56.7 | 57.0 | 74.1 | 46.1 | 36.8 | 54.1 |
| | 0.94B | 14 | T / M | Group | Diff-T | Diff | Sep | Joint | 2.728 | 2.852 | 57.4 | 55.9 | 73.1 | 46.5 | 35.4 | 53.6 |
| | 0.95B | 13 | T / M | Group | Diff-T | Diff | Sep | Sep | 2.719 | 2.838 | 58.3 | 57.2 | 73.2 | 45.9 | 35.4 | 54.0 |
| Hymba† | 0.94B | 14 | T / M | Group | Scale | Add | Sep | Joint | 2.726 | 2.846 | 56.2 | 56.8 | 73.2 | 45.9 | 36.2 | 53.7 |
| ∟Variants | 0.94B | 14 | T / M | - | - | Add | Sep | Joint | 2.738 | 2.861 | 58.2 | 56.5 | **74.2** | 46.3 | 37.0 | 54.4 |
| | 0.94B | 14 | T / M | Group | - | Add | Sep | Joint | 2.721 | 2.840 | 59.0 | 57.2 | 73.6 | 45.8 | 37.0 | 54.5 |
| | 0.94B | 14 | T / M | - | Scale | Add | Sep | Joint | 2.740 | 2.865 | 57.7 | 56.2 | 73.8 | 45.6 | 36.0 | 53.9 |
| | 0.96B | 15 | T / M | Group | Scale | Add | Joint | Joint | 2.729 | 2.850 | 56.9 | 57.2 | 73.9 | 45.8 | 36.6 | 54.1 |
| | 0.95B | 13 | T / M | - | - | Add | Sep | Sep | 2.720 | 2.842 | 58.8 | 57.3 | 73.0 | 45.5 | 35.8 | 54.1 |
| | 0.95B | 13 | T / M | Group | - | Add | Sep | Sep | **2.714** | 2.834 | 58.4 | 57.6 | 74.7 | 46.3 | 35.6 | 54.5 |
| | 0.95B | 13 | T / M | Group | Scale | Add | Sep | Sep | 2.720 | 2.841 | 57.7 | 56.4 | 72.9 | 45.2 | 35.2 | 53.5 |
| | 0.95B | 13 | T / M | - | - | Diff | Sep | Sep | 2.715 | 2.836 | 59.7 | 57.5 | 73.7 | 46.4 | 35.8 | 54.6 |
| | 0.95B | 13 | T / M | Group | - | Diff | Sep | Sep | 2.712 | 2.831 | **59.8** | **57.9** | 73.1 | **47.3** | 36.2 | **54.9** |
| | 0.95B | 13 | T / M | Group | Scale | Diff | Sep | Sep | 2.727 | 2.846 | 59.3 | 56.5 | 73.0 | 44.8 | 37.2 | 54.2 |
| | 0.94B | 14 | T / M | Group | - | Diff | Sep | Joint | 2.721 | 2.842 | 58.3 | 57.0 | 73.6 | 45.1 | 35.8 | 54.0 |
| Falcon-H1† | 0.95B | 13 | T / M | - | - | Conc | Sep | Joint | 2.720 | 2.840 | 56.6 | 56.8 | 73.0 | 46.3 | 35.2 | 53.6 |
| ∟Variants | 0.95B | 13 | T / M | Group | - | Conc | Sep | Joint | 2.715 | **2.833** | 59.0 | 57.7 | 73.7 | 45.8 | **37.6** | 54.8 |
| | 0.95B | 13 | T / M | Group | Scale | Conc | Sep | Joint | 2.724 | 2.842 | 57.8 | 57.0 | 73.6 | 47.2 | 36.0 | 54.3 |

Table 19: **Detailed ablation results related to architectural variants for intra-layer hybridization at 350M scale.** All models are pretrained on 60B tokens from the DCLM dataset. We replace all depths to intra-hybrid blocks (1:0 ratio), which consist of global attention (GQA) and Mamba primitives in a head-wise splitting manner. † denotes our re-implementation of prior methods, incorporating value dimension expansion (Ye et al., 2024) for GQA. We highlight the final best-performing candidates in gray. Note that we prioritize the NLL metric given the high variance observed in few-shot accuracy.

| Models | Base Config | | Design Choices | | | | | | NLL (↓) | | Few-shot Accuracy (↑) | | | | | |
|---|---|---|---|---|---|---|---|---|---|---|---|---|---|---|---|---|
| | N-emb | $N_L$ | Module | Norm | Scalar | Fusion | Value | Out | DCLM | PG19 | LD | HS | PQ | ARC | OB | Avg |
| Llama | 0.35B | 14 | - | - | - | - | - | - | 2.882 | 3.015 | 51.5 | 48.3 | 70.4 | 40.4 | 33.0 | 48.7 |
| Mamba | 0.37B | 11 | - | - | - | - | - | - | 2.880 | 3.024 | 49.4 | 49.1 | 71.3 | 41.1 | 32.6 | 48.7 |
| Diff-T | 0.35B | 14 | T / T | - | Diff-T | Diff | Joint | Joint | 2.858 | 2.991 | 56.1 | 50.0 | 70.7 | 40.3 | 32.8 | 50.0 |
| Diff-M | 0.37B | 11 | M / M | - | Diff-M | Diff | Sep | Sep | 2.879 | 3.024 | 50.1 | 49.4 | **71.9** | 41.2 | 34.2 | 49.4 |
| └Variants | 0.35B | 14 | T / T | - | Diff-T | Add | Joint | Joint | 2.865 | 3.001 | 51.9 | 49.2 | 71.4 | 41.4 | 32.8 | 49.3 |
| | 0.35B | 13 | T / M | - | Diff-T | Diff | Joint | Joint | 2.862 | 2.994 | 52.8 | 49.5 | 70.5 | **42.6** | 33.4 | 49.7 |
| | 0.35B | 13 | T / M | Group | Diff-T | Diff | Joint | Joint | 2.859 | 2.990 | **50.6** | 49.5 | 71.8 | 40.2 | 32.4 | 48.9 |
| | 0.34B | 12 | T / M | - | Diff-T | Diff | Sep | Joint | 2.864 | 3.002 | 52.4 | 49.9 | 69.9 | 40.2 | 31.6 | 48.8 |
| | 0.34B | 12 | T / M | Group | Diff-T | Diff | Sep | Joint | 2.856 | 2.989 | 50.8 | 50.1 | 71.6 | 41.5 | 33.0 | 49.4 |
| | 0.36B | 11 | T / M | Group | Diff-T | Diff | Sep | Sep | 2.853 | 2.987 | 54.9 | 50.2 | 71.0 | 41.1 | 34.4 | 50.3 |
| Hymba† | 0.34B | 12 | T / M | Group | Scale | Add | Sep | Joint | 2.861 | 2.995 | 51.5 | 49.4 | 71.4 | 42.2 | 33.8 | 49.7 |
| └Variants | 0.34B | 12 | T / M | - | - | Add | Sep | Joint | 2.870 | 3.006 | 52.6 | 49.2 | 71.1 | 39.5 | 32.8 | 49.0 |
| | 0.34B | 12 | T / M | Group | - | Add | Sep | Joint | 2.857 | 2.989 | 54.0 | 50.0 | 71.0 | 41.5 | 34.2 | 50.1 |
| | 0.34B | 12 | T / M | - | Scale | Add | Sep | Joint | 2.872 | 3.010 | 54.1 | 49.3 | 71.3 | 40.5 | 33.0 | 49.6 |
| | 0.35B | 13 | T / M | Group | Scale | Add | Joint | Joint | 2.861 | 2.993 | 52.4 | 49.4 | 71.0 | 41.3 | 32.8 | 49.4 |
| | 0.36B | 11 | T / M | - | - | Add | Sep | Sep | 2.851 | 2.985 | 51.9 | 50.0 | 70.5 | 40.0 | 33.0 | 49.1 |
| | 0.36B | 11 | T / M | Group | - | Add | Sep | Sep | 2.849 | 2.985 | 51.6 | 50.6 | 71.6 | 40.8 | 33.8 | 49.7 |
| | 0.36B | 11 | T / M | Group | Scale | Add | Sep | Sep | 2.849 | 2.987 | 51.7 | 50.4 | 70.8 | 40.8 | 32.8 | 49.3 |
| | 0.36B | 11 | T / M | - | - | Diff | Sep | Sep | 2.850 | 2.983 | 53.6 | 50.5 | 71.4 | 41.2 | 34.0 | 50.1 |
| | 0.36B | 11 | T / M | Group | - | Diff | Sep | Sep | 2.847 | 2.983 | 51.5 | 50.0 | 71.5 | 41.1 | 32.8 | 49.4 |
| | 0.36B | 11 | T / M | Group | Scale | Diff | Sep | Sep | 2.851 | 2.987 | 54.0 | 49.7 | 71.2 | 41.5 | 34.0 | 50.1 |
| | 0.34B | 12 | T / M | Group | - | Diff | Sep | Joint | 2.858 | 2.990 | 54.3 | 49.9 | 70.9 | 42.4 | 33.6 | 50.2 |
| Falcon-H1† | 0.36B | 11 | T / M | - | - | Conc | Sep | Joint | 2.855 | 2.989 | 54.2 | 50.3 | 70.5 | 42.2 | 32.6 | 50.0 |
| └Variants | 0.36B | 11 | T / M | Group | - | Conc | Sep | Joint | **2.846** | **2.978** | 55.2 | **50.6** | 71.1 | 41.7 | **34.4** | **50.6** |
| | 0.36B | 11 | T / M | Group | Scale | Conc | Sep | Joint | 2.851 | 2.983 | 53.1 | 49.9 | 72.3 | 41.7 | 34.8 | 50.3 |

To provide a granular understanding of the design space, we detail our observations and design decisions for each individual axis below:

- **Modules**: The combination of global attention and Mamba significantly outperforms both pure homogeneous models and differential models (Ye et al., 2024; Schneider et al., 2025).

- **Normalization**: Applying `Group` normalization to the output of each module generally improves performance. This appears to stabilize optimization by addressing the significant scale discrepancy ($> 10\times$) between the two modules.

- **Scaling strategies**: While various strategies exist for learning scaling factors, specific methods like `Diff-T` from Differential Transformer (Ye et al., 2024) or the `Group` normalization scheme used in Hymba (Dong et al., 2024) actually lead to performance degradation.

- **Fusion operations**: The choice of fusion operation shows marginal impact on performance, though it closely interacts with value sharing and output projection configurations. We hypothesize that because Mamba's implicit attention scores can inherently be negative, specific fusion mechanisms are not strictly necessary.

- **Value sharing**: Sharing values between modules causes only a minor performance drop. However, under a fixed parameter budget, this strategy results in an increase in the layer count ($N_L$), hurting efficiency.

- **Output projections**: Maintaining and computing separate output projections for each module (prior to fusion) proves crucial for achieving significant performance gains.

## P.2 Insights for Dimension Ratios

Table 20 presents the comprehensive ablation results for dimension allocation across both 1B and 350M scales. The results confirm that enlarging the Transformer dimension (query/key dimension) relative to the Mamba dimension (pre-expansion dimension) consistently improves model quality. This suggests that the Transformer component is highly critical for the reasoning capabilities of the hybrid model. However, this quality improvement comes at the direct cost of increased cache memory and computational overhead. Given that parallel execution throughput is ultimately bottlenecked by the Transformer layers, we conclude that a balanced 1:1 dimension allocation remains the most practical choice to co-optimize both efficiency and downstream performance.

Table 20: **Detailed ablation results on dimension ratios for the intra-hybrid models.** We investigate the dimension allocation ratios within the intra-hybrid block, defined by the proportion of the query/key dimension in the Transformer versus the pre-expansion dimension in Mamba. Specifically, we experiment with dimension ratios of 1:0, 2:1, 1:1, 1:2, 1:10, and 0:1. We adopt an architecture variant featuring Group normalization, no scaling, concatenation fusion, separate values, and a joint output projection. We utilize a 1:0 block ratio, replacing all standard blocks entirely with intra-hybrid blocks, and pre-train the model on 60B tokens. FLOPs and cache are calculated based on an 8K token sequence length. We highlight our final design choice in gray, considering both compute efficiency and overall model quality.

| | Base Config | | | | Dimension | | Compute | | | NLL ($\downarrow$) | | Few-shot Accuracy ($\uparrow$) | | | | | |
|---|---|---|---|---|---|---|---|---|---|---|---|---|---|---|---|---|---|
| Size | N-emb | $N_{\text{attn}}$ | $N_{\text{ssm}}$ | $N_{\text{hyb}}$ | Trans. | Mamba | Tok | FLOPs | Cache | DCLM | PG19 | LD | HS | PQ | ARC | OB | Avg |
| **1B** | 0.97B | 16 | 0 | 0 | 2048 | 0 | 60B | 4.5e20 | 256 | 2.750 | 2.875 | 56.1 | 55.2 | 72.8 | 43.2 | 32.7 | 52.0 |
| | 0.95B | 0 | 0 | 13 | 1792 | 1024 | 60B | 4.5e20 | 189 | **2.711** | **2.831** | **60.6** | **57.7** | 72.6 | **47.1** | 35.4 | 54.7 |
| | 0.95B | 0 | 0 | 13 | 1152 | 1152 | 60B | 4.2e20 | 180 | 2.715 | 2.833 | 59.0 | **57.7** | 73.7 | 45.8 | **37.6** | **54.8** |
| | 0.95B | 0 | 0 | 13 | 512 | 1280 | 60B | 3.8e20 | 56 | 2.724 | 2.848 | 58.7 | 57.6 | 73.6 | 45.2 | 36.2 | 54.3 |
| | 0.95B | 0 | 0 | 13 | 128 | 1408 | 60B | 3.6e20 | 22 | 2.752 | 2.883 | 55.5 | 55.9 | **73.9** | 44.8 | 34.8 | 53.0 |
| | 0.99B | 0 | 13 | 0 | 0 | 2048 | 60B | 3.7e20 | 13 | 2.758 | 2.891 | 53.7 | 55.1 | 73.8 | 43.9 | 35.2 | 52.3 |
| **350M** | 0.35B | 14 | 0 | 0 | 1536 | 0 | 60B | 1.9e20 | 224 | 2.882 | 3.015 | 51.5 | 48.3 | 70.4 | 40.4 | 33.0 | 48.7 |
| | 0.36B | 0 | 0 | 11 | 1408 | 768 | 60B | 2.0e20 | 126 | 2.875 | 3.012 | 54.6 | 48.5 | 70.4 | **42.2** | 33.4 | 50.4 |
| | 0.36B | 0 | 0 | 11 | 896 | 896 | 60B | 1.9e20 | 110 | **2.846** | **2.978** | **55.2** | **50.6** | 71.1 | 41.7 | **34.4** | **50.6** |
| | 0.36B | 0 | 0 | 11 | 512 | 1024 | 60B | 1.6e20 | 50 | 2.878 | 3.017 | 52.3 | 48.1 | 70.8 | 41.2 | 32.4 | 49.7 |
| | 0.36B | 0 | 0 | 11 | 128 | 1152 | 60B | 1.4e20 | 17 | 2.913 | 3.057 | 48.2 | 46.8 | 70.6 | 41.8 | 33.8 | 48.8 |
| | 0.35B | 0 | 11 | 0 | 0 | 1536 | 60B | 1.4e20 | 9 | 2.880 | 3.024 | 49.4 | 49.1 | **71.3** | 41.1 | 32.6 | 48.7 |

## P.3 Insights for Block Ratios and Positioning of Primitives

Table 21 presents a granular breakdown of the block positioning ablation for the 1B-sized intra-hybrid model. We evaluate three distinct patterns—Cluster, Scatter, and Sandwich—across varying $N_{\text{hyb}}$ to $N_{\text{ssm}}$ ratios. The empirical results validate the Scatter approach; evenly distributing intra-hybrid layers consistently outperforms other configurations in both NLL and few-shot accuracy metrics, regardless of the block ratio.

Table 21: **Detailed ablation results on block positioning patterns for the 1B Intra-Hybrid model.** We define patterns based on the position of the intra-hybrid blocks: placing them consecutively (Cluster), distributing them evenly (Scatter), or placing them at the beginning and end as well (Sandwich). FLOPs and cache are calculated based on an 8K sequence length. We highlight our final design choice in gray.

| | Base Config | | | Positioning | | Compute | | | NLL ($\downarrow$) | | Few-shot Accuracy ($\uparrow$) | | | | | |
|---|---|---|---|---|---|---|---|---|---|---|---|---|---|---|---|---|
| Ratio | N-emb | $N_{\text{ssm}}$ | $N_{\text{hyb}}$ | Pattern | Intra-H Indices | Tok | FLOPs | Cache | DCLM | PG19 | LD | HS | PQ | ARC | OB | Avg |
| 1:1 | 0.97B | 6 | 7 | Cluster | $\{5, 6, \ldots, 10, 11\}$ | 60B | 3.9e20 | 89 | **2.719** | 2.843 | 58.1 | 57.0 | 73.1 | 46.0 | **36.0** | 54.0 |
| 1:1 | 0.97B | 6 | 7 | Scatter | $\{2, 4, 6, 8, 10, 12, 13\}$ | 60B | 3.9e20 | 89 | 2.720 | 2.843 | **58.9** | **57.4** | **74.1** | **46.9** | 35.2 | **54.5** |
| 1:1 | 0.97B | 6 | 7 | Sandwich | $\{1, 2, 6, 7, 8, 12, 13\}$ | 60B | 3.9e20 | 89 | 2.721 | **2.842** | 58.7 | 57.0 | 73.6 | 46.3 | 34.6 | 54.0 |
| 1:3 | 0.98B | 10 | 3 | Cluster | $\{9, 10, 11\}$ | 60B | 3.8e20 | 50 | 2.728 | 2.853 | **57.4** | 56.4 | 73.1 | 45.6 | **36.4** | 53.8 |
| 1:3 | 0.98B | 10 | 3 | Scatter | $\{4, 7, 11\}$ | 60B | 3.8e20 | 50 | **2.724** | **2.848** | 55.9 | **57.0** | **74.5** | **46.5** | 35.4 | **53.9** |
| 1:3 | 0.98B | 10 | 3 | Sandwich | $\{1, 7, 13\}$ | 60B | 3.8e20 | 50 | 2.736 | 2.857 | 54.1 | 55.7 | 74.3 | 44.7 | 35.8 | 52.9 |
| 1:5 | 0.98B | 11 | 2 | Cluster | $\{9, 10\}$ | 60B | 3.7e20 | 38 | 2.729 | 2.856 | 56.2 | 57.3 | **73.9** | 46.3 | 35.0 | 53.8 |
| 1:5 | 0.98B | 11 | 2 | Scatter | $\{5, 9\}$ | 60B | 3.7e20 | 38 | **2.728** | **2.853** | **58.7** | **57.4** | 73.3 | **47.9** | **36.4** | **54.7** |
| 1:5 | 0.98B | 11 | 2 | Sandwich | $\{1, 13\}$ | 60B | 3.7e20 | 38 | 2.746 | 2.870 | 57.0 | 56.0 | 72.9 | 41.8 | 34.2 | 52.4 |

## Q  Analysis of Component Contributions in Hybrid Architectures

We analyze the contribution of each module to determine the optimal activation patterns for the Transformer and Mamba components within a hybrid model. To this end, we pretrain a 350M-parameter intra-layer hybrid architecture from scratch, where every layer consists of an intra-hybrid block. In this setup, we divide the heads evenly, normalize the outputs of each module using Group normalization, and combine them via a weighted sum scaled by $\alpha$ and $1-\alpha$ before the output projection. Here, the value of $\alpha$ serves as a quantitative measure of each module's contribution to next-token prediction.

Using a checkpoint trained on approximately 4B tokens, we investigate how $\alpha$ values are determined on the DCLM validation set. Figure 8 illustrates the average coefficients across each layer for about 100 samples. Consistent with our ablation studies, the model assigns a higher weight to the Mamba component in earlier layers—particularly the first layer—to optimize performance, while assigning higher weights to the Transformer component in the middle layers (specifically layers 6 and 7). This indicates that the global attention pattern at earlier depths—which tends to uniformly attend to all previous contexts—is not optimal in this hybrid setting, while at intermediate depths, the global attention mechanism appears to compensate for Mamba's limited recall capability.

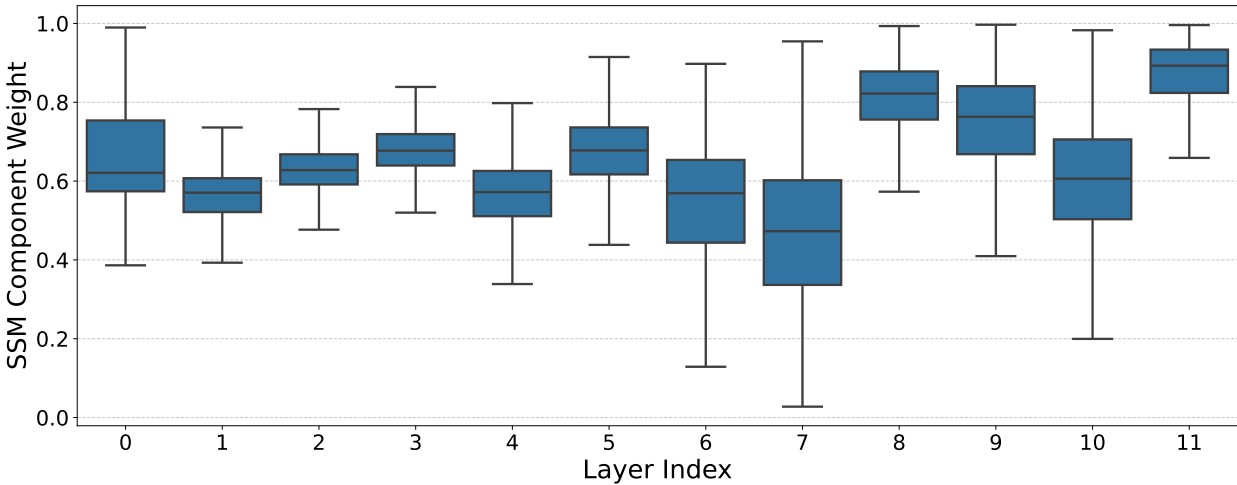

Figure 8: **Box plot of layer-wise component contributions to the output.** The y-axis value ($\alpha$) represents the weight assigned to the Mamba component, while $1-\alpha$ denotes the coefficient for the global attention module. We measure these values using a 12-layer 350M intra-hybrid model.

To provide a deeper analysis, we visualize the token-wise contributions in Table 22. We examine random samples to identify which module is strongly activated at specific layers for next-token prediction. There are several common and intriguing patterns. First, while the Mamba component plays a dominant role overall in the first layer, the Transformer component tends to be utilized more heavily at sentence boundaries—marked by punctuation such as periods (.) or question marks (?)—or at positions requiring the start of new information. Intuitively, these moments correspond to important 'forking tokens' for guiding reasoning; the Transformer appears to intervene specifically at these phrase starts to prevent the generation from being led astray.

Layer 6 exhibits a particularly interesting pattern: the model employs global attention to predict the next token when retrieving key context keywords or predicting numerical values and complex information. For example, in the first sample, global attention is activated to derive updated temporal information based on the context following the mention of a specific date like 'Oct 7'12'.

In the subsequent Layer 7, the Transformer component is frequently activated at the start of new syntactic units, such as after conjunctions, periods, or parentheses. Conversely, when a single word is split into sub-words by the tokenizer, the SSM component activates, leveraging its ability to process local information more efficiently.

Table 22: **Visualization of token-wise contribution from Transformer and Mamba modules.** We use an intra-layer hybrid 350M model where every layer consists of an intra-hybrid block. The fusion mechanism combines the outputs of each module by weighting them with $\alpha$ and $1-\alpha$, respectively. Red color corresponds to a larger coefficient for the Transformer component, and Blue corresponds to a larger coefficient for the Mamba component. For a model consisting of 12 layers in total, we visualize for three random samples at the first layer (index 0) and the middle layers (indices 6 and 7). Note that these coefficients are specifically optimized for *next*-token prediction.

| Layer | Text |
|---|---|
| Layer #0 | <\|begin_of_text\|> Take the 2 -minute tour × Here what happened with me today . Time Machine asked me whether I want to set a backup disk , I 've answered yes , but then , when I 've realized that in order to backup anything Time Machine will clean the disk , I 've changed my mind and canceled everything . And my disk suddenly became read only . What I 've tried before Go og ling : $ sudo ch flags - R n ouch g Elements / $ sudo chmod - R a +w Elements / But I 've failed with both of this , getting " read -only file system " messages . What I 've tried after Go og ling : 1 . Open Disk Utilities 2 . Click Repair Disk Permissions But this button is disabled , and I have no idea what exactly should be done to enable it . I have been using this disk for a quite a long time , and never had any permission issues with it . ( Disk is formatted as NT FS , if that helps . Capacity is 2 TB , of which 1 . 92 TB are available .) I 'd really appreciate if someone will give me a hint how this can be resolved . share \| im prove this question Try boot ing from the Recovery HD and see if the button is still disabled . – rien 333 Oct 7 ' 12 at 14 : 07 What exactly is it you need to do with the stuff on the NT FS disk ? – seg idd ins Oct 7 ' 12 at 15 : 26 @ Sam uel E .G idd ins I need to work with disk just as I 've worked yesterday , 2 days ago , 3 days ago etc . It looks like very annoying bug to me - to find out that out of a sudden this disk is considered to be read -only . – sh ab unc Oct 7 ' 12 at 15 : 34 do you have any add -ons that would |
| Layer #6 | <\|begin_of_text\|> Take the 2 -minute tour × Here what happened with me today . Time Machine asked me whether I want to set a backup disk , I 've answered yes , but then , when I 've realized that in order to backup anything Time Machine will clean the disk , I 've changed my mind and canceled everything . And my disk suddenly became read only . What I 've tried before Go og ling : $ sudo ch flags - R n ouch g Elements / $ sudo chmod - R a +w Elements / But I 've failed with both of this , getting " read -only file system " messages . What I 've tried after Go og ling : 1 . Open Disk Utilities 2 . Click Repair Disk Permissions But this button is disabled , and I have no idea what exactly should be done to enable it . I have been using this disk for a quite a long time , and never had any permission issues with it . ( Disk is formatted as NT FS , if that helps . Capacity is 2 TB , of which 1 . 92 TB are available .) I 'd really appreciate if someone will give me a hint how this can be resolved . share \| im prove this question Try boot ing from the Recovery HD and see if the button is still disabled . – rien 333 Oct 7 ' 12 at 14 : 07 What exactly is it you need to do with the stuff on the NT FS disk ? – seg idd ins Oct 7 ' 12 at 15 : 26 @ Sam uel E .G idd ins I need to work with disk just as I 've worked yesterday , 2 days ago , 3 days ago etc . It looks like very annoying bug to me - to find out that out of a sudden this disk is considered to be read -only . – sh ab unc Oct 7 ' 12 at 15 : 34 do you have any add -ons that would |
| Layer #7 | <\|begin_of_text\|> Take the 2 -minute tour × Here what happened with me today . Time Machine asked me whether I want to set a backup disk , I 've answered yes , but then , when I 've realized that in order to backup anything Time Machine will clean the disk , I 've changed my mind and canceled everything . And my disk suddenly became read only . What I 've tried before Go og ling : $ sudo ch flags - R n ouch g Elements / $ sudo chmod - R a +w Elements / But I 've failed with both of this , getting " read -only file system " messages . What I 've tried after Go og ling : 1 . Open Disk Utilities 2 . Click Repair Disk Permissions But this button is disabled , and I have no idea what exactly should be done to enable it . I have been using this disk for a quite a long time , and never had any permission issues with it . ( Disk is formatted as NT FS , if that helps . Capacity is 2 TB , of which 1 . 92 TB are available .) I 'd really appreciate if someone will give me a hint how this can be resolved . share \| im prove this question Try boot ing from the Recovery HD and see if the button is still disabled . – rien 333 Oct 7 ' 12 at 14 : 07 What exactly is it you need to do with the stuff on the NT FS disk ? – seg idd ins Oct 7 ' 12 at 15 : 26 @ Sam uel E .G idd ins I need to work with disk just as I 've worked yesterday , 2 days ago , 3 days ago etc . It looks like very annoying bug to me - to find out that out of a sudden this disk is considered to be read -only . – sh ab unc Oct 7 ' 12 at 15 : 34 do you have any add -ons that would |

| Layer | Text |
|---|---|
| Layer #0 | Hol zer at j ess ica .h ol zer @d ow j ones .com <\|end_of_text\|> <\|begin_of_text\|> When the Associated Press releases its 200 9 preseason college football poll this weekend , it 's all but guaranteed that Florida will be ranked No . 1 —and that the G ators will collect a fat majority of the first -place votes . Florida already received 90 % of first -place votes in the recent coaches ' poll . Florida quarterback Tim Te bow Associated Press It 's no mystery why . The G ators are the defending B CS national champions and their quarterback , Tim Te bow , and their entire starting defense is coming back . Rare ly has a defending champion enjoyed such riches . But there 's some evidence that Florida 's fans should think twice before ordering their commem orative " Repeat Champions !" T -shirts . Bad O men for the G ators Here are the 10 teams that received the highest percentages of first -place votes in the preseason AP football poll since 196 8 —and how they finished the season . 1 . USC , 200 7 ( 95 . 4 %) 11 - 2 ( 3 rd ) 6 . Florida State , 199 1 ( 81 . 7 %) 11 - 2 ( 4 th ) 2 . USC , 200 5 ( 92 . 3 %) 12 - 1 ( 2 nd ) 7 . Ohio State , 196 9 ( 78 . 8 %) 8 - 1 ( 4 th ) 3 . Oklahoma , 198 7 ( 91 . 7 %) 11 - 1 ( 3 rd ) 8 . Nebraska , 199 6 ( 74 . 6 %) 11 - 2 ( 6 th ) 4 . Oklahoma , 197 5 ( 90 %) 11 - 1 ( 1 st ) 9 . USC , 197 9 ( 74 . 6 %) 11 - 0 - 1 ( 2 nd ) 5 . USC , 197 3 ( 87 . 3 %) 9 - 2 - 1 ( 8 th ) 10 . Oklahoma , 198 6 ( 74 . 6 %) 11 - 1 ( 3 rd ) Since 196 8 , 10 teams have received at least 75 % of the first -place votes in the preseason AP poll . Like Florida , seven of them were defending national champions . But only one of those teams , Oklahoma in 197 5 , actually won the national championship . None of these teams went undefeated —in fact , their average |
| Layer #6 | Hol zer at j ess ica .h ol zer @d ow j ones .com <\|end_of_text\|> <\|begin_of_text\|> When the Associated Press releases its 200 9 preseason college football poll this weekend , it 's all but guaranteed that Florida will be ranked No . 1 —and that the G ators will collect a fat majority of the first -place votes . Florida already received 90 % of first -place votes in the recent coaches ' poll . Florida quarterback Tim Te bow Associated Press It 's no mystery why . The G ators are the defending B CS national champions and their quarterback , Tim Te bow , and their entire starting defense is coming back . Rare ly has a defending champion enjoyed such riches . But there 's some evidence that Florida 's fans should think twice before ordering their commem orative " Repeat Champions !" T -shirts . Bad O men for the G ators Here are the 10 teams that received the highest percentages of first -place votes in the preseason AP football poll since 196 8 —and how they finished the season . 1 . USC , 200 7 ( 95 . 4 %) 11 - 2 ( 3 rd ) 6 . Florida State , 199 1 ( 81 . 7 %) 11 - 2 ( 4 th ) 2 . USC , 200 5 ( 92 . 3 %) 12 - 1 ( 2 nd ) 7 . Ohio State , 196 9 ( 78 . 8 %) 8 - 1 ( 4 th ) 3 . Oklahoma , 198 7 ( 91 . 7 %) 11 - 1 ( 3 rd ) 8 . Nebraska , 199 6 ( 74 . 6 %) 11 - 2 ( 6 th ) 4 . Oklahoma , 197 5 ( 90 %) 11 - 1 ( 1 st ) 9 . USC , 197 9 ( 74 . 6 %) 11 - 0 - 1 ( 2 nd ) 5 . USC , 197 3 ( 87 . 3 %) 9 - 2 - 1 ( 8 th ) 10 . Oklahoma , 198 6 ( 74 . 6 %) 11 - 1 ( 3 rd ) Since 196 8 , 10 teams have received at least 75 % of the first -place votes in the preseason AP poll . Like Florida , seven of them were defending national champions . But only one of those teams , Oklahoma in 197 5 , actually won the national championship . None of these teams went undefeated —in fact , their average |
| Layer #7 | Hol zer at j ess ica .h ol zer @d ow j ones .com <\|end_of_text\|> <\|begin_of_text\|> When the Associated Press releases its 200 9 preseason college football poll this weekend , it 's all but guaranteed that Florida will be ranked No . 1 —and that the G ators will collect a fat majority of the first -place votes . Florida already received 90 % of first -place votes in the recent coaches ' poll . Florida quarterback Tim Te bow Associated Press It 's no mystery why . The G ators are the defending B CS national champions and their quarterback , Tim Te bow , and their entire starting defense is coming back . Rare ly has a defending champion enjoyed such riches . But there 's some evidence that Florida 's fans should think twice before ordering their commem orative " Repeat Champions !" T -shirts . Bad O men for the G ators Here are the 10 teams that received the highest percentages of first -place votes in the preseason AP football poll since 196 8 —and how they finished the season . 1 . USC , 200 7 ( 95 . 4 %) 11 - 2 ( 3 rd ) 6 . Florida State , 199 1 ( 81 . 7 %) 11 - 2 ( 4 th ) 2 . USC , 200 5 ( 92 . 3 %) 12 - 1 ( 2 nd ) 7 . Ohio State , 196 9 ( 78 . 8 %) 8 - 1 ( 4 th ) 3 . Oklahoma , 198 7 ( 91 . 7 %) 11 - 1 ( 3 rd ) 8 . Nebraska , 199 6 ( 74 . 6 %) 11 - 2 ( 6 th ) 4 . Oklahoma , 197 5 ( 90 %) 11 - 1 ( 1 st ) 9 . USC , 197 9 ( 74 . 6 %) 11 - 0 - 1 ( 2 nd ) 5 . USC , 197 3 ( 87 . 3 %) 9 - 2 - 1 ( 8 th ) 10 . Oklahoma , 198 6 ( 74 . 6 %) 11 - 1 ( 3 rd ) Since 196 8 , 10 teams have received at least 75 % of the first -place votes in the preseason AP poll . Like Florida , seven of them were defending national champions . But only one of those teams , Oklahoma in 197 5 , actually won the national championship . None of these teams went undefeated —in fact , their average |

| Layer | Text |
|---|---|
| Layer #0 | team lacks a Rapid Spinner. And some Pokemon who fail to 2HKO my Pokemon will 2HKO them with enough hazards on my team which could lead to me being swept. It would also take care of my Aggron weakness. The problem is, I don't see anything I can replace it with ... if someone could give me a good enough reason on why I should replace it with "X" Pokemon or why I shouldn't, that would be great. · Threat List (Sorry Eo but I jacked this from your RMT ;____;) Red means this Pokemon is a big threat. Blue means this Pokemon is a moderate threat. Black means this Pokemon is easily handled. UU Threats  Absol - Milotic and Weezing can take a +2 Attack, I just have to hope it doesn't crit either of them, and hopefully I have entry hazards up so it dies quickly from LO. Aggron - This Pokemon is a MAJOR threat. I can't switch in anything on it, as it 2HKOes everything. If it comes in on Chansey / Clefable, I have to sacrifice them as I cant switch in Milotic and risk Aggron being Jolly and being 2HKOd. A smart player can keep switching it in and out until my special walls are gone. Alakaz am -Switch to Spiritomb, Pursuit it and it's KOed. Not a big threat at all. Chansey can take it on as well, even though a Specs Focus Blast is going to hurt. (Noob th und and your Zam nom :P <3) Altaria - Clefable can Encore / Trick DD variants, Milotic can Haze / Ice Beam DD variants as well. Support variants are handled by Clefable easily ; just Encore and Seismic Toss it until it's KOed. · Azumarill - Choice Band variants are easily handled by Milotic, Weezing, and Spiritomb. SubPunch variants can be Encored by Clefable and taken on by Weezing. · Blaziken - Milotic is my only |
| Layer #6 | team lacks a Rapid Spinner. And some Pokemon who fail to 2HKO my Pokemon will 2HKO them with enough hazards on my team which could lead to me being swept. It would also take care of my Aggron weakness. The problem is, I don't see anything I can replace it with ... if someone could give me a good enough reason on why I should replace it with "X" Pokemon or why I shouldn't, that would be great. · Threat List (Sorry Eo but I jacked this from your RMT ;____;) Red means this Pokemon is a big threat. Blue means this Pokemon is a moderate threat. Black means this Pokemon is easily handled. UU Threats  Absol - Milotic and Weezing can take a +2 Attack, I just have to hope it doesn't crit either of them, and hopefully I have entry hazards up so it dies quickly from LO. Aggron - This Pokemon is a MAJOR threat. I can't switch in anything on it, as it 2HKOes everything. If it comes in on Chansey / Clefable, I have to sacrifice them as I cant switch in Milotic and risk Aggron being Jolly and being 2HKOd. A smart player can keep switching it in and out until my special walls are gone. Alakaz am -Switch to Spiritomb, Pursuit it and it's KOed. Not a big threat at all. Chansey can take it on as well, even though a Specs Focus Blast is going to hurt. (Noob th und and your Zam nom :P <3) Altaria - Clefable can Encore / Trick DD variants, Milotic can Haze / Ice Beam DD variants as well. Support variants are handled by Clefable easily ; just Encore and Seismic Toss it until it's KOed. · Azumarill - Choice Band variants are easily handled by Milotic, Weezing, and Spiritomb. SubPunch variants can be Encored by Clefable and taken on by Weezing. · Blaziken - Milotic is my only |
| Layer #7 | team lacks a Rapid Spinner. And some Pokemon who fail to 2HKO my Pokemon will 2HKO them with enough hazards on my team which could lead to me being swept. It would also take care of my Aggron weakness. The problem is, I don't see anything I can replace it with ... if someone could give me a good enough reason on why I should replace it with "X" Pokemon or why I shouldn't, that would be great. · Threat List (Sorry Eo but I jacked this from your RMT ;____;) Red means this Pokemon is a big threat. Blue means this Pokemon is a moderate threat. Black means this Pokemon is easily handled. UU Threats  Absol - Milotic and Weezing can take a +2 Attack, I just have to hope it doesn't crit either of them, and hopefully I have entry hazards up so it dies quickly from LO. Aggron - This Pokemon is a MAJOR threat. I can't switch in anything on it, as it 2HKOes everything. If it comes in on Chansey / Clefable, I have to sacrifice them as I cant switch in Milotic and risk Aggron being Jolly and being 2HKOd. A smart player can keep switching it in and out until my special walls are gone. Alakaz am -Switch to Spiritomb, Pursuit it and it's KOed. Not a big threat at all. Chansey can take it on as well, even though a Specs Focus Blast is going to hurt. (Noob th und and your Zam nom :P <3) Altaria - Clefable can Encore / Trick DD variants, Milotic can Haze / Ice Beam DD variants as well. Support variants are handled by Clefable easily ; just Encore and Seismic Toss it until it's KOed. · Azumarill - Choice Band variants are easily handled by Milotic, Weezing, and Spiritomb. SubPunch variants can be Encored by Clefable and taken on by Weezing. · Blaziken - Milotic is my only |

