# OpenReview forum: "Hybrid Architectures for Language Models: Systematic Analysis and Design Insights"
_TMLR — Decision pending for TMLR_

### Review · Reviewer_CThX · 2026-05-10

**Summary Of Contributions:**

The paper presents an analysis of two hybridization strategies for combining self-attention with Mamba-2 state space models: inter-layer and intra-layer. The authors train models up to 3B parameters on 60B tokens, while most of their comparisons are made for ~1B parameter models. They evaluate the models across perplexity, few-shot accuracy, long-context retrieval and also assess integration as Mixture of Experts, and compute optimal scaling.

The main contributions are:
1. a controlled inter-vs-intra hybrid comparison under matched data and FLOPs budgets, which has not been done in prior work
2. thorough ablations on inter-layer block ratios and positioning, identifying that ~1:5 Transformer:Mamba ratio balances quality and efficiency and that Transformer blocks should never be placed in early layers;
3. a search over intra-hybrid block design choices (normalization, scaling, fusion operation, output projection, value sharing) yielding a variant that outperforms re-implementations of Hymba and Falcon-H1
4. a finding and mechanistic explanation for why transformer blocks should not be placed in early layers of hybrid models
5. demonstration that hybrid architectures are compatible with MoE and exhibit intermediate compute-optimal scaling behavior between Transformer and Mamba.

**Additional Comments:**

- I cannot assess the anonymized benchmarks tasks "A", "B", and "C". This will have to be clarified in the camera-ready version.

**Audience:**

Yes

**Audience Explanation:**

Hybrid architectures combining attention and state space models are an active area of LLM research, with multiple production systems (Jamba, Qwen Next, Nemotron Nano 2, Hunyuan-TurboS, IBM Granite 4.0, MiniMax-01) adopting this design. The community lacks systematic comparisons of design choices, and this submission contributes meaningfully on several fronts. I particularly find the finding of never placing transformer blocks at the front very insightful and valuable. Especially because it is supported by quantitative ablations as well as a mechanistic insight through attention-score visualizations. The intra-layer design space exploration is the most thorough such study published. The findings are most directly useful to practitioners and researchers working specifically with Mamba-2-based hybrids, which may be a substantial fraction of the hybrid architecture community. Readers should be aware that the conclusions are derived from a single linear primitive and may not transfer to hybrids built on Gated DeltaNet, RWKV-7, HGRN-2, Kimi Delta Attention, or other recent linear attention variants — but within the Mamba-2 family, this is a meaningful empirical contribution.

**Broader Impact Concerns:**

No concerns

**Claims And Evidence:**

Yes

**Claims Explanation:**

- the authors train models in FLOP-matched and data-matched regimes, which is methodologically sound and improves on prior work that typically reports only one of them.
- the intra-layer fusion design ablation is the most thorough such study I am aware of, and the resulting architecture beats prior intra-hybrid designs in controlled comparison.
- the scaling law experiments covering four model scales and five compute budgets are well-executed

**Requested Changes:**

- In the prior works section, the authors state that prior work hasn't done systematic ratio analysis. However, the paper by Wang et al, 2025 presents exactly this, across multiple different hybrid architectures. The prior works section should reflect this or clarify the claim that such systematic ration analysis has been done.
- The abstract and title do not reflect that this study focuses on only a single state space model variant, Mamba-2. The authors claim general contributions to hybrid models, but it is not clear how well the findings on Mamba-2 would generalize to other variants. Looking at the study by Wang et al, 2025, it seems that different token mixing layers can perform quite differently (e.g., GatedDeltaNet performs best with not hybridization while all others perform better as hybrid models). Moreover, to be called a "systematic analysis of hybrid architectures", I would expect more models and analysis. The focus of this work is to study intra-layer and inter-layer design choices of Mamba-2-based hybrid models.
- The paper refers to Mamba-2 often as Mamba but these two architectures are different. This should be made more clear, and it should be refered to as Mamba-2 or SSD, or something that uniquely identifies the model variant used.
- the ablation in table 6 uses a 1:0 block ratio but the final architecture used in the main results uses a 1:5 ratio. design choices that win at 1:0 may not be optimal for 1:5, this should at least be mentioned.
- the inter-layer block-ratio ablations (tables 5, 16, 17) compare configurations at matched token budgets but not matched FLOPs budgets; FLOPs vary by up to ~22% across rows, which confounds the ratio-vs-quality conclusions. While the inference throughput confound was mentioned, the increase in training FLOPs was not raised.
- using RULER would be more rigorous than the simple Needle-in-a-Haystack benchmark.

---

> ### Author Response · Authors · 2026-06-08
> **Response to Reviewer CThX (1/2)**
>
> We thank the reviewer for their careful and constructive feedback. We appreciate that the reviewer recognized the value of our controlled inter-vs-intra hybrid comparison, extensive design ablations, and mechanistic analysis of block placement, as well as the relevance of our findings for practitioners working on Mamba-2-based hybrid architectures.
>
> We address the reviewer’s remaining concerns below and will incorporate the corresponding clarifications and revisions into the manuscript.
>
> ---
>
> > **[W1]** The prior-work discussion should better acknowledge Wang et al. (2025), which also performs systematic ratio analysis for hybrid architectures.
>
> **[A1]** We acknowledge that Wang et al. (2025) [1] ablated linear-to-full block ratios within inter-layer hybridization, particularly in Section 4.2 of their paper. In the revised manuscript, we will accurately reflect this prior work in the RQ1 paragraph in Section 3.1. Furthermore, we will update the experimental discussion to explain how the U-shaped sweet spot identified in our inter-layer block-ratio ablations aligns with the trends observed by Wang et al. across different linear architectures. We will also clarify that the exact optimal ratio can depend on the specific linear/SSM primitive and architectural setup.
>
> ---
>
> > **[W2]** The title, abstract, and framing should make clear that the study focuses on Mamba-2-based hybrid models rather than hybrid architectures in general.
>
> **[A2]** We agree with the reviewer that our empirical analysis is grounded in Mamba-2, and that the initial framing may sound broader than what our experiments directly validate. We will revise the abstract, introduction, and methodology sections to explicitly state that this work studies intra-layer and inter-layer design choices for Transformer/Mamba-2 hybrid models, so that they more accurately reflect the Mamba-2-based design space studied in this paper.
>
> We also acknowledge the reviewer’s point, consistent with Wang et al. (2025) [1], that different subquadratic token-mixing layers can behave differently. Therefore, we will clarify that the design recipes identified in this work should be interpreted as empirically validated within the Transformer/Mamba-2 hybrid design space. A full validation across additional subquadratic architectures remains an important direction for future work.
>
> As an initial step in this direction, we are currently conducting further experiments by applying the design recipe identified in our study to a hybrid architecture based on Gated DeltaNet [2]. We will include these additional results in the revised manuscript to provide further validation beyond the Transformer/Mamba-2 hybrid setting, while clearly distinguishing this from a comprehensive validation over all subquadratic architectures.
>
> ---
>
> > **[W3]** The manuscript sometimes refers to Mamba-2 simply as “Mamba,” although these are different architectures.
>
> **[A3]** We thank the reviewer for catching this issue. In the revised manuscript, we will carefully replace generic references to “Mamba” with “Mamba-2” wherever we refer to our model variant. This will ensure that the studied architecture is accurately identified throughout the paper.
>
> ---
>
> > **[W4]** The intra-layer ablation in Table 6 is conducted at a 1:0 block ratio, while the final main architecture uses a 1:5 ratio, so the optimal design choices may not transfer perfectly.
>
> **[A4]** We thank the reviewer for this important observation. Given the large intra-layer design space, we conducted the Table 6 ablation under the simplifying assumption that design choices optimized in the 1:0 setting would remain robust when combined with other block ratios, including the final 1:5 setting.
>
> We agree that this is a strong assumption, and that the choices selected at 1:0 may not be strictly optimal for the 1:5 architecture. In the revised manuscript, we will explicitly discuss this as a limitation of our design search. At the same time, our empirical results suggest that the final 1:5 intra-layer hybrid still outperforms standard Llama, Mamba-2, and inter-layer hybrid baselines in our main comparisons. We will clarify this point while avoiding any claim that the 1:0-selected design is guaranteed to be globally optimal for every ratio.

---

> ### Author Response · Authors · 2026-06-08
> **Response to Reviewer CThX (2/2)**
>
> > **[W5]** The inter-layer block-ratio ablations are token-budget matched but not FLOP-budget matched, so training FLOPs vary across rows and may confound the ratio-quality conclusion.
>
> **[A5]** We acknowledge that the inter-layer block-ratio ablations in Tables 5, 16, and 17 are matched by token budget rather than by total training FLOPs, and therefore should not be interpreted as fully FLOP-normalized comparisons.
>
> In the revised manuscript, we will explicitly clarify this distinction. Under the token-matched setting, the 1:1 ratio achieves the best quality in our experiments. However, because Mamba-2-heavy configurations require fewer FLOPs per token, a configuration such as the 1:5 ratio could process more tokens under a fixed FLOP budget. As a result, under a fully FLOP-matched training setup, Mamba-2-heavy ratios may narrow the gap with the 1:1 ratio.
>
> We will also explicitly refer readers to Figure 1b, which provides the ratio-quality comparison under a matched training FLOP budget. Finally, we will add this discussion to the revised manuscript and, where feasible, report or annotate the relative training FLOPs for the ratio-ablation tables to make the comparison clearer.
>
> ---
>
> > **[W6]** RULER would provide a more rigorous long-context evaluation than the simple Needle-in-a-Haystack benchmark.
>
> **[A6]** Results on RULER would provide a more comprehensive and rigorous evaluation of long-context capabilities. However, in this work, our models were trained with a limited context length of 8K, up to 3B parameters, and approximately 60B pretraining tokens, which constrained the scope of our long-context evaluation.
>
> For this reason, we used Needle-in-a-Haystack as a lightweight diagnostic benchmark for in-context retrieval behavior. In the revised manuscript, we will explicitly note this limitation and clarify that a more comprehensive long-context evaluation using RULER is an important direction for future work.
>
> ---
>
> > **[W7]** The anonymized benchmark tasks “A,” “B,” and “C” are difficult to assess and should be clarified in the camera-ready version.
>
> **[A7]**  Due to strict data confidentiality agreements, we cannot disclose the exact names of these benchmarks. However, we agree that the current anonymized presentation may make the results difficult to interpret. In the camera-ready version, we will either clarify these benchmarks by describing their domains, task formats, evaluation protocols, and statistical properties, or remove the corresponding results if they cannot be made sufficiently interpretable.
>
> ---
>
> **References:**
>
> [1] Wang, Dustin, et al. "A systematic analysis of hybrid linear attention." arXiv preprint arXiv:2507.06457 (2025).
>
> [2] Yang, Songlin, Jan Kautz, and Ali Hatamizadeh. "Gated delta networks: Improving mamba2 with delta rule." International Conference on Learning Representations. 2025.

---

### Review · Reviewer_3VNT · 2026-05-27

**Summary Of Contributions:**

This paper presents a systematic study of hybrid language model architectures that combine full attention and SSM layers. The paper concurrently explores two major hybrid paradigms, namely inter-layer and intra-layer. The paper consolidates the community's understanding that hybrid architectures outperform homogeneous baselines and offers practical design insights for model developers.

**Audience:**

Yes

**Audience Explanation:**

The paper addresses a highly timely topic of efficient architectural design. Efficient attention variants like SSMs are rapidly gaining traction in mainstream models (e.g., Qwen 3.5), and the community lacks systematic guidance on the design principles.

**Claims And Evidence:**

Yes

**Claims Explanation:**

Overall, yes. I liked the extensiveness of the empirical evaluations. This work fixed all models' number of parameters and training FLOPs to be the same, which is critical for an unbiased comparison between model architectures. The ablation studies are also extensive. That said, a few limitations remain that weaken the broader conclusions (although the existing experiments are generally strong enough for the paper's claims)
- Limited scale: as acknowledged by the authors, most experiments are conducted with small models (~3B params) and only pretrained on 60B tokens. As such, it remains unclear if the observed advantages will generalize to larger models. However, I also understand that these experiments are resource-intensive and difficult to carry out.
- Downstream benchmark diversity. The eval benchmarks is somewhat narrow, and the paper would be strengthened with broader reasoning and instruction-following evaluations (e.g., MMLU-style reasoning, coding, agentic applications, etc.)

**Requested Changes:**

Overall, I like this work quite a lot, and while I'm leaning toward accepting this paper, I believe addressing the following comments would improve the quality of this work.
- [Critical] Question: Table 5 suggests that a 1:1 Transformer:SSM ratio achieves the best quality. However, in prior work, "An empirical study of mamba-based language models" (Fig. 4), the conclusion was that the validation loss is the lowest when roughly 8% of the layers are attention layers. While the difference partially stems from the different evaluation metrics and experimental setups, I encourage the authors to discuss the deeper architectural reasons behind this discrepancy.
- [Critical] Follow-up comment: The paper would benefit from a more explicit comparison with Waleffe et al., An Empirical Study of Mamba-based Language Models. That work also studies Mamba/Mamba-2 hybrids and reaches several conclusions that appear to overlap with this submission, including that pure SSM models have weaknesses on in-context learning or retrieval and that adding attention layers can substantially improve performance while retaining efficiency benefits. Could the authors explicitly discuss which conclusions are shared between the two papers and which are new or different here? This would fit naturally in Section 5, “Distinctions from Prior Works.” In particular, it would be helpful to clarify differences in experimental regime, model scale, hybrid design space, block-ratio findings, placement strategy, downstream/long-context evaluations, and the role of intra-layer hybrids, which are a more central focus of this submission.
- [Critical] Throughput is a key metric throughout the paper. However, it depends heavily on the underlying implementation (e.g., kernels). Is throughput measured on the Transformers backend? The paper should include more discussions on these underlying details. I apologize in advance if I missed it in the paper.
- In the discussion of other subquadratic attention variants (e.g., RWKV, GDN), the authors claim that "Furthermore, recent findings indicate that Mamba2 actually outperforms GDN when its A and dt_bias parameters are randomly and correctly initialized" -- is there a citation for this claim? On a high level, the paper could benefit from a more in-depth articulation of which findings are Mamba-specific and which are expected to generalize.
- Most results only report the average, not variance or confidence intervals across runs. Including those results would improve the statistical significance of the results.
- Format and grammar nits (non-exhaustive). I recommend that the authors use Grammarly to take a full pass over the camera-ready version of the paper.
    + The line after the formula in Section 2 overlaps with the line below.
    + "have not been clearly shared to the community" -> "have not been clearly shared with the community"
    + "one of key architectural considerations" -> "one of the key architectural considerations"
    + "FLOPs reduction in hybrid models translate into faster" -> "FLOPs reduction in hybrid models translates into faster"
    + "hybrid models leverage these advantage" -> "hybrid models leverage this advantage"
    + "hybrid-Mamba models extrapolates well" -> "hybrid-Mamba models extrapolate well"
    + "Hybrid architectures demonstrates" -> "Hybrid architectures demonstrate"
    + "being a little slightly more data-hungry"
    + "These finding support prior work" -> "These findings support prior work"

---

> ### Author Response · Authors · 2026-06-08
> **Response to Reviewer 3VNT (1/3)**
>
> We thank the reviewer for their careful and constructive feedback. We appreciate that the reviewer recognized the relevance of our study, the extensiveness of our empirical evaluations, and the importance of comparing architectures under matched parameter counts and training FLOPs. We also appreciate the reviewer’s positive assessment of our systematic analyses of inter-layer and intra-layer Transformer/SSM hybrids, design ablations, and practical guidance for efficient architecture design.
>
> We address the reviewer’s remaining comments below and will incorporate the corresponding clarifications and revisions into the manuscript.
>
> ---
>
> > **[W1]** The experiments are mostly limited to models up to around 3B parameters and 60B pretraining tokens, so it remains unclear whether the observed trends generalize to larger-scale models.
>
> **[A1]** We sincerely thank the reviewer for their understanding regarding the computational constraints of this study. We fully acknowledge that our experiments are limited in model scale and training tokens compared to frontier-scale language models. As the reviewer noted, scaling up pretraining experiments is highly resource-intensive. In our case, this challenge was further amplified because our goal was to comprehensively ablate a wide configuration space across both inter-layer and intra-layer hybrid designs. The combinatorial expansion of ratios, placements, fusion strategies, and dimension splits made it practically infeasible to evaluate all design choices at a much larger scale.
>
> Nevertheless, the performance differences among architectural choices are consistent and meaningful up to 3B scales, suggesting that the identified design principles may remain informative for larger models. We will make this limitation explicit in the revised manuscript and discuss larger-scale verification as an important direction for future work.
>
> ---
>
> > **[W2]** The downstream benchmarks are somewhat narrow, and broader reasoning, coding, instruction-following, or agentic evaluations would strengthen the paper.
>
> **[A2]** We agree with the reviewer that broader evaluations on advanced reasoning, instruction-following, and agentic tasks would further strengthen the paper. We would like to clarify, however, that our current evaluation suite already covers several complementary aspects of model behavior, including few-shot benchmarks, downstream QA and summarization tasks, and in-context retrieval. These evaluations were chosen to isolate architecture-level differences under controlled pretraining and compute budgets.
>
> A comprehensive evaluation of reasoning and agentic capabilities would likely require larger-scale models, longer pretraining, and additional post-training stages. We therefore view this as an important next step beyond the scope of the current study. We will explicitly mention this limitation and future direction in the revised manuscript.
>
> ---
>
> > **[W3]** Table 5 suggests that a 1:1 Transformer:SSM ratio works best, while Waleffe et al. report that validation loss is lowest when only about 8% of layers are attention layers. The paper should discuss the architectural reasons behind this discrepancy.
>
> **[A3]** We thank the reviewer for this insightful comment. We believe the discrepancy mainly comes from differences in how the hybrid blocks are structurally defined, especially regarding the placement and integration of MLP blocks.
> In Waleffe et al. [1], Mamba, Attention, and MLP are treated as decoupled computational primitives. Their architecture starts and ends with Mamba layers, interleaves Attention layers, and distributes MLPs across Mamba positions. In contrast, our architecture treats both Transformer and Mamba-2 layers as integrated blocks that include their own MLP components. Therefore, the reported Transformer:SSM ratio in our setting reflects a different block-level design space from the attention-layer ratio analyzed in Waleffe et al.
>
> Under this integrated block definition, our experiments show that a balanced 1:1 ratio yields the best performance at both the 350M and 1B scales. This also aligns with other hybrid architectures that use less skewed ratios, such as 1:3, rather than extremely sparse attention placement [2,3]. We will add this discussion to the revised manuscript to clarify why the optimal ratio differs across these architectural settings.

---

> ### Author Response · Authors · 2026-06-08
> **Response to Reviewer 3VNT (2/3)**
>
> > **[W4]** The paper should more explicitly compare with Waleffe et al., including which conclusions are shared and which contributions are new or different in this work.
>
> **[A4]** A clearer comparison with Waleffe et al. [1] would improve the positioning of our work, and we will expand the discussion in Section 5 accordingly. We share several key conclusions with Waleffe et al. In particular, both studies observe that pure SSM models have weaknesses in in-context learning and retrieval, and that adding Attention layers can substantially mitigate these weaknesses while retaining the efficiency benefits of SSM-based models.
>
> However, our work differs in both scope and focus. While Waleffe et al. provide a foundational study of Mamba/Transformer hybrids, our goal is to systematically identify effective design choices for both inter-layer and intra-layer hybrid architectures. In particular, our work provides more fine-grained ablations over block ratios, jointly analyzes the effect of block placement, compares inter-layer hybrids with recently developed intra-layer hybrid designs, and studies the tradeoffs between these two hybrid paradigms. We also include controlled quality-efficiency comparisons, long-context retrieval behavior, and preliminary scaling and MoE-compatibility analyses. We will revise the manuscript to more clearly distinguish which conclusions are consistent with prior work and which findings are new to this study.
>
> ---
>
> > **[W5]** Throughput is a key metric, but it depends heavily on implementation details such as kernels and inference backends. The paper should provide more details on how throughput was measured.
>
> **[A5]** We thank the reviewer for pointing this out, and we apologize for not providing sufficient implementation details in the current version. Throughput in our paper was measured using optimized inference implementations for each architecture, rather than relying only on a generic backend.
>
> Our general inference pipeline uses sequence packing, `torch.compile`, and a standard two-stage generation procedure consisting of a single prefill phase followed by token-by-token decoding. For Transformer modules, we use a static KV cache that is compatible with compiling. During prefill, we use FlexAttention with a block causal attention mask, while during decoding, we use standard Scaled Dot-Product Attention (SDPA). For SWA models, we further reduce the KV-cache footprint and optimize the cache-update mechanism accordingly.
>
> For SSM/Mamba modules, we use the optimized original Mamba implementation. The convolution and state caches are initialized during the prefill phase and then iteratively updated during decoding. We also apply input packing during the prefill phase, using optimized custom kernels for variable-length processing in Mamba. We will add these implementation details to the revised manuscript so that the throughput results can be interpreted more clearly and reproducibly.
>
> ---
>
> > **[W6]** The claim that Mamba-2 outperforms GDN under certain initialization settings needs a citation, and the paper should better clarify which findings are Mamba-specific versus more broadly generalizable.
>
> **[A6]** We thank the reviewer for raising this point. The claim was based on recent empirical findings documented in the official Flash Linear Attention repository, specifically Pull Request #739 [4]. In our experiments, we followed the initialization discussion from this source: we initialized the `dt_bias` parameters using the same random-value strategy, and applied appropriate random initialization to the `A` parameter independently across different heads. In the revised manuscript, we will formally cite this repository discussion and explicitly clarify the corresponding initialization details.
>
> We also agree that the paper would benefit from a clearer distinction between Mamba-2-specific findings and more general hybrid-architecture trends. We will clarify that the design recipes identified in this work should be interpreted as empirically validated within the Transformer/Mamba-2 hybrid design space studied in our experiments. A full validation across additional subquadratic architectures remains an important direction for future work.
>
> As an initial step in this direction, we are currently conducting further experiments by applying the design recipe identified in our study to a hybrid architecture based on Gated DeltaNet [5]. We will include these additional results in the revised manuscript to provide further validation beyond the Transformer/Mamba-2 hybrid setting, while still clearly distinguishing this from a comprehensive validation over all subquadratic architectures.

---

> ### Author Response · Authors · 2026-06-08
> **Response to Reviewer 3VNT (3/3)**
>
> > **[W7]** Most results report averages without variance or confidence intervals, so additional uncertainty estimates would improve statistical reliability.
>
> **[A7]** Due to the high computational cost of pretraining, conducting repeated runs for all configurations is prohibitively expensive. However, we re-checked cases where multiple independent runs were conducted, including the Transformer baseline and several inter-layer hybrid configurations.
>
> Empirically, under our controlled setup, the maximum discrepancy in DCLM validation loss across runs was at most **0.008**. This suggests that the main observed gaps are larger than typical run-to-run variation in our experiments. Although we do not have repeated runs for every configuration, all architectures were trained with the same corpus ordering and identical hyperparameters, which provides a fair basis for comparison across architectures. We will include this clarification and the observed repeated-run variance details in the revised manuscript where feasible.
>
> ---
>
> > **[W8]** The paper contains several formatting and grammar issues that should be fixed before the camera-ready version.
>
> **[A8]** We sincerely thank the reviewer for their careful reading. We will correct all of the mentioned issues identified by the reviewer. We will also conduct a full proofreading pass over the camera-ready version to improve clarity, grammar, and formatting consistency throughout the manuscript.
>
> ---
>
> **References:**
>
> [1] Waleffe, Roger, et al. "An empirical study of mamba-based language models." arXiv preprint arXiv:2406.07887 (2024).
>
> [2] Lieber, Opher, et al. "Jamba: A hybrid transformer-mamba language model." arXiv preprint arXiv:2403.19887 (2024).
>
> [3] Wang, Dustin, et al. "A systematic analysis of hybrid linear attention." arXiv preprint arXiv:2507.06457 (2025).
>
> [4] fla-org. "Flash Linear Attention, Pull Request #739." GitHub, https://github.com/fla-org/flash-linear-attention/pull/739.
>
> [5] Yang, Songlin, Jan Kautz, and Ali Hatamizadeh. "Gated delta networks: Improving mamba2 with delta rule." International Conference on Learning Representations. Vol. 2025. 2025.

---

### Review · Reviewer_sAVn · 2026-05-28

**Summary Of Contributions:**

This paper provides a study of transformer-mamba hybrid language models, focusing on two integration strategies: (1) inter-layer hybrids, where transformer and mamba blocks are mixed across depth, and (2) intra-layer hybrids, where the two primitives are fused within individual layers. The main contributions are (i) controlled quality/efficiency comparisons against transformer, mamba, and SWA baselines, (ii) ablations identifying useful design choices for ratios, placement, fusion, and dimension splits, (iii) evidence that hybrids improve the tested quality-throughput tradeoff and long-context retrieval behavior, and (iv) preliminary scaling and MoE-compatibility analyses up to 3B parameters.

**Additional Comments:**

Minor: I noticed a few reference/citation cleanup issues. Some model names appear without clear author/year citations, e.g. Qwen3-Next, IBM Granite 4.0, Dragon, and Llama 4, and there also seems to be a typo in "OpenaAI et al." I did not check this exhaustively, but it would be worth doing one pass over the references and model-name citations before publication.

**Audience:**

Yes

**Audience Explanation:**

This paper contributes interesting/relevant findings that are usually withheld, omitted, or never considered in papers proposing new LLM architecture designs.

**Broader Impact Concerns:**

(no concerns)

**Claims And Evidence:**

No

**Claims Explanation:**

"No", but due to minor presentation issues:

One claim that seems stronger than the evidence is the statement that hybrid models consistently match or outperform transformers on downstream tasks. Table 3 looks more mixed: the hybrid models clearly avoid the large drops seen for pure mamba or SWA, but they are not uniformly better than transformer baseline across the reported tasks. I think the result would be more accurately described as hybrids largely preserving transformer-like downstream performance while improving efficiency, rather than consistently outperforming the transformers.

I also found the "optimal design recipes" framing a bit too broad. The ablations are useful and give practical guidance, but the evidence is still limited to a specific family of transformer/mamba-2 hybrids, mostly around the 350M-1B scale, with scaling experiments up to 3B. I would suggest qualifying these claims as optimal within the tested design space, rather than as general recipes for hybrid language models.

**Requested Changes:**

On top of fixing the angle/presentation of the claims mentioned above, there are other points that could be addressed:

The paper focuses mainly on pretraining and limited fine-tuning evaluations. I think it would be useful to briefly discuss whether the identified architecture trends are expected to transfer to post-training regimes such as RLHF/RLVR for heavy reasoning tasks. In these settings, efficiency bottlenecks can be different: long rollouts, many sampled completions, verifier/reward calls, cache size, and long-generation stability may matter more than pretraining FLOPs alone. I do not think full post-training experiments are necessary for this paper, but a short discussion would help clarify how the proposed design recipes should be interpreted for research into modern reasoning-oriented models.

Some of the reported differences between architectures are relatively small, especially in the main quality comparisons. I do not think this undermines the results, but it would be helpful to include light uncertainty estimates where feasible, such as confidence intervals over evaluation examples or repeated runs for a small subset of the key 1B comparisons. This would make it easier to tell which improvements are robust versus likely within normal evaluation or training variance. Even limited uncertainty reporting for the main tables would strengthen the paper without requiring a much larger experimental sweep.

I found the efficiency analysis useful, but I think the interpretation could be qualified more carefully. Section 4.2 emphasizes FLOPs reductions and measured step-time gains, but mamba/hybrid models also appear to have somewhat different training-memory requirements. Since memory affects feasible batch size, activation checkpointing, recomputation, and parallelization choices, the current results are best interpreted as a fixed-system comparison rather than a fully resource-normalized efficiency comparison. It would be helpful for the authors to briefly discuss whether the conclusions are expected to hold under different memory budgets or parallelization strategies, especially since the reported measurements are from a specific 8xH200/FSDP setup and some intra-hybrid speedups depend on additional parallelism.

---

> ### Author Response · Authors · 2026-06-08
> **Response to Reviewer sAVn (1/2)**
>
> We thank the reviewer for their careful and constructive feedback. We appreciate that the reviewer recognized the relevance of our study to the TMLR audience and the value of providing controlled empirical evidence for Transformer-Mamba-2 hybrid architectures, including quality-efficiency comparisons, design ablations, long-context behavior, and preliminary scaling/MoE analyses.
>
> We address the reviewer’s remaining concerns below and will incorporate the corresponding clarifications and revisions into the manuscript.
>
> ---
>
> > **[W1]** The claim that hybrid models “consistently match or outperform” Transformers appears too strong, since Table 3 shows mixed downstream results.
>
> **[A1]** We sincerely thank the reviewer for pointing this out. Our intended meaning was that the hybrid models generally *match or outperform* the standard Transformer baseline across the reported downstream tasks. However, we agree that this phrasing could be misleading, since Table 3 does not show uniform improvements over the Transformer baseline on every task. In the revised manuscript, we will clarify that the main takeaway is that hybrid models largely preserve Transformer-like downstream performance while improving computational efficiency, rather than uniformly outperforming Transformer baselines across all tasks.
>
> ---
>
> > **[W2]** The “optimal design recipes” framing should be qualified, since the evidence is limited to the tested Transformer/Mamba-2 hybrid design space and model scales.
>
> **[A2]** We appreciate this constructive suggestion. We fully acknowledge that presenting our findings as general optimal design recipes for all hybrid language models is too broad, especially given that our empirical evidence is based on a specific family at a relatively smaller scale. We will rephrase our conclusions to accurately reflect that these optimal configurations and practical guidance apply specifically within our tested design space, rather than presenting them as universal recipes. This better reflects the scope of our experiments while preserving the practical guidance provided by our ablations.
>
> In addition, we are currently conducting further experiments by applying the design recipe identified in our study to a hybrid architecture based on Gated DeltaNet [1]. We will include these additional results in the revised manuscript to provide further validation beyond the Transformer/Mamba-2 hybrid setting.
>
> ---
>
> > **[W3]** The paper would benefit from a short discussion on whether the observed architecture trends may transfer to post-training regimes such as RLHF/RLVR for reasoning-oriented models.
>
> **[A3]** We thank the reviewer for raising this great point. While our experimental scope focused on pretraining and limited fine-tuning, we agree that discussing post-training implications would improve the paper. In the revised manuscript, we will add a brief discussion clarifying that the inherent efficiency of hybrid architectures—particularly regarding reduced KV cache memory footprint and lower latency—should provide substantial benefits in post-training regimes like RLVR, where massive rollout generations and verifier calls (e.g., LLM-as-a-judge) are major bottlenecks.
>
> Meanwhile, validating the long-horizon generation stability of these hybrids for complex reasoning tasks, compared to pure Transformers, remains an essential area for future work. Nevertheless, given the recent successes of various hybrid reasoning models [2,3,4,5] in post-training settings, we believe that the structural benefits identified in our pretraining experiments may also be relevant in such regimes.
>
> ---
>
> > **[W4]** Some quality differences are relatively small, so light uncertainty estimates or repeated-run evidence would help clarify robustness.
>
> **[A4]** We agree that adding uncertainty-related evidence would make the empirical comparison clearer. To examine this, we re-checked cases where multiple independent runs were conducted, including the Transformer baseline and several inter-layer hybrid configurations. The maximum discrepancy in DCLM validation loss across these runs was at most **0.008**, suggesting that the main observed gaps are larger than typical run-to-run variation in our setup. Although we do not have repeated runs for every configuration, all architectures were trained with the same corpus ordering and identical hyperparameters. Therefore, we believe these repeated-run results provide a reasonable estimate of the training variance in our main experiments. We will include this clarification and the observed variance details in the revised manuscript where feasible.

---

> ### Author Response · Authors · 2026-06-08
> **Response to Reviewer sAVn (2/2)**
>
> > **[W5]** The efficiency analysis should be qualified as a fixed-system comparison, since Mamba/hybrid models may have different training-memory and parallelization behavior.
>
> **[A5]** We appreciate this insightful comment. The higher training-memory requirements of Mamba-2 modules can influence the feasible batch size and parallelization choices. We will add a brief discussion addressing hardware dynamics. Specifically, while our current 8xH200/FSDP environment omitted tensor parallelism (TP) due to the limited node scale, Mamba-2 is compatible with tensor parallelism, suggesting that the observed scalability trends may extend to other parallelization layouts. Furthermore, we will note that Mamba's activation storage can be demanding—potentially giving fully optimized Transformers a slight relative advantage over our current results in a fully resource-normalized sweep—while this bottleneck may be mitigated through activation checkpointing and optimized kernel fusion. Finally, we will clarify that while inter-layer hybrids already have memory profiles close to Transformers, the peak memory spikes in intra-layer hybrids may be further mitigated via expert parallelism (EP).
>
> ---
>
> > **[W6]** Some model references and citations need cleanup, including missing citations for recent models and a typo in “OpenaAI et al.”
>
> **[A6]** We thank the reviewer for their meticulous reading and for pointing out these reference issues. For models that currently lack formal technical reports, we omitted standard paper citations in the current version; however, we will include URL links or repository references for them in the revised version. Regarding "OpenaAI et al.", this typo occurred while shortening a lengthy author list using the organization's name. We will correct this error and thoroughly review all model citations and references to ensure they are clean and accurate for the final publication.
>
> ---
>
> **References:**
>
> [1] Yang, Songlin, Jan Kautz, and Ali Hatamizadeh. "Gated delta networks: Improving mamba2 with delta rule." International Conference on Learning Representations. Vol. 2025. 2025.
>
> [2] Qwen Team. "Qwen3-Next." Qwen Blog, https://qwen.ai/blog?id=4074cca80393150c248e508aa62983f9cb7d27cd (2025).
>
> [3] Team, Kimi, et al. "Kimi linear: An expressive, efficient attention architecture." arXiv preprint arXiv:2510.26692 (2025).
>
> [4] Li, Aonian, et al. "Minimax-01: Scaling foundation models with lightning attention." arXiv preprint arXiv:2501.08313 (2025).
>
> [5] Basant, Aarti, et al. "Nvidia nemotron nano 2: An accurate and efficient hybrid mamba-transformer reasoning model." arXiv preprint arXiv:2508.14444 (2025).

---

### Decision · Action_Editor_8BYd · 2026-06-29

**Recommendation:** Accept with minor revision

**Audience:**

Yes

**Audience Explanation:**

As I mentioned in the previous answer, the submission contains a highly relevant and very comprehensive empirical study of hybrid architectures, as all reviewers have agreed.

**Claims And Evidence:**

Yes

**Claims Explanation:**

The submission is an extensive empirical study of hybrid Transformer and Mamba-2 architecture for language modelling. The authors compare inter-layer and intra-layer hybridization with controlled data and compute settings, as well as other ablations.

The reviewers broadly agree that the paper is useful for the TMLR audience, and the empirical study is very comprehensive. All reviewers ultimately recommended acceptance.

The remaining concerns are regarding scope and presentation rather than any technical errors. For example, the paper should
 - clearly frame its claims as applying to tested Transformer/Mamba-2 design space
 - soften claims of uniform downstream improvement
 - distinguish token-match from FLOP-matched ratio ablations
 - provide implementation details for throughput
 - clarify or remove anonymized downstream tasks

I recommend acceptance conditional on these revisions being incorporated in the final manuscript.